# Leishmania protein KMP-11 modulates cholesterol transport and membrane fluidity to facilitate host cell invasion

Achinta Sannigrahi[1,7], Souradeepa Ghosh [2], Supratim Pradhan[2], Pulak Jana [1], Junaid Jibran Jawed[3], Subrata Majumdar[4], Syamal Roy[5,8], Sanat Karmakar [6], Budhaditya Mukherjee [2✉] & Krishnananda Chattopadhyay [1✉]

## Abstract

**The first step of successful infection by any intracellular pathogen relies on its ability to invade its host cell membrane. However, the detailed structural and molecular understanding underlying lipid membrane modification during pathogenic invasion remains unclear. In this study, we show that a specific *Leishmania donovani* (LD) protein, KMP-11, forms oligomers that bridge LD and host macrophage (MΦ) membranes. This KMP-11 induced interaction between LD and MΦ depends on the variations in cholesterol (CHOL) and ergosterol (ERG) contents in their respective membranes. These variations are crucial for the subsequent steps of invasion, including (a) the initial attachment, (b) CHOL transport from MΦ to LD, and (c) detachment of LD from the initial point of contact through a liquid ordered (Lo) to liquid disordered (Ld) membrane-phase transition. To validate the importance of KMP-11, we generate KMP-11 depleted LD, which failed to attach and invade host MΦ. Through tryptophan-scanning mutagenesis and synthesized peptides, we develop a generalized mathematical model, which demonstrates that the hydrophobic moment and the symmetry sequence code at the membrane interacting protein domain are key factors in facilitating the membrane phase transition and, consequently, the host cell infection process by Leishmania parasites.**

**Keywords** Phospholipid Membranes; Leishmania; Host–Pathogen Interaction; Phase-transition; Cholesterol Transport
**Subject Categories** Membranes & Trafficking; Microbiology, Virology & Host Pathogen Interaction

## Introduction

The successful infection of any intracellular pathogen relies on its efficient invasion through the host cell membrane, which strongly depends on host membrane properties (morphology, fluidity, and rigidity). It is well established that the host membrane morphology (e.g., fluidity/rigidity) alters during the process of infection of various intracellular pathogens (Cossart and Roy, 2010; Mishra et al, 2019; Sherling and van Ooij, 2016), which facilitates particle uptake by lowering the free energy barrier of the internalization process (Ayala et al, 2023). In this context, lipid unsaturation (Beney and Gervais, 2001) and cholesterol (CHOL) contents play crucial roles (Yèagle, 1989), by modulating membrane fluidity, and stiffness at the interface of host–pathogen interaction (Canepa et al, 2021). The presence of CHOL influences the structural properties of biological membranes and is central to the organization, dynamics, function, and sorting of lipid bilayers in vivo (Simons and Ikonen, 2000). Since CHOL has become an attractive target for many pathogens via which they can influence host cell dynamics, several pathogens have developed ingenious ways to use CHOL towards recognizing and interacting with host cell membranes, including the use of specific machinery to alter lipid conformation and CHOL content during the invasion process to ensure successful infections. In some cases, there are CHOL quenching events during the onset of infection, which are facilitated by pathogen-derived proteins that transfer CHOL (Correa et al, 2021; Pradhan et al, 2021). Since the accessibility of CHOL in the host membrane is crucial for CHOL-scavenging pathogens to transport from the host to the pathogen, there is a requirement of disrupting the sequestered (non-accessible) CHOL complex in host cell membrane, thereby increasing CHOL accessibility (Abrams et al, 2020; Palladino et al, 2022; Zhang et al, 2009). Although these events are known to exist for multiple intracellular pathogens (Correa et al, 2021; Lige et al, 2009; Nazarova et al, 2017) such as *Mycobacterium*, *Toxoplasma*, *Plasmodium*, *Leishmania*, and even for viruses like

[1]Structural Biology & Bio-Informatics Division, CSIR-Indian Institute of Chemical Biology, 4, Raja S. C. Mallick Road, Kolkata, West Bengal 700032, India. [2]School of Medical Science and Technology, IIT-Kharagpur, Kharagpur, West Bengal 721302, India. [3]Department of Life Sciences, Presidency University, Kolkata, West Bengal 700156, India. [4]Department of Molecular Medicine, Bose Institute, Kolkata, West Bengal 700054, India. [5]Infectious Diseases and Immunology, CSIR-Indian Institute of Chemical Biology, 4, Raja S. C. Mallick Road, Kolkata, West Bengal 700032, India. [6]Department of Physics, Jadavpur University, 188, Raja S. C. Mallick Road, Kolkata, West Bengal 700032, India. [7]Present address: Department of Molecular Genetics, University of Texas Southwestern Medical Center, 5323 Harry Hines Blvd, Dallas, TX 75390, USA. [8]Present address: INSA Senior Scientist, Indian Association for the Cultivation of Science, Kolkata, West Bengal 700032, India. ✉E-mail: bmukherjee@smst.iitkgp.ac.in; krish@iicb.res.in

SARS-CoV-2, the mechanism of CHOL transport and the modifications in the host membrane during the initial process of host cell invasion remain an unresolved problem.

An optimized physiological concentration of host membrane CHOL has been found to be a requirement for the successful host infection by *Leishmania* parasites (Pucadyil et al, 2004). Alternatively, it has also been observed that LD infection leads to a significant reduction in CHOL levels at the host plasma membrane, which occurs through an undefined mechanism, thereby affecting lipid raft-dependent processes (Chakraborty et al, 2005). Through pulse-chase experiments with radio-labeled cholesteryl esterified fatty acids, Semini et al have demonstrated the presence of host-derived CHOL coating on the parasite surface during the early amastigote stage, which corresponds to the post-invasion stage of *Leishmania* parasites (Semini et al, 2017). Since *Leishmania* parasites lack de novo CHOL synthesis (Roberts et al, 2003), and leishmania promastigotes only contain ERG as a sterol component, the detected CHOL in the parasites must be obtained either from the host cell or from the environment. It is also worth noting that lesion-derived amastigotes exhibit an increased content of CHOL relative to the total amount of sterols (Tetley et al, 1986) with respect to promastigotes (Berman et al, 1986; Ginger et al, 2001). These findings clearly indicate that the process of CHOL transfer might be important in *Leishmania* infection, and there might be specific *Leishmania* proteins, which would be responsible for executing the transfer of CHOL from the host membrane to parasite during invasion.

One of the major structural components of the surface membrane of *Leishmania* parasites is the kinetoplastid membrane protein-11 (KMP-11). KMP-11 has been implicated in regulating the overall lipid bilayer morphology of the parasite membrane (Jardim et al, 1995). In our initial study, we made two important observations regarding KMP-11. First, through sequence analysis, we identified CHOL interaction motifs within the KMP-11 sequence. These motifs consist of two CRAC-like domains (**C**holesterol **R**ecognition/interaction **A**mino acid **C**onsensus sequence) (Azzaz et al, 2022; Wang et al, 2022) and one CARC domain (reverse order of CRAC motif), which are highly conserved across all *Leishmania* species (Fig. 1A). Second, we noticed structural similarity between KMP-11 and lipid transfer proteins (LTPs). By utilizing the Raptor X tool (Xu et al, 2021) for structural alignment, we found notable overlaps between KMP-11, the CHOL transport START domain (Tsujishita and Hurley, 2000) and CHOL sensing GRAM domain (Doerks et al, 2000) of LTPs (Fig. 1B,i,ii,iii). Based on these analyses, we hypothesize that KMP-11 may possess a lipid transfer function, potentially enabling the *Leishmania* parasites to transport CHOL from the host membrane.

This study demonstrates that KMP-11 facilitates the initial step of LD infection. Knock-out LD parasites (KO) lacking KMP-11 exhibit altered morphology with compromised efficiency in host cell attachment and invasion. Our findings reveal that KMP-11 is involved in a novel CHOL transfer mechanism between the host and the parasite, which takes place through the potential formation of a KMP-11 induced bridge between LD and MΦ. Using tryptophan scanning mutagenesis, we identify a critical sequence stretch at the amino-terminal region of KMP-11 (amino acids 1–19), which may be responsible for a crucial phase transition at the host membrane. To further investigate, we synthesize different peptides based on the amino acid sequence at the amino terminal

(1–19) and demonstrate that this region generates a Hydrophobic Moment, which plays a critical role in the phase transition. Furthermore, we developed a mathematical model that incorporates the Hydrophobic Moment and the number of amino acid residues (N) forming the mirror sequence of the interacting domain at the amino terminal. This model helps elucidating various membrane modulating events and the infection process. To our knowledge, this is the first study which comprehensively characterized the pivotal role of KMP-11 to initiate host cell invasion process of *Leishmania* by mediating CHOL transport. Experiments are being carried out in our group to develop potential small molecules, which can target KMP-11 mediated phase transition as a potential therapeutic intervention against leishmaniasis.

## Results

### KMP-11 modulates MΦ membrane fluidity and facilitates host cell infection

Although LD promastigote surface is covered with glycoconjugates, previous report suggests that these glycoconjugates can be reoriented as promastigotes invade the host cell, which increases probability of a small protein like KMP-11 to get exposed and interact with host membrane (Forestier et al, 2015). In addition, as reported earlier the close proximity between LD and MΦ membrane (~10 nm) and tight binding during initial interactions makes it feasible for KMP-11 to create contact with MΦ membrane (Horta et al, 2020; Hsiao et al, 2011). Using a combination of biochemical assay and Confocal microscopy we observed a significant presence of KMP-11 at LD surface, which should be available during the initial attachment with host MΦ plasma membrane (Appendix Fig. S1A–C). Since, bioinformatics analyses (Fig. 1A,B) suggest KMP-11's potential role in CHOL transfer process, and LD infection has been convincingly linked with a fluid host membrane (Chakraborty et al, 2005; Majumder et al, 2012), we first checked if purified recombinant KMP-11 (r-KMP-11) and in vitro LD infection would affect host membrane fluidity (MF). r-KMP-11 was purified using previously published procedure (Sharma et al, 2013) (Appendix Fig. S1Di,ii). We measured membrane fluidity using 1,6-diphenyl hexatriene (DPH) fluorescence anisotropy (FA) and Laurdan (commonly used membrane-sensitive probes) generalized polarization (GP) measurements. Using host MΦ cells in vitro, we found that LD infection induces a large increase in MF, and the treatment with 1.5 mM CHOL partially offsets the effect of LD infection (Fig. 1Ci,ii), without having any significant effect on cell viability (Appendix Fig. S2A). Interestingly, exogenous treatment with r-KMP-11 also led to an increase in MF (for FA, Fig. 1Ci, inset) in a dose-dependent manner. The reduction in FA of MΦ membranes in the presence of 50 µM r-KMP-11 (concentration at which the change of FA plateaued) was found to be comparable to those obtained with LD infections (Fig. 1C). Previously, we found that the membrane binding activity of r-KMP-11 gets saturated in the µM concentration range, with r-KMP-11 to lipid ratio (P/L) of 0.004. In addition, our calculation based on the P/L ratio and µM protein concentration shows a significant availability of KMP-11 in LD (see Appendix Methods) to enable a productive infection. Although *Leishmania* infection has been reported to inhibit host cell

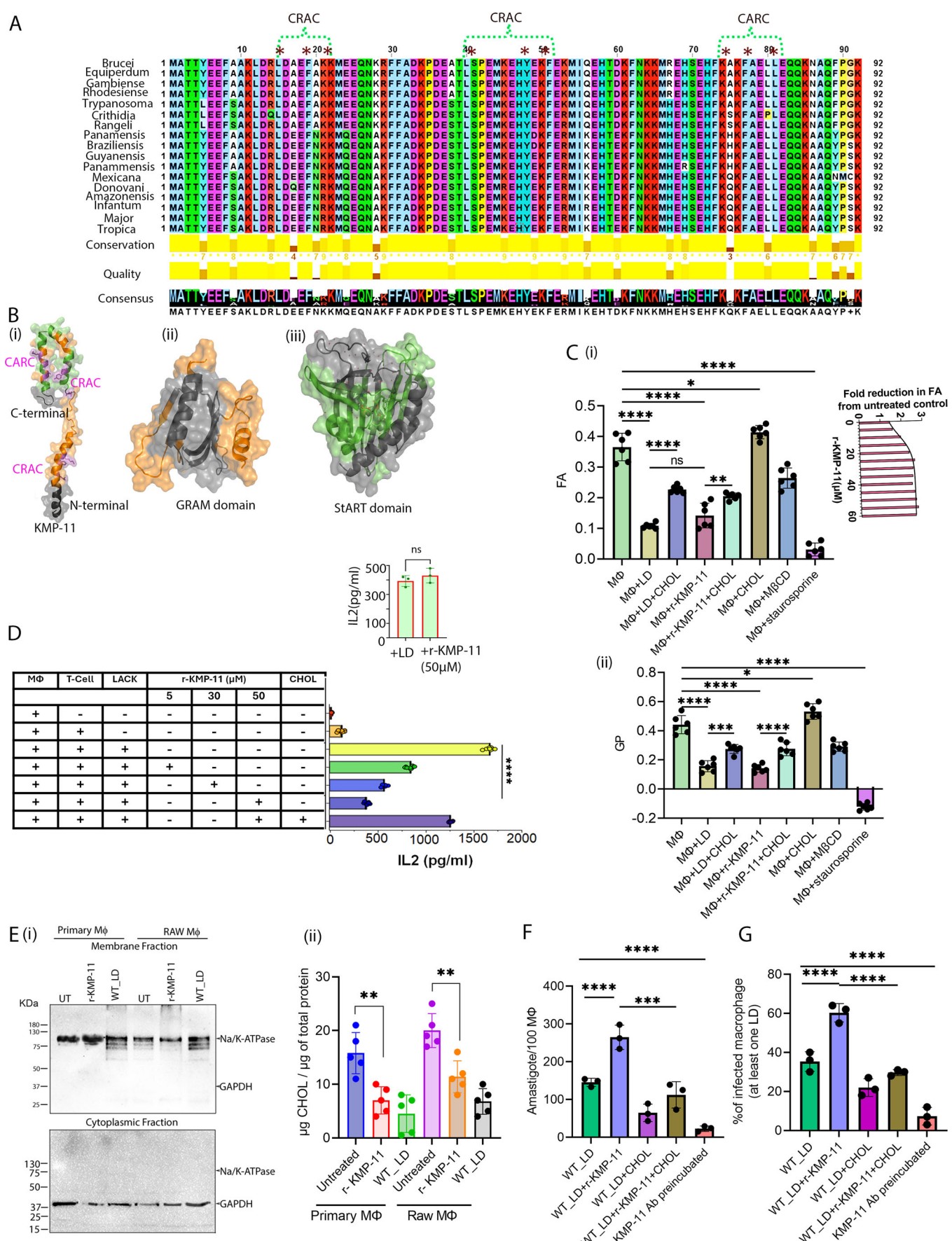

**Figure 1. Structural features of KMP-11 leading to cholesterol quenching and promoting fluidity of Macrophage (MΦ) membrane.**

(A) Multiple sequence alignment of KMP-11 protein sequences for different *Leishmania* spp. The CRAC and CARC domains (*marked) are found to be conserved. Sequence alignment is obtained from Clustal W analysis. (B) Structural similarity between (i) KMP-11 and (ii) CHOL sensing GRAM domain (orange) and (iii) CHOL transport StART domain (green). The structural similarity was evaluated using Raptor X structural alignment tool. (C) (i) The values of FA of MΦ membrane upon the treatment of LD, r-KMP-11, CHOL, mβCD and staurosporine. These FA values were determined using DPH fluorescence. Typical protein concentration was 50 μM and the liposomal CHOL concentration was 1.5 mM for each experiment (in liposomal CHOL where Lipid/CHOL ratio was 1:2). Each of mβCD and staurosporine was kept 1 mM as treatment concentration. The results were expressed as the mean ± SD derived from multiple data points. Statistical significance in the observed differences was determined through analysis of variance (ANOVA) utilizing GraphPad prism software (version 9.0).The number of independent experiments for each group $n = 6$. *P value = 0.0151, **P value = 0.0017, ****P value = 0.00008, $^{ns}$P value = 0.0760.Inset shows the fold reduction in FA under different dose treatment of r-KMP-11 with respect to untreated MΦ. The solid line through the data points was obtained using a hyperbolic curve fitting. (ii) GP of Laurdan fluorescence of MΦ membranes upon the treatment with LD, r-KMP-11 and CHOL. Typical protein concentration used was 50 μM and the liposomal CHOL concentration was 1.5 mM for each experiment. The results were expressed as the mean ± SD derived from multiple data points. Statistical significance in the observed differences was determined through analysis of variance (ANOVA) utilizing GraphPad prism software (version 9.0). The number of independent experiments for each group $n = 6$. *P value = 0.0108, ***P value = 0.0003, ****P value = 0.000053. (D) IL2 production from MΦs under different treatment conditions. 5 μM concentration was used as the low dose of r-KMP-11 and 30 μM as the medium dose while 50 μM was used as the optimal dose. The MΦs were cultured with T cell hybridoma – LMR7.5 activated by LACK antigen. Inset compares the level of IL2 released from with treatment of 50 μM r-KMP-11 or LD infection for 12 h. The number of independent experiments for each group was $n = 7$. The results were expressed as the mean ± SD derived from multiple data points. Statistical significance in the observed differences was determined through ANOVA utilizing GraphPad prism software (version 9.0). ****P value = 0.000067. (E) (i) Western blot showing expression of Na/K-ATPase (plasma membrane marker) and GAPDH (cytoplasmic marker) in the purified membrane fraction (upper panel) and cytoplasmic fraction (lower panel) isolated from r-KMP-11 treated PEC or RAW macrophages for 4 h. While purified plasma membrane fraction showed prominent expression of Na/K-ATPase but no GAPDH, reverse is observed for cytoplasmic fraction. (ii) Plasma membrane fraction was isolated from the MΦs by ultracentrifugation. Total membrane cholesterol (free and esterified) was estimated using the Amplex-Red kit and expressed as μg of cholesterol/μg of total protein. While the membrane fraction of untreated macrophages measured about 15 μg of cholesterol, r-KMP-11 treatment reduced this amount roughly to 7 μg. Similar results were observed for RAW MΦs represented on the right side of the graph, which showed ~19 μg and 11 μg of total cholesterol in untreated and r-KMP-11 treated conditions, respectively.The results were expressed as the mean ± SD derived from multiple data points ($n = 5$) obtained from five independent experiments. Statistical significance in the observed differences was determined through an unpaired Student's $t$ test utilizing GraphPad Prism software (version 9.0). **P value = 0.0022, $^{ns}$P value = 0.9044. (F) The plot shows the number of intracellular amastigotes per 100 MΦs under different treatment conditions. Measurements were done 4 h post-infection after rigorous washing of extracellular LD promastigotes. The results were expressed as the mean ± SD derived from multiple data points ($n = 3$) obtained from three independent experiments. Statistical significance in the observed differences was determined through ANOVA utilizing GraphPad prism software (version 9.0). ***P value = 0.0001, ****P value = 0.000079. (G) The plot shows the percentage of infected MΦs with at least one intracellular LD under different treatment conditions. Measurements were done 4 h post-infection after thorough washing of extracellular LD promastigotes. The results were expressed as the mean ± SD derived from multiple data points ($n = 3$) obtained from three independent experiments. Statistical significance in the observed differences was determined through ANOVA utilizing GraphPad Prism software (version 9.0). ****P value = 0.000058. Source data are available online for this figure.

apoptosis (Moore and Matlashewski, 1994), to determine if the observed fluidity changes come from potential apoptotic processes we checked the percentage of apoptosis by performing Annexin V staining (Invitrogen) in r-KMP-11 (50 μM) treated MΦs for 12 h. MΦs treated with 50 μM r-KMP-11 for 12 h resulted in similar percent of apoptotic population (6.23%) like untreated (8.44%) or LD infected control (4.74%) (Appendix Fig. S2B) thus ruling out any possibility that KMP-11 is inducing apoptosis in treated or infected MΦs.

To understand the effect of membrane fluidity indirectly, we then measured antigen presentation ability by treating MΦ with r-KMP-11 (Fig. 1D). Using I-A$^d$ restricted anti-LACK (Leishmania-activated C-kinase antigen) T-cells hybridoma, IL2 production was monitored in the presence or absence of LACK antigen. We observed a dose-dependent decrease in IL2 production in response to r-KMP-11 treatment. A maximum effect of r-KMP-11 on antigen presentation was observed at 50 μM concentration with a reduction in IL2 expression comparable to that of LD infected MΦ (Fig. 1D, inset). As a result, 50 μM of r-KMP-11 treatment were used for all subsequent experiments unless otherwise mentioned. Interestingly, the presence of r-KMP-11 does not seem to have any effect on IL10 and IL12 production (Appendix Fig. S2C). Furthermore, similar to LD infection, the presence of r-KMP-11, significantly decreases Cholera Toxin-B (CTX-B) binding to the cell surface (as evident from the reduction in raft cluster populations) which was restored upon liposomal-CHOL treatment (Appendix Fig. S2D,E). Subsequently, we performed direct CHOL measurement experiment by treating RAW 264.7 MΦs and primary peritoneal MΦs isolated from BALB/c mice with r-KMP-11

(50 μM) for 4 h. Plasma membrane fractions isolated from r-KMP-11 treated MΦs were used to quantify the total CHOL content using Amplex-Red Kit (Invitrogen) as compared to untreated control (Fig. 1Ei,ii). We found a significant drop in membrane CHOL content in the presence of r-KMP-11 which was comparable to LD-infected MΦs plasma membrane (Fig. 1Ei,ii).

Finally, we infected MΦs using stationary phase LD parasites with or without pre-incubation of KMP-11 antibody, and in the presence and absence of r-KMP-11. We observed a significant increase in both the percentage of infected MΦs (at least one LD/MΦ) and the number of intracellular amastigote counts after 4 h of co-incubation with 50 μM of r-KMP-11 as compared to only LD infected control (Fig. 1F,G). In contrast, co-incubation with liposomal CHOL or pre-incubating LD with anti-KMP-11 antibody reduced the infectivity in terms of amastigote load in MΦ (Fig. 1F,G). We reasoned that the liposomal CHOL treatment increases the rigidity of MΦ membrane and thus reduces the LD infectivity. Thus, CHOL quenching event by KMP-11 is directly correlated with the initial infectivity (and invasion) by LD.

## KMP-11 knock-out LD fails to attach and invade the host MΦ

To confirm the role of KMP-11 on LD infection, we generated a KMP-11 knock-out LD line (LD_KMP-11_KO) using CRISPR based strategy, which allows single cassette transfection for replacing both the alleles as reported previously by multiple groups (Beneke et al, 2017; Zhang and Matlashewski, 2015). The presence of bleomycin cassette in the genomic locus of *kmp-11*was confirmed

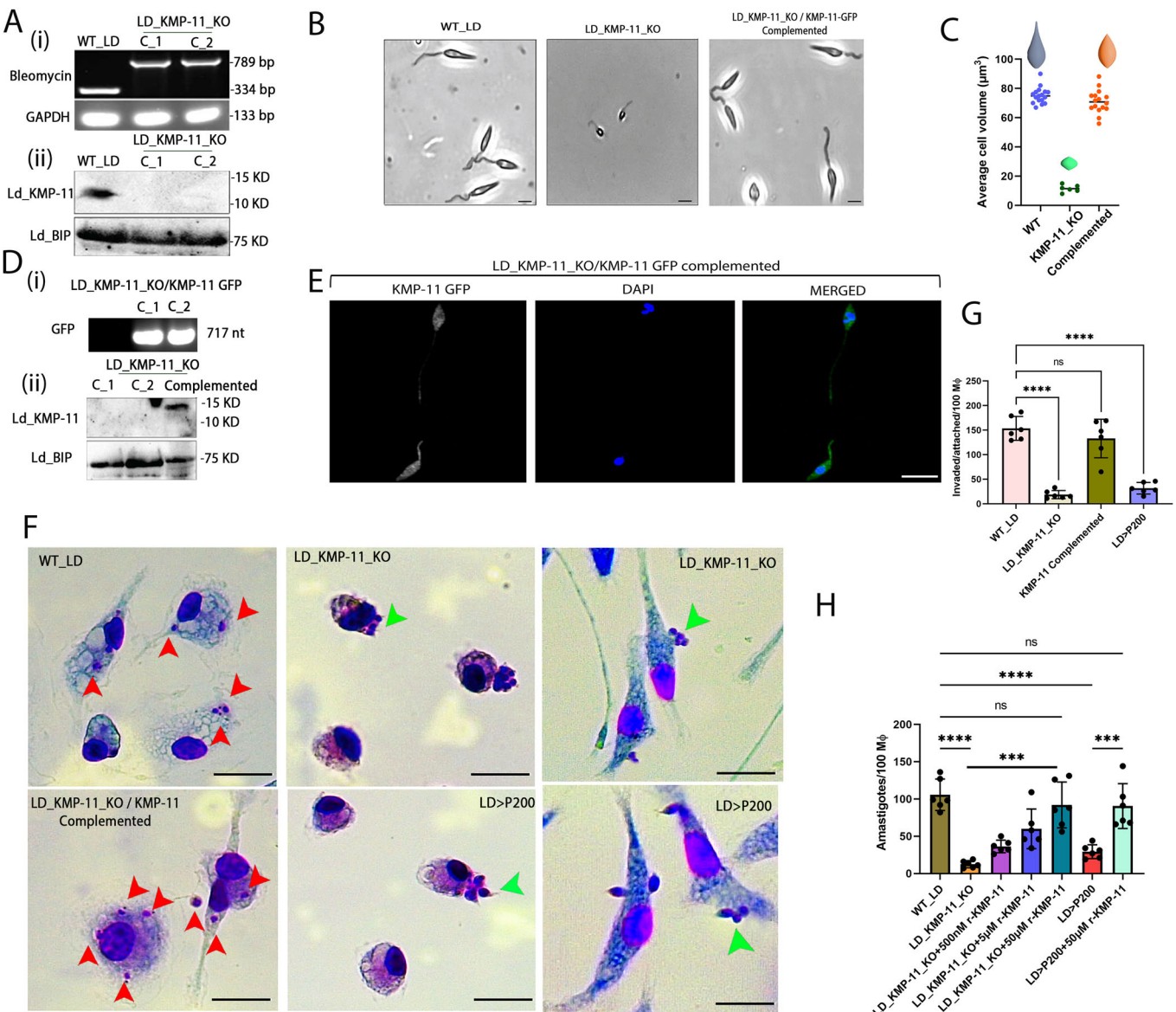

through PCR (Fig. 2A,i). While two independent clones (C_1 and C_2) of LD_KMP-11_KO showed integration of bleomycin cassettes, it was absent in wild-type parental strain (WT_LD) (Fig. 2A,i). Next, the expression of KMP-11 was compared between WT_LD and LD_KMP-11_KO clones. Our result showed no detectable expression of KMP-11 in either of the LD_KMP-11_KO clones although Ld_BIP was expressed constitutively similar to WT_LD (Fig. 2A,ii). Interestingly, using light microscopy we found that LD_KMP-11_KO parasites are significantly smaller in size when compared to WT_LD, with severely reduced flagella which are also evident from the reduction in volume of LD_KMP-11_KO parasites (Fig. 2B,C). Complementation with wild type copy of *kmp-11* in LD_KMP-11_KO lines was performed by ectopic expression of KMP-11-GFP as confirmed by the expression of GFP (717nt, Fig. 2D,i) in LD_KMP-11_KO /KMP-11 complemented LD lines by PCR and by fluorescence microscopy (Fig. 2D,E). Complementation was further confirmed by the reappearance of

KMP-11 in complemented LD lines by Western blot (Fig. 2Dii) with recovery of wild type parental morphology in bright field (Fig. 2B, right most panel). 3D reconstitution analyses confirmed a significant volumetric reduction in LD_KMP-11_KO (volume ~10.99 μm³) in comparison to parental WT_LD (volume ~72.98 μm³), whereas a gain of volume was observed for LD_KMP-11_KO /KMP-11 complemented with KMP-11-GFP (volume ~68 μm³) (Fig. 2C). LD_KMP-11_KO promastigotes showed a slower growth kinetics when compared to the parental line, although consistent increase in the number of LD_KMP-11_KO promastigotes were observed from day 1 to day 6, which was similar to WT_LD and LD_ KMP-11_KO /KMP-11 complemented lines (Fig. EV1A). Parental, LD_KMP-11_KO and LD_ KMP-11_KO /KMP-11 complemented lines reflected ~90% of viable parasites suggesting knocking out KMP-11 did not have any significant effect on the viability (Fig. EV1B). We then wanted to check whether LD_KMP-11_KO lines would exhibit any defect to

**Figure 2. KMP-11 knock out in LD hampers its attachment and invasion of host MΦ.**

(A) (i) Integration PCR of Bleomycin cassette in genomic locus of KMP-11 using genomic DNA of wild type (WT_LD), and KMP-11 knock out LD lines (LD_KMP11_KO). (ii) KMP-11 expressions in WT_LD, LD_KMP11_KO lines (C_1 and C_2) using anti-KMP-11 antibody. Ld_BIP is used as endogenous control. (B) Light microscopy representing morphology of WT_LD, LD_KMP11_KO, and LD_KMP-11_KO/KMP-11 Complemented lines. The scale bar in each image is 2 μm. (C) Volumetric calculation of the WT_LD ($n = 17$), LD_KMP11_KO ($n = 6$) and LD_KMP-11_KO/KMP-11 ($n = 16$) Complemented lines show a significant volume reduction in LD_KMP11_KO which is recovered upon complementing with WT copy of KMP-11. This figure also shows the 3D representations of the 3D-reconstituted parasites using blender computational tool. '$n$' stands for the number of parasites accounted for the volumetric calculations. Data are represented as mean ± SD. Bright-field images of the parasites of (B) were processed in blender tool (a tool generally used for 3D designing and animation). Using this tool, we selectively draw a Bezier curve on the single parasite which exhibits the similar shape pointing to an axial symmetry. Next, by rotating the Bezier spline we extruded a 3d reconstruction of an individual parasite. Then, we used a volume plug-in to evaluate the volume of individual parasites. We then repeated the same steps for multiple parasites to obtain an average volume of the parasites for each case. (D) (i) PCR showing expression of EGFP in LD_KMP-11_KO lines complemented with wild-type copy of KMP-11 fused with EGFP (KMP-11-GFP). (ii) Western blot showing recovery of KMP-11 expressions in LD_KMP-11_KO/KMP-11 Complemented LD lines. Ld_BIP is used as endogenous control. (E) GFP-positive expression of KMP-11-GFP in LD_KMP-11_KO lines by Fluorescence microscopy. Scale bar is 5 μm. (F) Representative Giemsa-stained LD infected MΦ at 4 h post-infection showing intracellular amastigotes (red arrows) in case of infection with WT_LD and LD_KMP-11_KO/KMP-11 Complemented lines, while similar infection with LD_KMP11_KO and LD promastigotes cultured in M199 for greater than 200 passages (LD > P200) showed loosely attached parasites (green arrow)without any internalized amastigotes.The right most panel shows stretched MΦs bearing loosely attached parasites (KMP-11_KO and LD > P200) marked in green arrows. Scale bar in each image is 15 μm. (G) Number of LD invaded/attached per 100 MΦs when MΦs were infected WT_LD, LD_KMP11_KO, LD_KMP-11_KO/KMP-11 Complemented, and LD > P200. LD interaction with MΦ is carried out for 4 h. The data were expressed as the mean ± SD derived from independent experiments $n = 6$. Statistical significance in the observed differences was determined through ANOVA utilizing GraphPad prism software (version 9.0). ****$P$ value = 0.000047, $^{ns}P$ value = 0.0813. (H) LD amastigote load per 100 MΦs for WT_LD, LD_KMP11_KO, LD > P200, and LD_KMP-11_KO, LD > P200 supplemented with different doses of recombinant KMP-11 (r-KMP-11, 500 nM, 5 μM and 50 μM). Amastigote count was measured after 4 h post-infection. The data were expressed as the mean ± SD derived from the independent experiments $n = 6$. Statistical significance in the observed differences was determined through ANOVA utilizing GraphPad prism software (version 9.0). ***$P$ value = 0.0003 (LD_KMP-11_KO vs LD_KMP-11_KO + 50 μM r-KMP-11), ***$P$ value = 0.0002(LD > P200 vs LD > P200 + 50 μM r-KMP-11), ****$P$ value = 0.000086, $^{ns}P$ value = 0.9201(WT_LD vs LD > P200 + 50μM r-KMP-11) and $^{ns}P$ value = 0.8758 (WT_LD vs LD_KMP-11_KO + 50 μM r-KMP-11). Source data are available online for this figure.

invade host MΦs. Since infectivity of LD parasites is linked with percentage of metacyclics (Lira et al, 1998), we compared percentage metacyclics between WT_LD, LD_KMP-11_KO and LD_ KMP-11_KO /KMP-11 complemented lines using Flow cytometry (Saraiva et al, 2005). Comparable number of metacyclics was observed in the case of LD_KMP-11_KO, which was similar to parental and KMP-11 complemented lines (Fig. EV1C,D). However, it should be mentioned that as LD_KMP-11_KO lines exhibit a dramatic reduction in their size and morphology, the light scattering based determination might not provide an accurate measurement of percent metacyclics for LD_KMP-11_KO promastigotes.

Next, host cell infection assays were performed by sorting an equal number of metacyclics for WT_LD, LD_KMP-11_KO and LD_KMP-11_KO/KMP-11 complemented lines using CytoFLEX SRT (Beckman Coulter). Compared to WT_LD and LD_KMP-11_KO/KMP-11 complemented, LD_KMP-11_KO lines showed a significant impairment of host cell attachment and invasion (Fig. 2F,G). Using Giemsa staining of infected MΦ, we found that while infection with WT_LD and LD_KMP-11_KO /KMP-11 complemented resulted in intracellular amastigotes, infection with LD_KMP-11_KO parasites produced very few MΦs with attached promastigotes and significantly lower number of intracellular amastigotes (Fig. 2F,G). This defect in host cell infection in case of LD_KMP-11_KO line, seems comparable with initial infectivity of LD lines which were repeatedly passed through in vitro culture for more than 200 passages (LD > P200) without transforming through animal (Fig. 2F,G,H). LD > P200 also exhibited a significant decrease in KMP-11 expression as compared to WT_LD lines which was also reported earlier (Mukhopadhyay et al, 1998) (Fig. EV1E). Pre-incubation of LD > P200 with r-KMP-11 resulted in increased infectivity (Fig. 2H), again suggesting KMP-11 expression in the LD surface is directly linked to its ability to set up a productive infection in the host cell. Additionally, we measured both of the fluidity and CHOL in plasma membrane of

MΦ after treatment of WT_LD, LD_KMP-11_KO, LD_KMP-11_KO/KMP-11 complemented, LD > P200, LD_KMP-11_KO (supplemented with r-KMP-11) and LD > P200 (supplemented with r-KMP-11) (Fig. EV1F,G). These results clearly showed that KMP-11 is responsible for CHOL quenching from MΦ plasma membrane and the alteration in fluidity. Thus, our study indicates that KMP-11 driven fluidity change and CHOL quenching event are correlated with attachment and infection process by LD in MΦ.

## KMP-11 binds differentially to the gel (So) and liquid disordered (Ld) states

In the previous sections, we have established that r-KMP-11 facilitates LD infection and increases the fluidity in MΦ membrane. Subsequently, we wanted to understand a molecular and biophysical picture of the effect of KMP-11 in host cell invasion. For many of these experiments, we needed to use synthetic model membranes as simplified mimics of biological membrane. Biological membranes and their different states has been discussed extensively in the literature (Kaiser et al, 2009).

MΦ membrane predominantly contains liquid-ordered lipid domains (Lo) and DPPC:DOPC: CHOL (1:1:1) is extensively used as a raft model for the cell membranes (Ahmadi et al, 2022; Veatch and Keller, 2003). Also, neutral lipid (PC, mainly 16:0) is found predominantly in higher fractions in the outer leaflet of MΦ plasma membrane (Montenegro-Burke et al, 2016) and hence, we used DPPC as the lipid component of our model membrane. It may be noted that the plasma membrane contains ~30% CHOL (Subczynski et al, 2017), and hence we used DPPC containing 30% CHOL (DPPC-CHOL) as a simplified mimic of the host MΦ membrane. In contrast, the parasite LD membrane contains ERG natively (McCall et al, 2015), and hence, we used DPPC vesicles containing 30% ERG (DPPC-ERG) as a simple mimic for LD membrane. Like MΦ membranes, both DPPC-CHOL and DPPC-ERG remain predominantly in the liquid-ordered (Lo) phase

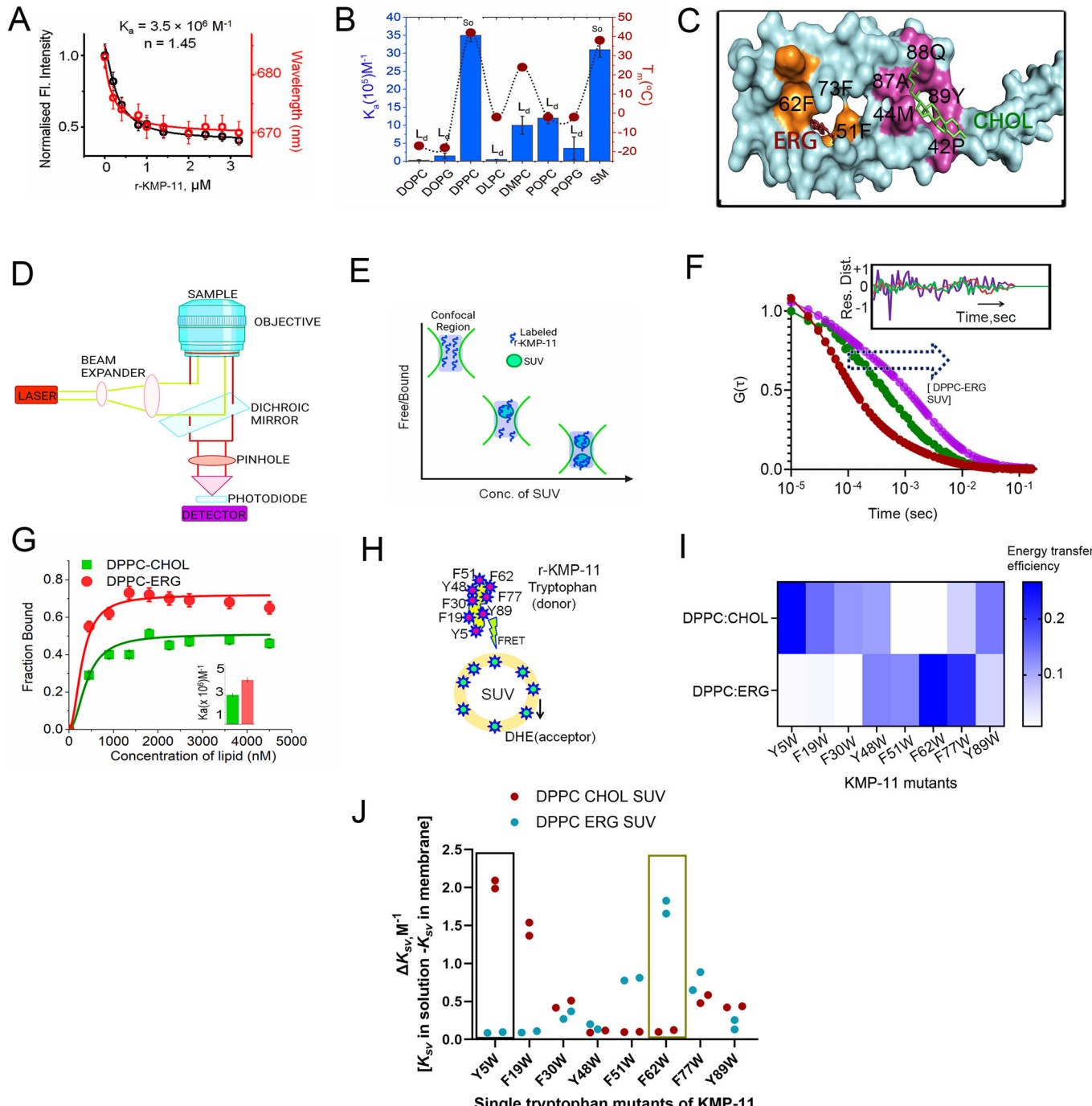

(de Almeida and Joly, 2014). DPPC in the absence of CHOL and ERG, in contrast, corresponds to gel or solid ordered (So) phase (de Almeida and Joly, 2014). Also, above the transition temperature or in the presence of KMP-11 beyond a particular protein-to-lipid ratio (as shown later in this paper), the lipid molecules in DPPC would be present at their liquid-disordered (Ld) phases.

Employing a recently demonstrated membrane-specific fluorophore DiD quenching assay (Sannigrahi et al, 2017) (Figs. 3A and

EV2A), we found that r-KMP-11 has stronger binding towards saturated phospholipid, like DPPC ($K_a = 3.5 \times 10^6$ M$^{-1}$) (Appendix Tables S1 and S2). Interestingly, the affinity of r-KMP-11 was found higher with lipids at their gel (So) phase [DPPC, SM etc], when compared to the lipids at their Ld phase at 25 °C [POPC, POPG, DOPC, DOPG, DLPC, and POPG etc.] (Fig. 3B). The reduced binding of r-KMP-11 with DPPC at a higher temperature (44 °C, that corresponds to the Ld phase) in comparison to DPPC

**Figure 3. Binding of KMP-11 with membrane depends significantly on the sterol components.**

(A) The binding of r-KMP-11 with DPPC model membranes was monitored by the decrease in DPPC SUV bound DiD fluorescence intensity (left y axis) as well as the blue shifts of the fluorophore (right y axis) with increasing concentration of the added proteins. This assay was discussed in detail in our previous publications (Sannigrahi et al, 2017). The fluorophore DiD does not have significant fluorescence in the aqueous buffer when excited at 600 nm, while the addition of phospholipid vesicles significantly enhances its fluorescence intensity. The high environmental sensitivity of this fluorophore makes it an excellent candidate to monitor protein–lipid binding. There may be several processes that can occur in the excited state of the fluorophore leading to a decrease in the intensity (quenching) due to protein binding. We find that the average lifetime of the fluorophore decreases from 4.2 nsec (in the absence of KMP-11) to 3.7 nsec as we add KMP-11, suggesting a strong contribution of dynamic quenching, which presumably occurs through the solvent environments (Appendix Table S1). The data were expressed as the mean ± SD derived from independent experiments $n = 3$. (B) This plot shows the binding of KMP-11 with different lipid variants possessing different phase transition temperatures. The binding interactions of KMP-11 with different lipids were studied at room temperature. Here, So and Ld stand for solid ordered and liquid disordered phase, respectively. The data were expressed as the mean ± SD derived from independent experiments $n = 3$. (C) The optimum docked structure of KMP-11 with CHOL and ERG as obtained from molecular docking study is shown. (D) A schematic of a typical FCS setup. (E) A schematic representation of the membrane binding experiments using FCS, which shows that with increasing concentration of SUVs, the Alexa-488 maleimide labeled monomeric protein populations (the fast component) decrease with a concomitant increase in the membrane-bound labeled protein (the slow component). (F) Correlation functions of Alexa-488-maleimide labeled KMP-11 S38C with increasing concentration of DPPC-ERG SUVs. We observed right shift in the correlation functions with increasing SUV concentrations indicating the binding events of the proteins with SUVs. The inset shows the residual distribution of the fit. The randomness indicates the goodness of the fit. (G) The fraction of the bound proteins is plotted against the concentrations of DPPC-CHOL SUVs and DPPC-ERG SUVs to evaluate the binding constants of Alexa-488 labeled KMP-11 towards membranes. The inset shows the binding constants with DPPC-CHOL and DPPC-ERG suggesting that KMP-11 has stronger binding towards DPPC-ERG. The data were expressed as the mean ± SD derived from independent experiments $n = 3$. (H) A schematic diagram shows the FRET pairs to measure the region-specific binding of KMP-11 with liposomes. Here, single tryptophan residues of the single mutants (Y5W, F19W, F30W, Y48W, F51W, F62W,F77W&Y89W)of KMP-11 are used as donor and membrane-bound DHE is used as the acceptor molecule in order to understand the binding regions of KMP-11 with ERG and CHOL. (I) A heatmap shows the FRET efficiencies of eight single tryptophan mutants. FRET efficiencies have been calculated using DPPC-CHOL and DPPC-ERG to show the orientation of KMP-11 during initial attachment. We found that the energy transfer efficiency between tryptophan and DHE depends strongly on sterols contents. The color bar stands for the energy transfer efficiency. (J) Plot shows the relative changes in Stern–Volmer constants ($\Delta K_{sv} = K_{solution} - K_{membrane}$) of eight single tryptophan mutants of KMP-11 in presence of DPPC-CHOL and DPPC-ERG SUVs. Box marked regions indicate the highest changes in $\Delta K_{sv}$ in DPPC-CHOL and DPPC-ERG membrane environments. The number of independent experiments for each group was $n = 2$. Source data are available online for this figure.

below its phase transition temperature (DPPC So/Ld transition temperature is 42 °C) validates the binding specificity of r-KMP-11 towards gel phase lipid components (Fig. EV2B).

## Sterol components regulate the binding and structural alignment of KMP-11 towards LD and MΦ membranes and oligomerization of KMP-11

Since sterols have been reported to exhibit profound impact on the binding and activity of diverse lipid binding proteins (Endapally et al, 2019) and LD and MΦ membranes are comprised of two different sterol components (ERG and CHOL, respectively), we surmised the existence of precise effects of CHOL (MΦ sterol) and ERG (LD sterol) on KMP-11-lipid interactions. Therefore, in order to determine the binding regions and binding energy of KMP-11 with DPPC, ERG, and CHOL, we employed computational docking using Gemdock (Yang and Chen, 2004) (Figs. 3C and EV2C,D,E). The in silico analysis revealed that the binding affinity of KMP-11 with ERG was significantly higher than that with DPPC and CHOL (Fig. EV2D,E). This in silico analysis also predicted that the binding site for ERG would be different from that of CHOL (Figs. 3C and EV2E).

To experimentally validate the above predictions, we used fluorescence correlation spectroscopy (FCS) to probe how the binding of KMP-11 would be influenced by the presence of ERG (DPPC-ERG, mimicking LD membrane) and CHOL (DPPC-CHOL, mimicking MΦ) (Fig. 3D). For these experiments, we prepared a single cysteine mutant (S38C) of r-KMP-11, which was labeled with a fluorescent dye called Alexa488-maleimide. A schematic diagram of the behavior of the labeled proteins and protein–lipid complexes (Sannigrahi et al, 2021) inside the confocal volume is shown in Fig. 3E. We performed FCS analysis by measuring the correlation functions using 50 nM of the Alexa488-labeled KMP-11 S38C variant (Alexa488-r-KMP-11) in the presence of increasing concentrations of DPPC-ERG and DPPC-CHOL SUVs (Fig. 3F). We employed a two-component diffusion

model to fit the correlation functions, where the fast and slow diffusing components corresponded to the free (with hydrodynamic radius, $rH_1 = 10.5$ Å) and lipid-bound protein (hydrodynamic radius, $rH_2 = 240$ Å), respectively. With increasing DPPC SUV concentrations, we observed an increase in the percentage of the slow component (Fig. 3F,G), which occurred at the expense of the fast component. A sigmoidal fit of these two components provided the values of Ka (Fig. 3G, inset). Our data indicated that in compliance with the above in silico prediction, r-KMP-11 exhibited three-fold stronger binding to DPPC-ERG when compared to DPPC-CHOL (Fig. 3G, inset). Additionally, we observed a decrease in binding affinity with increasing CHOL concentrations in DPPC (Fig. EV2F). This is expected since the addition of CHOL changes DPPC from the So to Lo phase, and as previously shown, r-KMP-11 has a higher binding affinity towards the So phase.

To determine experimentally the lipid binding regions of KMP-11 we used FRET. Since wild type r-KMP-11 lacks tryptophan residues in its native sequence, we used tryptophan scanning mutagenesis technique to generate eight single tryptophan mutants (Y5W, F19W, F30W, Y48W, F51W, F62W, F77W and Y89W) for FRET experiments. These tryptophan residues, positioned at different locations within the KMP-11 sequence, acted as the donors in our FRET assay, while externally added dehydroergosterol (DHE), in the SUVs(DPPC-CHOL and DPPC-ERG) served as the acceptor (Brahma and Raghuraman, 2022) (Fig. 3H). The FRET data with F62W suggested a significant increase in energy transfer efficiency for DPPC-ERG compared to DPPC-CHOL (Fig. 3I), indicating that the F62W mutation region is the probable binding site for DPPC-ERG membrane. Figure 3I further suggested that the regions near the Y5W mutation may be close to the interaction site for DPPC-CHOL membrane since Y5W showed highest energy transfer efficiency with DPPC-CHOL membrane. We also complemented the FRET analysis with the measurement of the tryptophan exposures of those eight single tryptophan mutants in solution and membrane bound conditions. We performed

acrylamide quenching experiment on the tryptophan mutants in the absence and presence of DPPC-CHOL and DPPC-ERG SUVs separately. Subsequently, we evaluated the relative changes in stern-Volmer constants [$\Delta K_{sv}$ = $K_{sv}$ (in solution)- $K_{svm}$ (membrane bound)]. While we found the highest $\Delta K_{sv}$ value for Y5W mutant with DPPC-CHOL SUVs (Fig. 3J), we observed highest $\Delta K_{sv}$ value for F62W mutant with DPPC-ERG SUVs (Fig. 3J). This result further inferred that while the amino terminal domain of KMP-11 is the probable binding site for DPPC-CHOL membrane, with 62nd residue region the probable binding site for DPPC-ERG membranes. We note that the observations from the FRET and tryptophan quenching experiments aligned well with the docking prediction. Based on sequence analyses, docking, tryptophan quenching and FRET data, we developed a plausible schematic alignment of KMP-11 during the process of host attachment (Fig. EV2G), illustrating how KMP-11 can form a bridge between LD and MΦ.

To provide further support for the bridging between LD and MΦ and the involvement of KMP-11, using the membrane mimics we developed a single vesicle imaging method using TIRF microscopy within a microfluidic chamber (Sako et al, 2000). For this measurement, we created a PEG cushioned supported bilayer platform inside the chamber. The supported bilayer is comprised of DPPC-CHOL: PEG5000 PE (0.5%). We then introduced Alexa488-labeled-r-KMP-11 and subsequently added DiD-labeled DPPC-ERG SUVs onto the supported lipid bilayer (SLB) channel (Fig. 4A). After 5-minute incubation, when we gently washed the channel to remove unbound DPPC-ERG SUVs, we observed a considerable number of bound SUVs on the PEG cushioned SLB surface (Figs. 4Bi and EV3A; Movie EV1). In contrast, when we performed the same experiment using DiD-labeled DPPC-CHOL SUVs (and not DPPC-ERG SUVs), we did not observe any significant population of docked vesicles on the surface of the PEG-SLB (Fig. 4Bii). Furthermore, no bound population was observed when we did not add r-KMP-11 (Fig. EV3B). By employing co-localization, we found that r-KMP-11 molecules were present between the SLB and DPPC-ERG SUVs (Figs. 4Bi and EV3C–E). We then estimated the mean diffusion coefficients of the DiD-labeled DPPC-ERG vesicles bound to the PEG SLB through the r-KMP-11 bridge by analyzing all particle trajectories (Fig. 4C). We found that most of the particles exhibited mobilities <2 μm²/s, which was comparable to the reported range for peripheral membrane proteins (0.8–2 μm²/s) and transmembrane proteins (0.02–0.2 μm²/s) (Sathyanarayana et al, 2018) (Fig. 4C,D). We inferred from these measurements that KMP-11 can strongly bridge two membranes containing different sterol molecules. This inter-membrane bridging property of r-KMP-11 was also supported by dynamic light scattering data (Fig. EV3Fi,ii).

We then investigated the attachment of WT_LD and LD_KMP-11_KO and LD_KMP-11_KO/KMP-11 complemented on DPPC-CHOL SLB, which showed remarkable adhesion of WT_LD and LD_KMP-11_KO/KMP-11 complemented on SLB but not for LD_KMP-11_KO (Fig. EV3Gi,ii,iii).

While in the previous paragraph we provided extensive supports for the bridging between LD and MΦ membrane and the role of KMP-11, those experiments were carried out using membrane mimics. A set of FRET measurements was carried out using LD infection in MΦ to further support bridge formation between KMP-11 on LD membrane with host MΦ membrane. For these FRET studies, we used GFP labeled KMP-11 complemented LD and Rhodamine DHPE labeled MΦ (Appendix Fig. S3A–C). It may be noted that for these FRET measurements, GFP was used as the fluorescence donor and Rhodamine as the acceptor. We found significant FRET between GFP and Rhodamine indicating an apparent donor-acceptor distance ranging between 20 and 60 Å which further suggests that KMP11 and macrophage membrane can interact strongly (Appendix Fig. S3A,C). In contrast we did not see significant FRET when we pretreated KMP-11 GFP LD with anti KMP-11 antibody (Appendix Fig. S3B,C).

Next we wanted to understand the nature of the protein molecules, which we believe makes the formation of the bridge possible. Employing computational analysis using Pasta 2.0 (Walsh et al, 2014), we found strong oligomerization propensity for KMP-11 (Appendix Fig. S4A,B). Initial experimental support of this prediction came from concentration-dependent FCS (Fig. 4E), which showed an increase in diffusion time (and a decrease in diffusion coefficient) with protein concentrations. Since protein self-association increases the effective size of the protein, this behavior is expected for a protein, which is oligomerizing in a concentration-dependent manner. This behavior was enhanced in the presence of salt (Fig. 4E), which is expected as salt increases the dielectric constant of the medium. To further characterize the oligomeric status of KMP-11, we then employed a single molecule photobleaching step analysis approach (Sathyanarayana et al, 2018). Since each r-KMP-11 monomer carries a single dye molecule, there would be a stepwise decrease in intensity as each dye would photo-bleach upon irradiation. By counting the number of these photobleaching steps, we determined the presence of r-KMP-11 tetramers (Fig. 4F,G). Complementary experiments using negative stain TEM and AFM both also showed the presence of oligomeric r-KMP-11 molecules (Fig. 4H,I; Appendix Fig. S4C,D). While TEM clearly showed the presence of tetrameric molecules (Fig. 4I, inset), there was heterogeneity in the particle structures and morphology in AFM. Furthermore, the photobleaching steps analysis in the inter-membrane bridged state of r-kMP-11 also suggested that r-KMP-11 forms oligomer during inter-membrane bridging condition (Fig. 4Ji,ii,iii). Since the above experiments clearly suggest that KMP-11 has a tendency to oligomerize and there is expected to be an exclusion volume effect with LD and MΦ membrane closing in, we speculate that KMP-11 may exist in an oligomeric state within the bridge complex.

## KMP-11 quenches CHOL from host through inter-membrane trafficking

To study the transfer of CHOL from the host membrane using the membrane mimic systems, we conducted a FRET-based assay using DHE-incorporated DPPC SUVs as a potential donor of CHOL (host membrane) and DAUDA-incorporated DPPC-ERG SUVs as an acceptor (membrane of the parasite) (Naito et al, 2019). The schematic representation of the CHOL transport assay is shown in Fig. 5A. Initially, DHE was loaded exclusively into the donor liposomes, and the transfer of DHE from the donor to the DAUDA-rich acceptor liposomes was monitored over time by measuring FRET between the transferred DHE and DAUDA in the acceptor liposomes (Fig. 5A). In the absence of r-KMP-11, no transport was observed, whereas the presence of r-KMP-11 led to a rapid increase in the FRET signal, indicating efficient extraction of

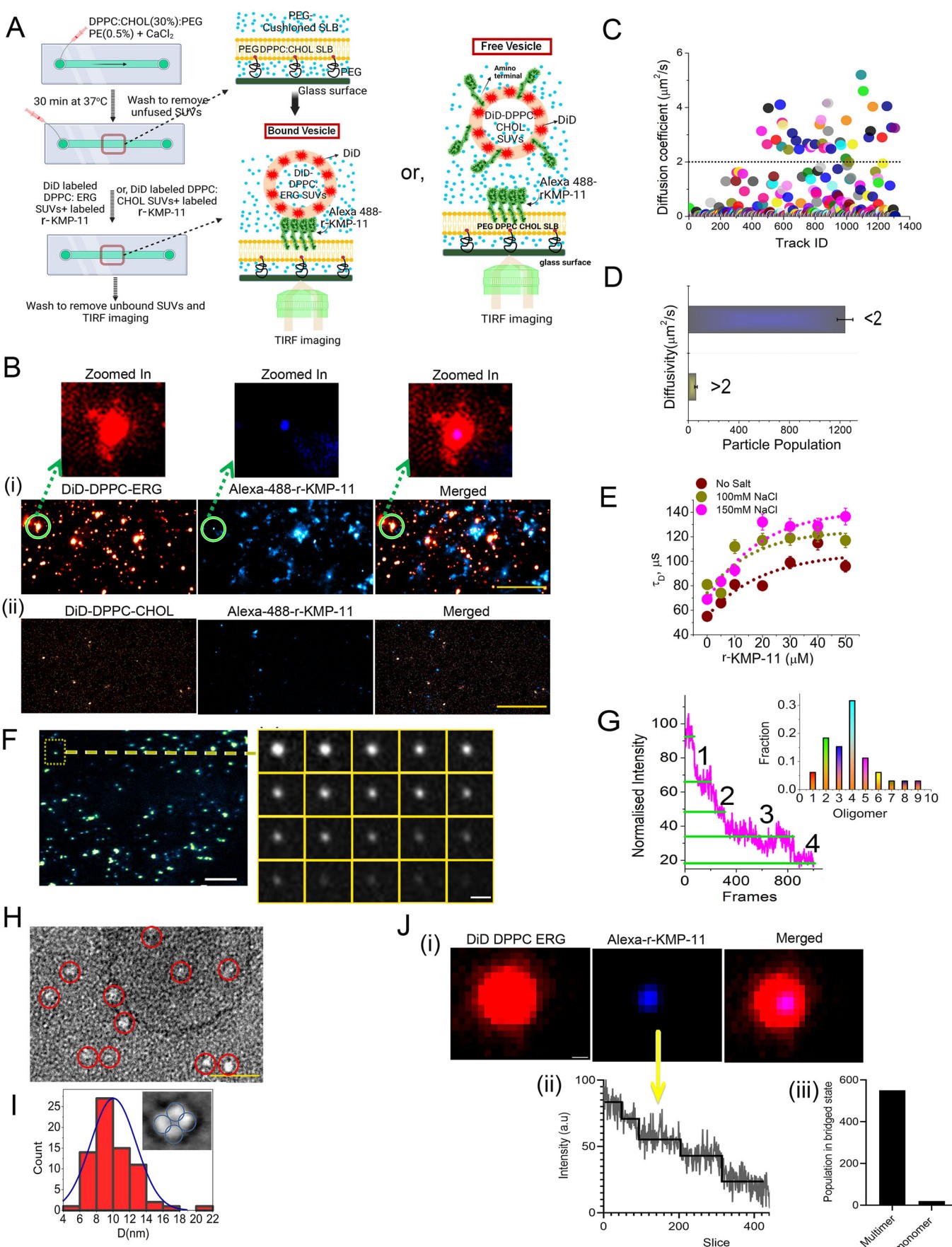

**Figure 4.　Inter-membrane bridging and oligomerization of KMP-11.**

(A) PEG 5000 PE was used to elevate the SLB from the glass surface, thereby providing sufficient space for hydration. Thereafter, DPPC-CHOL (30%): PEG PE 5000 (0.5%) SUVs were added with 30 mM CaCl$_2$ and spread inside the microchannel of a microfluidics chamber to make PEG cushioned SLB. Next, 100 nM Alexa488Maleimide labeled r-KMP-11 was added to the chamber and 500 nM DiD-labeled DPPC-CHOL (30%) SUVs and DPPC-ERG (30%) SUVs in separate channels were added subsequently. After gently washing the channels 10 times, TIRF microscopic imaging was done to quantify the SUV-r-KMP-11-SLB complex populations. (B) (i) TIRF microscopic images of the DiD-labeled DPPC-ERG (red), Alexa-labeled r-KMP-11 (blue) and the merged image. Zoomed in images of r-KMP-11(blue) bound DPPC-ERG SUVs (red) on SLB made of PEG PE:DPPC:CHOL. (ii) TIRF microscopic images of DiD-labeled DPPC-CHOL SUVs (red), Alexa-labeled r-KMP-11 (blue), and the merged image. Scale bar for (i) and (ii) is 10 μm. (C) Diffusion coefficients of DiD-labeled DPPC-ERG SUVs on SLB were plotted against the track ID of individual particles on SLB. Diffusion coefficients were evaluated by single particle tracking of the docked vesicles on SLB by r-KMP-11 employing track mate in ImageJ. (D) The plot of the population of DiD-labeled DPPC-ERG SUVs based on their diffusivity as obtained from single particle data analysis. The number of single SUV used for the data analysis was $n = 1300$. Here, vesicle population with diffusivity >2 μm$^2$/s is much less than those with <2 μm$^2$/s). The data were expressed as the mean ± SD derived from three independent experiments ($n = 3$) for both groups. (E) Understanding of the oligomerization status of KMP-11 by FCS: Plot of the diffusion times (τ$_D$) of r-KMP-11 (50 nM Alexa488Maleimide labeled r-KMP-11 in the presence of unlabeled r-KMP-11) against different total concentration of r-KMP-11. Self-association of a protein is expected to increase the diffusion time as a function of total protein concentrations (as observed here). τ$_D$ data were taken in the absence and presence of different salt concentrations, which clearly suggests that the addition of salt facilitates self-association. The data were represented as the mean ± SD derived from three independent experiments ($n = 3$ for each data point). (F) Representative TIRFM image (upper panel) of r-KMP-11 oligomers in PEG-SLB (PEGPE:DPPC:ERG i.e LD mimic). Scale bar is 5 μm. The labeled protein concentration used was 50 nM. The lower panel shows the time-lapse images of a single particle as selected by the yellow box in the upper panel. Each box represents the same region separated in time by 2.5 s. Scale bar is 200 nm. (G) Representative photobleaching event of a single particle shows four steps of photobleaching indicating probable tetramer formation on PEG-DPPC-ERG SLB. Inset shows the histogram of the fractional population of different multimers based on the number of constituent monomer units. Here, the x axis stands for the number of monomer units in the oligomers. (H) TEM micrograph of r-KMP-11 oligomeric structures in the presence of membranes, which shows the presence of oligomeric structures. The scale bar is 50 nm. (I) The histogram shows the size distribution of the r-KMP-11 oligomers. Zoom in the image of abundant oligomeric structures of r-KMP-11 (inset) with an average pore diameter of ~1.2 nm. It also shows possible tetrameric particle formation by r-KMP-11. (J) (i) Representative image of DiD DPPC-ERG vesicles (red) and Alexa-488 maleimide labeled r-KMP-11 (blue) and their merged form. This image shows the bridging of ERG-containing vesicle by KMP-11 on DPPC-CHOL SLB. Scale bar is 50 nm. These images are processed from Fig. 4Bi. (ii) photobleaching steps of the bridged KMP-11 suggested multiple steps which infer the existence of KMP-11 oligomer in inter-membrane bridged condition. (iii) The multimerization status of KMP-11 in a bridged state was evaluated by counting the photobleaching steps for 735 single particle spots in TIRF microscope. Source data are available online for this figure.

DHE from the donor liposomes and its loading onto the acceptor liposomes (Fig. 5B). The CHOL (in this case, DHE) transfer rate was estimated from the FRET data by fitting the temporal FRET data using a linear calibration curve that correlated DHE concentration with the FRET signal (Fig. 5C,D). Increasing the concentration of r-KMP-11 reduced the time required for the FRET signal to reach a plateau (Fig. 5D,E). At a concentration of 20 μM (P/L ratio 0.02), r-KMP-11 exhibited the most efficient transfer of DHE, corresponding to ~10 DHE molecules per minute (Naito et al, 2019).

To understand the transport process further, we compared CHOL transport rates under two conditions: incubating 20 μM r-KMP-11 with DPPC-ERG SUVs for 30 min (Condition 1) and with same concentration of protein without any incubation (zero incubation time, Condition 2). Surprisingly, no incubation resulted in significantly slower CHOL transport (Appendix Fig. S4E) compared to the 30-minute incubation. We speculate that time-dependent oligomerization of KMP-11 on the LD surface enhances CHOL transport. Additionally, examining transport kinetics without ERG in the acceptor liposomes revealed delayed transport (Appendix Fig. S4F,G) highlighting the impact of ERG on CHOL transport from MΦ. According to prior reports using toxin sensors, CHOL sequestration can happen due to interactions with sphingomyelin as well as other phospholipids (Gay et al, 2015; Radhakrishnan et al, 2020; Sokolov and Radhakrishnan, 2010). However, when we measured the FRET signal from our in vitro CHOL transport assay using different mol percentage of CHOL (here, DHE), we observed a linear correlation between FRET signals and mol percentage of DHE (Appendix Fig. S4H). We did not find any equivalence point of CHOL transfer. This linear correlation indicates that CHOL transfer by KMP-11 presumably does not depend on CHOL accessibility or KMP-11 interaction can make membrane CHOL more accessible probably by its transient pore formation mechanism, which we have reported earlier (Halder et al, 2020).

We complemented the above CHOL transport assays (which were carried out using membrane mimics) by monitoring the transport of NBD-labeled CHOL during LD infection of host MΦ (Fig. 5F,G). Our data revealed that there is a significant transfer of NBD CHOL from host MΦ to LD promastigotes in an event of successful LD infection (Fig. 5F,G). In the absence of LD, a continuous persistence of NBD CHOL was observed in host MΦ without any significant loss in NBD fluorescence intensity (Fig. 5Fi), which is diminished and appears on attached/infecting LD (Fig. 5Fii). Measurement of fluorescence intensity and line scan profiles of NBD CHOL suggest that there is a significant decrease in the average NBD intensity in MΦ with a concomitant gain in NBD intensity on the surface of LD during the process of infection. The observation of a significant overlap between KMP-11 signal and NBD CHOL signal in infected LD further confirms KMP-11 induced CHOL transport (Fig. 5Gi,ii,iii).

## KMP-11 detaches from MΦ after CHOL transport

We have shown above that KMP-11 establishes a bridging mechanism between LD and MΦ to transport CHOL from MΦ membrane to LD. The next question we wanted to ask was whether, during the process of parasite internalization, KMP-11 remains attached to MΦ or if it detaches itself to retain on the surface of invading LD. To answer this, we infected host MΦs with LD promastigotes for 4 h, washed extensively to remove unattached free LD, and fixed. Subsequent staining of KMP-11 revealed no significant co-location of KMP-11 with MΦ surface marker CD11b (Fig. 6A). Furthermore, time lapse microscopy was used to monitor distribution of KMP-11 on the surface of invading LD (Fig. 6B,C). Our data clearly indicate that KMP-11 is not significantly retained on the surface of host MΦ during or at the end of LD internalization into host MΦ. This suggests that the retained KMP-11 on the LD surface after invasion may be reused towards

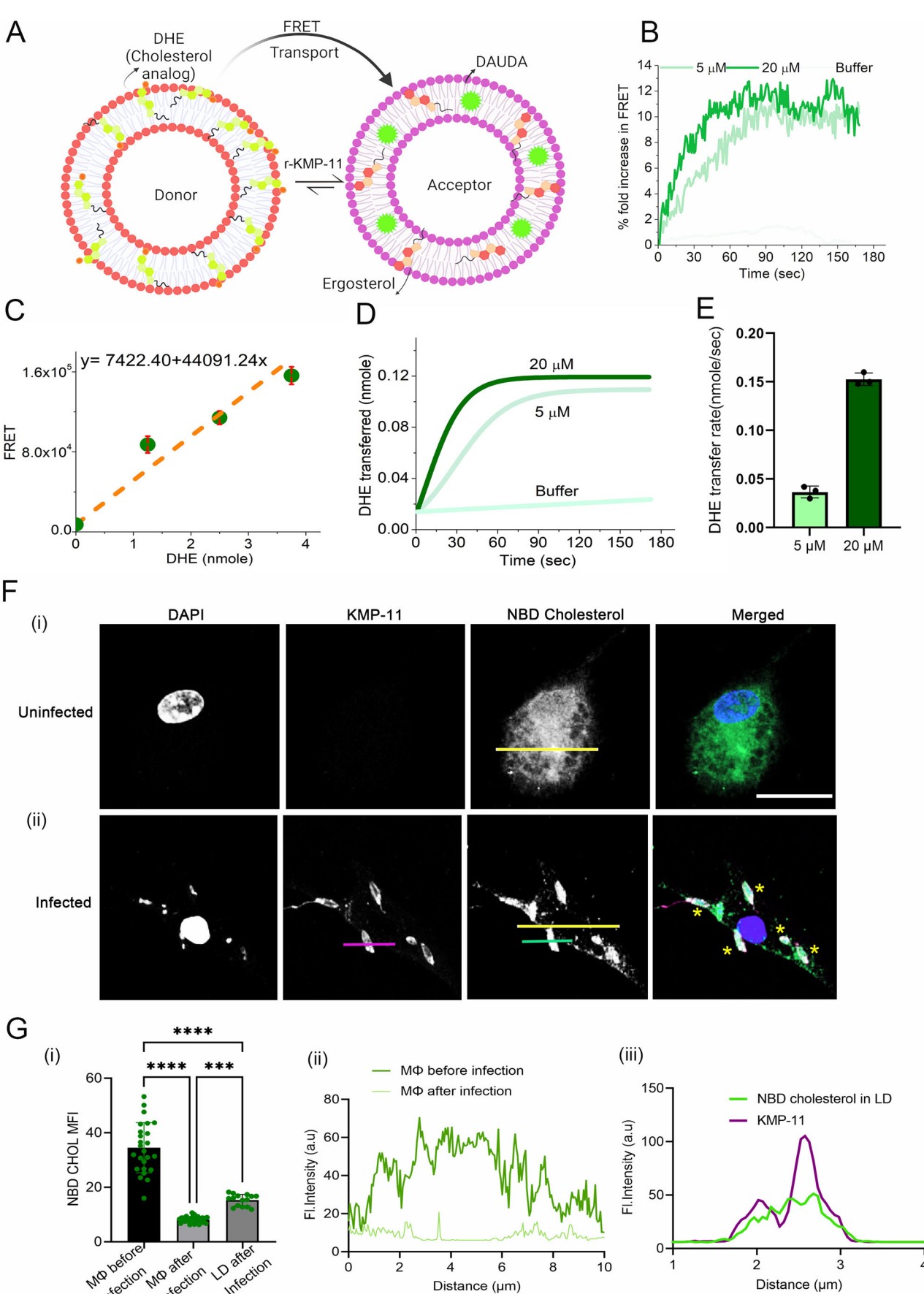

**Figure 5.  KMP-11 mediated CHOL transport.**

(A) A schematic diagram, which shows the design of the in vitro CHOL transfer assay. Donor liposomes (10% DHE, 90% DPPC) and acceptor liposomes [2.5% DAUDA:30%ERG:67.5%DPPC] were incubated with the r-KMP-11. In this assay, the amount of DHE (a fluorescent analog of CHOL) in liposomes was quantitatively measured using FRET between DHE and DAUDA. Transfer of DHE from donor to acceptor liposomes, which results in an increase in FRET between DHE and DAUDA in acceptor liposomes, was monitored using a fluorometer. (B) Time course of the fold increase in FRET signals. The percentage of fold increase in FRET signal due to the initial addition of KMP-11 (5 μM and 20 μM) indicates significant CHOL (here DHE) transport from DPPC-DHE membrane to DPPC-ERG membrane. (C) Calibration curve obtained from the plot of FRET signal against DHE concentrations in donor liposomes. This calibration curve was used to evaluate DHE transfer kinetics and transfer rates as shown later. The data were presented as the mean ± SD derived from three independent experiments ($n = 3$ for each data point). (D) DHE transfer kinetics in the presence of 5 μM and 20 μM r-KMP-11, which shows that higher concentration results in a higher rate of CHOL transfer. (E) Violin-plot shows DHE transfer rates indicating considerably higher for 20 μM r-KMP-11 compared to that for 5 μM r-KMP-11. The data were presented as the mean ± SD. The number of biological replicates was $n = 3$. (F) MΦs were pre-labeled with NBD CHOL for 16 h followed by wash and rest for 2 h after which LD infection for 4 h or left uninfected. Confocal images showing (i) constant signal of NBD distributed throughout MΦ in uninfected condition (ii) decrease in the NBD CHOL intensity from the MΦ membrane and co-localization of KMP-11 and NBD in case of LD infection. KMP-11 is labeled with Alexa 594 (magenta), NBD CHOL is showing green and the nucleus is DAPI (blue) stained. Yellow * indicates the white region (KMP-11 signal overlapped with NBD CHOL). Compared to extracellular LD, KMP-11 shows a slightly punctated distribution in LD attached with the macrophage membrane. Scale bar is 10 μm. (G) (i) Plot showing the distribution of mean NBD fluorescence intensity in MΦ before ($n = 25$), and after LD infection ($n = 19$) and on LD membrane after infection ($n = 15$). Here $n$ stands for the number of cells counted for analysis using the images of three independent imaging experiments for each group ($n = 3$/group). The plot shows a gain of NBD fluorescence in LD surface after infection along with a loss of intensity on the MΦ surface. The data were expressed as the mean ± SD. Statistical significance in the observed differences was determined through ANOVA utilizing Graphpad prism software (version 9.0). ***$P$ value = 0.0008, ****$P$ value = 0.000074. (ii) Line scan of NBD fluorescence in MΦ before infection and after infection (yellow solid line in the images F (i), (ii)). (ii) Line profiles (magenta and green lines in the image F) for NBD CHOL (green) and KMP-11 (magenta) on LD surface after infection. The scale bar in (F) is 10 μm. Source data are available online for this figure.

further internalization into the host MΦ. This microscopy data was also complemented by employing TIRF microscopy imaging on SLB (made of PEG DPPC-CHOL) which was treated with r-KMP-11 and subsequently DiD-labeled DPPC-ERG vesicles were introduced to the chamber. Temporal imaging showed the detachment of the docked vesicles from the SLB surface (Fig. 6D,E) thus supporting the detachment process as observed in live cell imaging of LD with MΦ. While this is in apparent agreement with the biophysical interpretation as KMP-11 has stronger binding towards DPPC-ERG (i.e., at LD membrane) when compared to DPPC-CHOL (at MΦ membrane), there is one important contradiction. After CHOL transport, the MΦ membrane is expected to be CHOL deficient representing So (and not Lo) state, which has the strongest binding affinity towards KMP-11. This contradiction essentially means that in order to enable the detachment of KMP-11, DPPC membrane must experience a further change in its morphology as explained below.

## KMP-11 binding induces a gel (So) to fluid (Ld) transition in DPPC

Since the secondary structure of r-KMP-11 conformation remains unaltered by lipid binding (Fig. EV4A), we next ATR-FT-IR to determine if r-KMP-11 binding leads to any change in DPPC membrane. A schematic diagram in Fig. EV4B illustrates the different regions of the lipid. We examined the choline moiety (head groups) by analyzing the –C–N–C– vibrations (Fig. EV4C) and the phosphatidyl group by studying the $PO_2$ vibrations (Fig. EV4D). The tail region of the lipid was investigated by monitoring the symmetric deformation of $CH_2$ (Fig. EV4E) and symmetric stretching (Fig. EV4F) vibrations. Protein binding induced significant changes in both the head and tail regions of the lipid. FTIR data clearly demonstrated that r-KMP-11 caused a thinner DPPC bilayer with increased flexibility (Fig. EV4Gi,ii). Furthermore, the nonhydrogen bonded frequency increased more than the hydrogen bonded ester carbonyl frequency, showing a dose-dependent relationship (Fig. EV4Giii). We observed an increase in the extent of nonplanar kink+gtg' conformers (Lewis

and McElhaney, 2013) in DPPC nonpolar tail region (Figs. 7Ai,ii and EV4Hi,ii) due to r-KMP-11 binding.

We have shown above that KMP-11 increases the fluidity in MΦ membrane leading to a phase transition. Since membrane phase transition is typically and conveniently probed by temperature dependence (Chapman, 1975), we used this method to monitor the effect of KMP-11 in MΦ membrane phase behavior. By determining the GP of Laurdan at varying temperatures, we found that the MΦ membrane undergoes a phase transition between the So and a more fluid Ld state (Fig. EV4Ii). At low temperature and in the absence of KMP-11, the Laurdan fluorescence spectrum was characterized by a sharp emission maximum at 440 nm (λex = 375 nm) (Fig. EV4Ii). With the increase in temperature, a second shoulder appeared at 490 nm, which was accompanied by a large decrease in fluorescence intensity (Fig. EV4Ii) and this transition from So to Ld yielded a mid-point at 45 °C. Interestingly, when we added r-KMP-11 at low temperature, the MΦ membrane was found to be at Ld as we found significant drop in GP value (Fig. EV4Iii,iii).

We then complemented MΦ temperature dependence data using DPPC, whose phase transition behavior (between So and Ld phases) has been extensively investigated (Leekumjorn and Sum, 2007). We systematically determined the melting temperature ($T_m$) of DPPC in the presence of different concentrations of r-KMP-11 (Fig. 7B; Appendix Fig. S5A,B; Appendix Table S3). Inset of Fig. 7B show the variation of $T_m$ with r-KMP-11/DPPC (P/L) ratios, where we found that the increase in the relative concentration of the protein in P/L reduced $T_m$. At P/L of 0.004 and above, DPPC remained at the Ld state even at the physiological temperature (which is much lower than the chain melting temperature of DPPC in the absence of protein, 44 °C) (Appendix Fig. S5C).

Subsequently, we confirmed r-KMP-11 induced phase transition in DPPC using small angle X-ray scattering (SAXS). Figure 7Ci,ii shows the intensity profiles obtained from SAXS at different P/L. Measurements were performed at two different temperatures of 25 °C and 50 °C respectively (Fig. 7Ci,ii). A large number of lamellar reflections at 25 °C (Fig. 7C, and relatively less number of lamellar reflections at 50 °C for only DPPC (Fig. 7C) characterizes the gel (at 25 °C) and fluid (at 50 °C) phases, respectively. In the

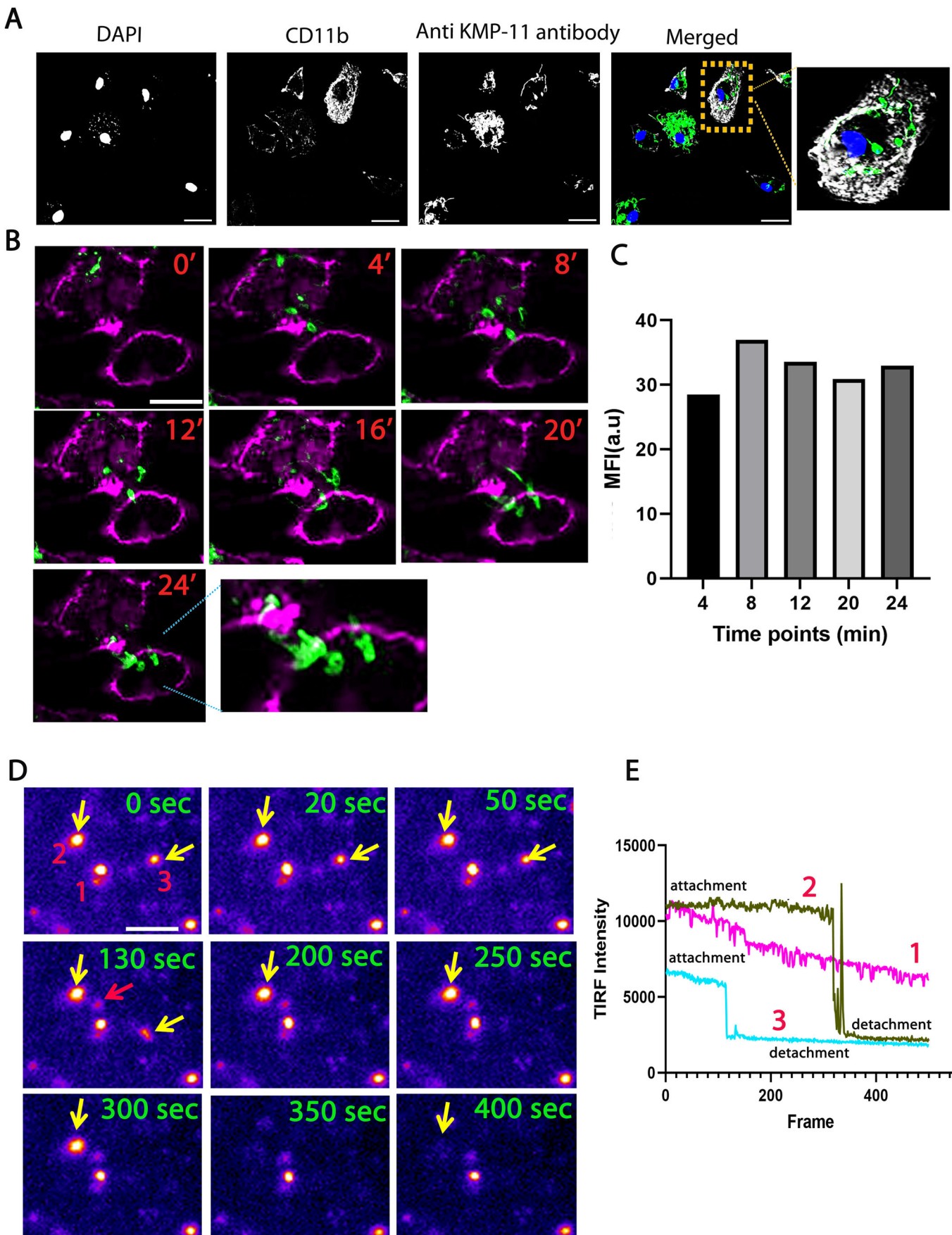

◀ **Figure 6. KMP-11 is retained on LD surface during the process of MΦ invasion.**

(A) Confocal images showing distribution of KMP-11 on the LD during host infection. MΦs were infected with LD for 4 h, washed, and fixed. KMP-11 on LD surface is stained with anti-KMP-11 antibody and MΦ membrane is stained with CD11b. Scale bar is 10 μm. (B) Time-lapse images of MΦ expressing CD11b-RFP infected with LD_KMP-11_KO/KMP-11 Complemented lines expressing EGFP for 4 h. Images were taken for every 4 min with initial capture denoted as $T_O = 0$ min (4 h time point). Total seven images were captured. EGFP signal is retained on the LD surface without getting distributed over the MΦ membrane. The scale bar for each image is 10 μm. (C) Graph showing temporal GFP fluorescence signal on LD surface during the invasion process of LD_KMP-11_KO/KMP-11 Complemented lines expressing EGFP. (D) Temporal images using TIRF to show the docked DiD-labeled DPPC-ERG vesicles on PEG cushioned DPPC-CHOL SLB surface pre-treated with r-KMP-11. The images also show the detachment of some single vesicles (here 2 and 3) with time. Here, the yellow arrow indicates the single vesicles docked on the SLB via KMP-11 and the red arrow indicates the newly docked vesicle. Images were processed using the LUT tool in Fiji image j. The dimension of each image was kept at 12.03 × 10.76 μm². Scale bar is 2 μm. This detachment data was generated and analysed from the same experiment as shown in Fig. 4Bi. (E) TIRF intensity of selected three vesicles (1, 2, 3 in the image in D) with time suggests the detachment process. Source data are available online for this figure.

presence of r-KMP-11, the number of lamellar reflections reduced even at 25 °C, suggesting the presence of more flexible bilayers with respect to the gel phase (Fig. 7C, P/L 0–0.004). The SAXS profile at 25 °C for P/L of 0.004 showed the presence of only two distinct reflections, which was identical to the SAXS profile of DPPC fluid phase at 50 °C, respectively. From the SAXS experiments, we inferred that the bilayer undergoes gel (So) to fluid (Ld) transition at 25 °C for P/L of ~0.004 and above.

We then complemented the above measurements using several fluorescence-based assays. The first one was REES (Red Edge Excitation Shift) measurements using an NBD PC. REES assay has been used popularly to study membrane phase transition, and its principle is well established (Raghuraman et al, 2007). We found that the value of REES increased with increasing temperature (Appendix Fig. S5D), from which we calculated $T_m$ (Appendix Fig. S5D; Appendix Table S3). We observed a decrease in $T_m$ as we increased the concentration of r-KMP-11, indicating slow rate of solvent relaxation relative to the NBD lifetime due to the re-orientation of solvent dipoles around the excited state of fluorophore. Such solvent relaxation in the immediate vicinity of fluorophore depends on motional restriction, i.e, the change in the rotational diffusion of lipid molecules induced by protein in the lipid bilayer. Next, we carried out dithionite-induced fluorescence quenching experiments using NBD-PC in DPPC membrane, which showed an increase in the relative fraction of NBD analog on the outer leaflet ($P_0$) in the presence of r-KMP-11 ($P_0$ for NBD-DPPC ~ 64.25% whereas for NBD-DPPC-r-KMP-11 ~ 91.89%) (Appendix Fig. S5E,F). This result indicated that the protein binding significantly enhanced the flip-flop rate of lipid molecules. This data also implied a reduction in membrane rigidity with concomitant increase in the flexibility.

## Hydrophobic moment of amino-terminal-KMP-11 regulates the So-Ld transition in DPPC

An in silico (OPM, http://opm.phar.umich.edu/server.php) calculation shows that the interaction between KMP-11 and membrane would occur through the amino-terminal (residues 1–19) of the protein (Appendix Fig. S6A). To validate this experimentally, we used tryptophan scanning mutagenesis (using eight single tryptophan mutants described above) to study the binding interactions with DPPC membranes (Fig. 7D). The difference in binding constant between the WT and a tryptophan mutant ($\Delta K_a = K_{a,WT} - K_{a,mutant}$) would provide an estimate of the effect of mutation site on the binding constant. In addition, we calculated the difference in $K_{sv}$ for each mutants ($\Delta K_{sv} = K_{sv}$(free)-$K_{sv}$(membrane bound)) using the fluorescence quenching measurements. We find that the mutant

Y5W (tryptophan inserted at the amino-terminal) shows the maximum effect on $\Delta K_a$ and $\Delta K_{sv}$ values (Fig. 7E; Appendix Fig. S6B; Appendix Table S2), which is clearly in line with the OPM prediction.

The involvement of the amino terminus of KMP-11 towards membrane binding was further determined by calculating the energy transfer efficiency (E) between inserted tryptophan (donor) and membrane-bound DHE (acceptor) (Fig. 7F, inset). We observe that Y5W shows the highest E of ~20%) (Fig. 7F). In contrast, Y48W and Y89W exhibited intermediate values ($E_{Y48W}$ ~ 6.4% and $E_{Y89W}$ ~ 8.5%, respectively). F62W shows the lowest ($E_{F62W}$ ~ 1.72%) (Fig. 7G; Appendix Table S4). From the energy transfer results, we observe that apparent donor-acceptor distances follows the trend: Y5W (11.39 Å) <F19W(12.8 Å)<Y89W(13.64 Å)<Y48W (14.13 Å)<F51W(15.6 Å)<< F62W (24.67 Å) (Appendix Table S4; Fig. 7G).

When arranged in terms of residue-specific hydrophobicity (H) and polarity (P), the amino-terminal of KMP-11 (1–19) ([1]MATTYEEFSAKLDRLDQEF[19]) forms a mirror stretch (Fig. 7H). Since an ideal α–helix makes exactly five turns, we used a helical wheel model to investigate the hydrophobic face and to calculate the hydrophobic moment ($\mu_H$) of the specific interacting domain. Heliquest-compuparam (http://heliquest.ipmc.cnrs.fr/cgi-bin/ComputParamsV2.py) analysis show that helical wheel model of KMP-11-amino-terminal domain (1–19 AA) contains a hydrophobic face consisting of LFMFLY (Fig. 7H; Appendix Table S5) with a net hydrophobic moment of 0.505. Subsequently, we designed a few new stretches theoretically using this helical wheel model and altering the positions of the amino acids in a way that the hydrophobic moments would be gradually altered (Fig. 7Hi,ii,iii; Appendix Table S5) without changing the net hydrophobicity. We computed the binding energies between the designed peptides and membranes ($\Delta G_t$, Kcal/mole), which showed that $\Delta G_t$ increased with the increase in the $\mu_H$ of the sequence (Appendix Table S5). To investigate in detail why sequence symmetry (which makes the difference in the hydrophobic moment and membrane affinity) is a critical determinant for the binding phenomenon, we determined the number of residues (N), which constituted the mirror stretch. Appendix Table S5 also shows that there was a direct relation between N and $\Delta G_t$. To validate this relation experimentally, we synthesized three different peptides of varying hydrophobic moments with similar hydrophobicity (namely peptide1 (pep1), peptide2 (pep2), and peptide3 (pep3) in Fig. 7H). Binding studies of these peptides with DPPC model membrane suggested that binding affinity followed the trend pep1>pep 2> pep3 (Fig. 7I, inset; Appendix Fig. S6C). By measuring Laurdan fluorescence, we found that So-Ld transition temperature ($T_m$) was directly proportional to $\mu_H$ of the peptides

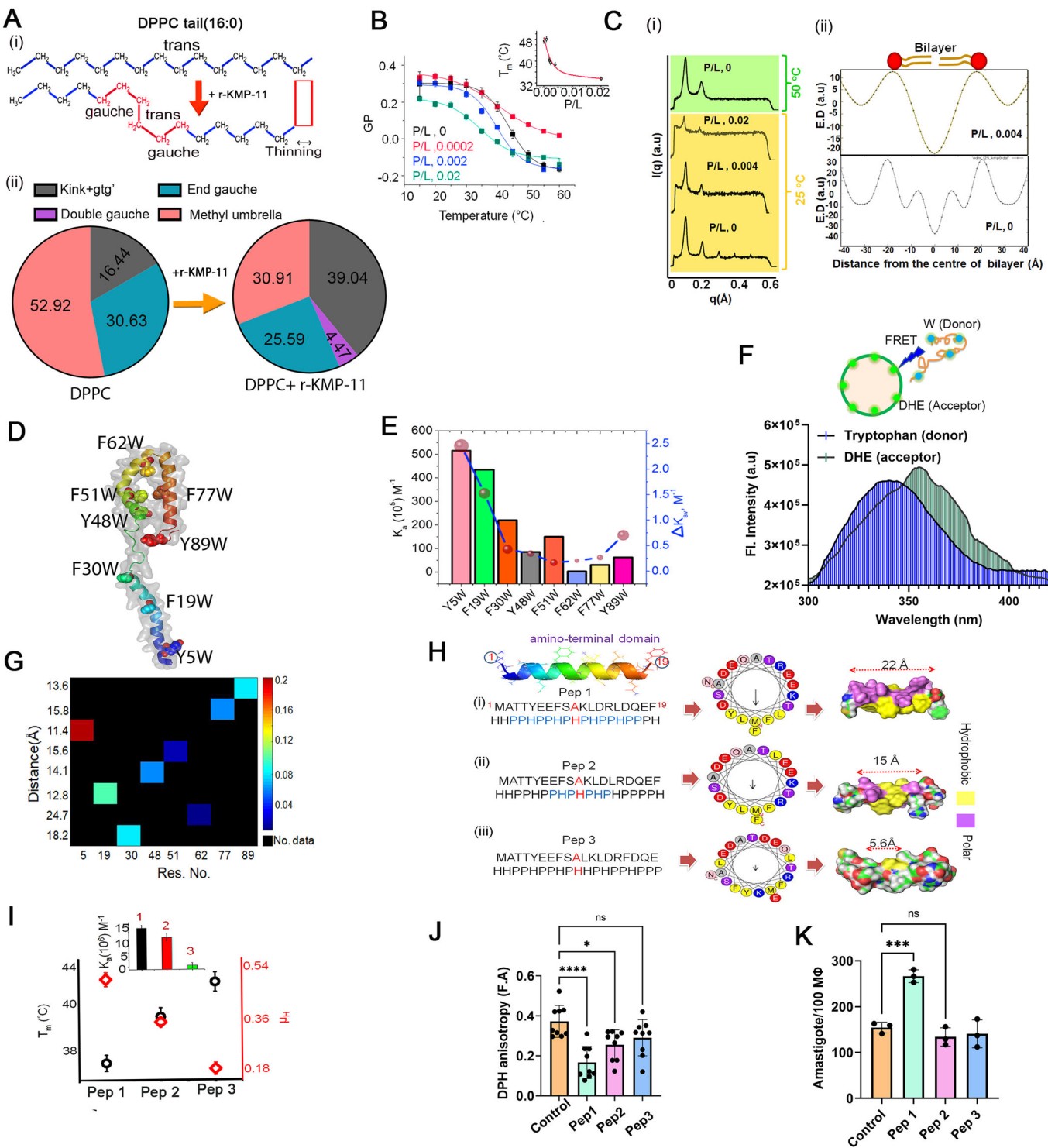

(Fig. 7I). Next, we found that MΦ membrane fluidity was higher for pep1 when compared to pep2 and pep3 (Fig. 7J; Appendix Fig. S6D). When we used peptide pulsed MΦ for the LD infection studies, we found that intracellular parasite count was greater for pep1 compared to pep2 after 4 h of incubation (Fig. 7K). No considerable increase in parasite load was observed when MΦ cells were pulsed with pep3 (Fig. 7K).

# Discussion

## KMP-11 induced membrane phase transition (Lo to Ld) and host cell invasion

Subversion of host membrane machinery is important for the uptake, survival, and replication of diverse intracellular pathogens.

**Figure 7. The amino terminus of KMP-11 induces phase transition in DPPC membranes.**

(A) (i) The structural orientation of DPPC (16:0) nonpolar tail domain. The trans-conformations are favorable when the lipid chain is in the ordered phase (So and Lo). The transformation from trans-orientation to the gauche iso-form (gtg') results in the increase in disorder-ness in the membrane (Ld phase). This conformational transition (trans to gtg') occurs when DPPC SUVs are heated above the transition temperature (42 °C) or DPPC SUVs are treated with r-KMP-11. (ii) The morphological states were evaluated by measuring quantitatively the $CH_2$ wagging band frequency (1280–1440 $cm^{-1}$) of the nonpolar region of DPPC using the FTIR de-convolution technique. The color pie charts indicate the percentage population of different lipid conformations in the absence (left pie chart) and presence (right pie chart) of r-KMP-11. These percentage populations were evaluated from the de-convolution as shown in EV4H. FTIR spectral signatures of the $CH_2$ wagging band frequency of DPPC in the absence and presence of r-KMP-11 indicates the significant increase in the gauche conformer population (1367 $cm^{-1}$ signature) in the lipid structure further suggesting the r-KMP-11 induced morphological changes in lipid chains. (B) The variation in GP with an increase in temperature at different P/L molar ratios. The inset shows the change in DPPC-chain-melting temperature ($T_m$) with varying P/L. All data were expressed as the mean ± SD derived from three independent experiments ($n = 3$/group). (C) (i) SAXS data of DPPC MLV at different P/L ratios at 25 °C and 50 °C. (ii) Electron density maps at two different P/L ratios at 25 °C as derived from SAXS results. (D) The model structure of KMP-11. The inserted tryptophan residues were marked. (E) Plot of binding affinities ($K_a$, $M^{-1}$) of eight single tryptophan mutants with DPPC model membranes as evident from the quenching of DPPC SUV bound DiD fluorescence and tryptophan exposure (in terms of $\Delta K_{SV}$) for single tryptophan mutants. (F) Representative FRET signal of tryptophan residue (donor) in Y5W mutant of r-KMP-11 and DHE (acceptor) embedded in DPPC membrane. The inset shows the donor and acceptor pairs. (G) Contour heatmap shows the plot of donor-acceptor distances and efficiency of energy transfer against each tryptophan mutant. The color bar represents energy transfer efficiency. (H) Sequences and characteristics in terms of hydrophobic and polar arrangements of three synthesized peptides [(i) pep1, (ii) pep2, and (iii) pep3]possessing different hydrophobic moments as revealed by the helical wheel model. (I) Hydrophobic moments and phase transition temperatures of DPPC SUV induced by three peptides plotted altogether. Inset shows the plot of binding affinity of three peptides [(i) pep1(1), (ii) pep2 (2), and (iii) pep3(3)] towards DPPC model membrane. The data were represented as the mean ± SD derived from three ($n = 3$/group) independent experiments. (J) DPH anisotropy changes in MΦ membranes by three peptides. The data were expressed as the mean ± SD derived from nine ($n = 9$) independent experiments. Statistical significance in the observed differences was determined through ANOVA utilizing Graphpad prism software (version 9.0). *$P$ value = 0.0230, ****$P$ value = 0.000093, $^{ns}P$ value = 0.1711. (K) Parasite load inside MΦs in 4 h of incubation with three different peptides. The data were expressed as the mean ± SD derived from three ($n = 3$) independent experiments. Statistical significance in the observed differences was determined through ANOVA utilizing GraphPad prism software (version 9.0). ***$P$ value = 0.0001, $^{ns}P$ value = 0.5113. Source data are available online for this figure.

Also, host membrane CHOL is a crucial element in maintaining the integrity and cellular functionality of the host cell membranes. Different pathogens have developed smart mechanisms of transferring CHOL from the host in order to manipulate host membranes. For example, CHOL in the membranes of *Helicobacter* spp. (Hirai et al, 1995), *S. aureus* (Haque et al, 1995), Anaplasma-phagocytophilum (Lin and Rikihisa, 2003) and Chlamydia EB (elementary body, infectious form) and RB (reticulate body, a vegetative form) membranes (Wylie et al, 1997) is of host origin. A sterol-binding protein in *Toxoplasma* has been identified that presumably optimizes pathogen handling of host cell derived CHOL (Lige et al, 2009). It has been found that toxoplasma contains an ancestral D-bifunctional protein containing two sterol-carrier protein-2 domains, which are needed for lipid uptake and trafficking. Recently it has been demonstrated that Mce1 functions as a fatty acid transporter in MTB, which facilitates CHOL and fatty acid import via Rv3723/LucA (Nazarova et al, 2017). In addition to bacteria and protozoan pathogens, viruses also exploit the host lipidome by extracting CHOL from the host membrane thus modulating its property. It has been shown recently that SARS-COV-2 S protein can quench CHOL from the host and it is one of the most important parts of infection (Correa et al, 2021). Although *Leishmania* infection has been reported to quench CHOL from the host membrane increasing the membrane fluidity, the mechanism of CHOL transport was not known, neither the LD-specific proteins, which could drive this transport and manipulate host cell membrane, were identified. In this study we show that LD, just like many other intracellular pathogens, is equipped with a surface protein for CHOL transport from the interacting host membrane.

Figure 8A summarizes the presented data to provide a molecular understanding of the process of LD-host cell invasion, in which KMP-11 plays a major role. It may have implications in three events related to the host–pathogen interactions: (a) the initial attachment with the host using a KMP-11 bridge, (b) the CHOL transfer from the host to invading pathogen, and (c) the process of KMP-11 detachment.

Although these events are specific to LD, a generalized picture may be derived as KMP-11 is conserved across *Leishmania* spp, and KMP-11 like proteins (like Mce1 in MTB, D-bifunctional protein in toxoplasma and SARS COVID S-protein etc) do exist for other intracellular pathogens. We believe that these events are coordinated by a careful orchestration of pathogen biology and the physical chemistry of lipids/lipid–protein interactions.

The process of host cell infection by *Leishmania* parasites has been long believed to be a passive invasion process which depends on receptor mediated endocytosis (Ueno and Wilson, 2012). However, there is now evidence from real time imaging data, which shows that LD can induce local lysosomal exocytosis and host cell plasma membrane wounding indicating possible involvement of an alternative mechanism (Forestier et al, 2011). Interestingly multiple recent reports have also suggested that *Leishmania* parasites can infect non phagocytic cells by inducing phagocytosis independent host cytoskeleton disassembly (Caval-cante-Costa et al, 2019; Haldar et al, 2020). All these evidences suggest that the mechanism of host invasion by *Leishmania* parasites is still a black box which requires further characterization.

Till date only two widely studied *Leishmania* specific molecules have been implicated in establishing *Leishmania* infection. These are zinc-metalloprotease gp63 and lipophosphoglycan (LPG), a complex glycolipid. While gp63 has primarily been implicated in the cleavage and degradation of various host derived kinases and transcription factors (Isnard et al, 2012), the role of LPG has been suggested in the prevention of complement mediated lysis (Clarkson, 2003) during infection process. Interestingly, it has been reported that *L.mexicana* mutants lacking LPG are capable of causing infection (Ilg, 2000). Similarly, it has been shown for LD and *L.major* that, although ablation of LPG and gp63 results in compromised infection, it does not abrogate the invasion process (Handman, 2001; Ilg, 2000; Naderer et al, 2004; Olivier et al, 2012). This observation, along with the fact that both LPG and gp63 have significantly low expression in amastigotes (Schneider et al, 1992; Späth et al, 2003), clearly indicate that apart from gp63 and LPG other parasite surface molecules can

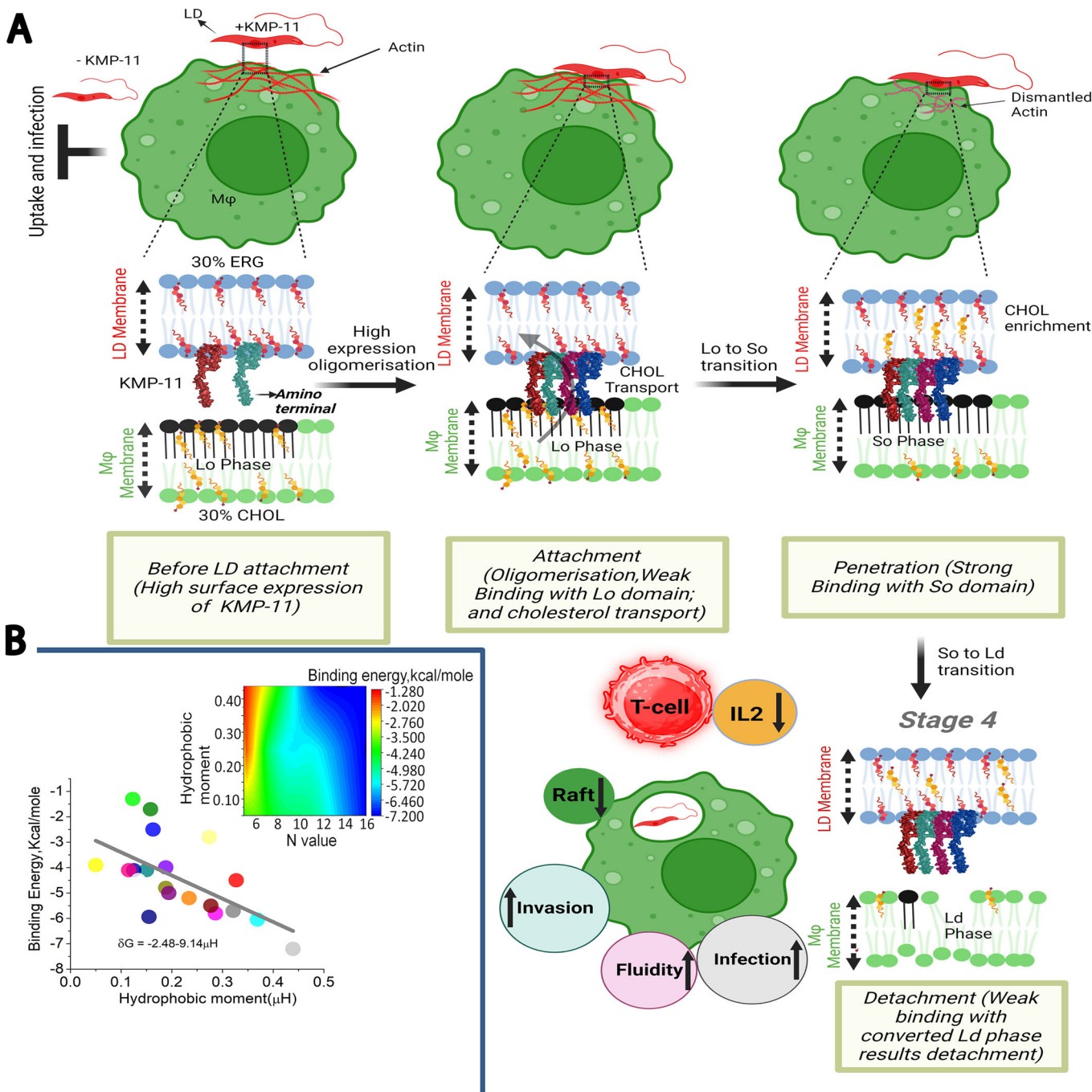

**Figure 8. KMP-11 induced membrane phase transition model for *Leishmania*-infection.**

(A) Our model shows how KMP-11 induces the infection in MΦ through host membrane modification that has been described by four principal stages. Stage 1: Significant abundance of KMP-11 molecules is exposed on the surface of LD as suggested by our results. Based on our data, it is also evident that KMP-11 remains attached by its 50–65 AA sequence stretch with ERG containing LD membrane keeping amino terminal away. Stage 2: Attachment: KMP-11 acts as a bridge between LD and MΦ during the initial attachment. It may be recalled from the attachment/invasion data and TIRF microscopic observation that without KMP-11, the attachment efficiency of parasites is significantly reduced. Since KMP-11 tends towards self-association, a high copy number of KMP-11 would favor its oligomerization (mostly tetramer formation) at the LD surface. The difference in the sterol composition between LD and MΦ surfaces and the change in membrane properties play crucial roles in the establishment of the LD infection at the host. At this stage MΦ membrane is in Lo phase, which would result in KMP-11 weak binding with Lo domain and CHOL transfer from the host to LD. CHOL quenching from the MΦ membrane may result in actin disassembly. Stage 3: Penetration: Due to CHOL quenching, the So phase (with the strongest KMP-11 binding) would enable the penetration of the amino-terminal region of the protein. Stage 4: Detachment: This stage is characterized by a KMP-11-induced phase transition from So phase to Ld. Our results unveiled the weak binding affinity of KMP-11 with the Ld phase of the lipid membrane, facilitating the detachment, which is also shown by the imaging experiments. (B) This figure shows a linear correlation between membrane binding energy and hydrophobic moment of the interactomes of twenty different pathogenic proteins (please see Appendix Table S6 also). The inset shows a contour plot between the hydrophobic moment, N-values, and membrane-binding energy for different membrane-binding pathogenic proteins. The color bar indicates the membrane binding energy. Source data are available online for this figure.

also contribute through invasion of *Leishmania* parasites (Clarkson, 2003; Späth et al, 2003). In this regard, a leucine-rich-repeats protein in *L. amazonensis* has recently been identified, which increases in vitro invasion in host MΦ by interacting with a Glucose receptor on its surface. We find here that a significant amount of KMP-11 is retained on the parasite surface, which interacts with host MΦs forming a transient bridge. In this regard, it should also be mentioned that although LPG covers most of the promastigote surface, there is a significant re-orientation of LPG during the process of host infection, thus possible interaction between small proteins like KMP-11 on *Leishmania* surface and host MΦ membrane become feasible (Forestier et al, 2015; Horta et al, 2020). Our FRET data conclusively showed that KMP-11 on the LD surface can strongly bind with macrophage membrane during attachment process, as we found the donor-acceptor distance ranges 20–60 Å (Appendix Fig. S3). Our data further suggest that interaction between LD derived KMP-11 and MΦ membrane results in quenching of CHOL from MΦ membrane to invading LD parasites.

The absence of KMP-11 severely reduces parasite size and the length of the flagella. Since it is well known that the persistent flagellar activity leads to initial attachment (stage 1, Fig. 8A) with host cell membrane, the stunted flagella probably explain defective attachment and subsequent host cell invasion as observed with LD_KMP-11_KO parasites. The observation of the reduced size of LD_KMP-11_KO lines is also in line with the previous observation that KMP-11 can help to maintain overall bilayer morphology of LD (Jardim et al, 1995) and not only acting as an invasive factor. In addition, we observed significant reduction in LD infection in presence of anti-KMP-11 antibody (Fig. 1F,G), and complete impairment of MΦ attachment and invasion displayed by metacyclic LD_KMP-11_KO parasites (Fig. 2F,G). The presence of r-KMP-11 and complementation of LD_KMP-11_KO lines with WT copy of *kmp-11*, significantly elevate the MΦ infection. Interestingly, we have also noticed that LD promastigotes, which were continuously propagated in vitro for more than 200 passages (LD > P200), and showed compromised ability to infect hosts MΦ also displayed a significant loss of KMP-11. All these observations substantiate further that KMP-11 could contribute as a major virulence factor in LD further indicating primary involvement of this protein in the process of MΦ infection. In addition, both KMP-11 and LD treatments increase the fluidity of MΦ plasma membrane significantly and CHOL treatment can revert back the MΦ membrane morphology and reduces infectivity as well, which further inferred the interplay between KMP-11 directed membrane fluidity alteration and LD infection. It should be mentioned here although KMP-11 appears to play a critical role in initial MΦ infection; its role in amastigote infection process within mammalian host still needs to be validated. We would also like to mention that our attempts to generate axenic amastigotes for LD_KMP-11_KO lines was unsuccessful either due to inability of KMP-11 depleted promastigotes to get transformed into axenic amastigotes or due to our inability to properly detect KMP-11 depleted axenic amastigotes owing to their severe reduction in size and possibly altered morphology. Hence, it was further not possible for us to determine the role of KMP-11 in the process of amastigote infection.

As depicted at the stage 2 of Fig. 8A, the protein KMP-11 would remain in a 'hanging bat' orientation keeping the amino-terminal domain away from the parasite surface, acting as a bridge between LD and MΦ. The bridging is possible because KMP-11 has distinct binding regions of its sequence for different membranes (DPPC-ERG aka LD vs. DPPC-CHOL aka host MΦ, Figs. 3H,I

and EV2E,G). Single molecule imaging studies using TIRFM, FCS and TEM imaging indicate that KMP-11 is possibly a tetramer in its bridged state between LD and MΦ. Since both LD (approx. size ~72.98 μm³) and MΦ (approx. size ~4990 μm³) are many orders of magnitudes larger than KMP-11 (approx. size ~14.539 nm³), the multimer formation may be necessary to stabilize the complex. In addition, the expression efficiency ($2 \times 10^6$ copy/LD) of KMP-11 at LD surface is very high, which is a favorable condition towards self-association of a protein like KMP-11. Moreover, as mentioned earlier r-KMP-11 has an inherent tendency towards self-association, which we showed both by experiments (Fig. 4E–J) and structural analyses (Appendix Fig. S4A,B).

KMP-11 possesses stronger affinity towards the gel state/So domain (DPPC SUVs or MΦ without CHOL) compared to the Lo (DPPC-CHOL or host MΦ membrane) state as evident from our binding study. As a result, CHOL transfer from MΦ (or DPPC-CHOL) by KMP-11 may be beneficial towards establishing its stronger binding and penetration within MΦ plasma membrane (stage 2 and 3 in Fig. 8A). In contrast, KMP-11 binding is weaker for the Ld state of the membrane. The Ld state is formed by the protein induced Lo to Ld phase transition. Since the presented data clearly show no significant retention of KMP-11 on host MΦ membrane as the LD parasites completes their process of invasion into the host, it is likely that KMP-11 detaches itself from MΦ once the Ld state is attained (stage 4, Fig. 8A). To summarize, the data presented in this manuscript strongly indicate that the protein KMP-11 and membrane CHOL complement each other in establishing the early phase of LD infection in the host MΦ cells. During the internalization process, KMP-11 may play a role in the transfer of host membrane CHOL and makes a CHOL coat around the intracellular parasite membrane, which has been reported previously (Semini et al, 2017).

## Conservation of the local sequence symmetry in global processes of host cell infection

Using tryptophan scanning mutagenesis and synthesized peptides, we show that a special kind of sequence symmetry of the amino-terminal domain of KMP-11 based on residue-specific hydro-phobicity and polarity is responsible for the membrane phase transition. A mathematical interpretation of the phase transition of the lipid membranes due to protein binding comes from previous literatures, which show that the incorporation of proteins into bilayers can alter membrane tension resulting in a membrane curvature (Elson et al, 2010; Shi and Baumgart, 2015). Recent study using micropipette aspiration along with linear instability theory show that membrane tension and coupling between local protein density and membrane curvature mediate the shape change in the phospholipid GUV (Shi and Baumgart, 2015). It is likely that strong binding of KMP-11 can cause membrane instability due to protein density fluctuations. A threshold protein to lipid ratio is required for the onset of phase transition, which is clearly validated by the presented SAXS data. We find here that the gel to fluid transition occurs at or above a P/L of 0.004. It should be noted that the expressed copy number (Fuertes et al, 1999) of KMP-11/LD parasite surfaces is $\sim 2 \times 10^6$. On the other hand, our calculation shows that $\sim 5 \times 10^4$ copy number of KMP-11/MΦ (detailed calculations are provided in Appendix Methods) is required for phase transition induced LD infection. Therefore, P/L, 0.004 is

sufficient for MΦ phase transition and inducing infection at physiological temperature.

Protein–lipid interactions are driven by several factors including hydrophobic interaction. Although it is generally accepted that globular proteins fold with a hydrophobic core and hydrophilic exterior, the centroid of the spatial distribution of amino acid residue provides the origin of moment expansion. This spatial distribution has been described as a two components spherical model where interacting partners had been considered as spheres (Silverman, 2001). Using these considerations, we derived a "hydrophobic moment and sequence symmetry-oriented phase transition model" which correlates (Eq. (1)) the transition temperature of the membrane ($T_m$) with the number of amino acid residues presents in the mirror sequence of the interacting protein stretch (N): (please refer to the Appendix Methods for the details):

$$T_m = 42.02 - 0.466N \tag{1}$$

To summarize, the hydrophobic moment ($\mu_H$) of a sequence (and not the hydrophobicity, as all peptides used here have the same hydrophobicity) is critical to alter the conformation of the phospholipid hydrocarbon tail region through bond rotation during protein-membrane interactions leading to the change in membrane fluidity. We found a strong connection between the N values (no. of residues constituted the mirror stretch) and the apparent binding free energy of the proteins, an observation which validates the present model (Fig. 8B, inset). It was found that the amino-terminal mirror domain pattern of KMP-11 remains conserved in other pore forming and disease promoting proteins and most of the proteins contain this sequence stretch in their pore forming regions (Appendix Table S6). We considered twenty pathogenic proteins in which this sequence pattern is conserved, and we measured the binding energies and hydrophobic moments of the membrane interacting regions (Appendix Table S6). Interestingly, we observed a linear correlation between $\mu_H$ and binding energy (Fig. 8B). Thus, our results show for the first time a nice correlation between $\mu_H$, binding energy and N for different membrane interacting pathogenic proteins.

Collectively, we demonstrate here that a novel mechanism of protein binding induced membrane phase transition facilitates LD infection. We have also developed a general framework of this mechanism by suggesting that significant N-value of the membrane interacting domain would be globally essential for the disease-promoting proteins which facilitate the disease process through host membrane destabilization strategy. This study will help towards the development of therapeutic interventions by means of targeting the specific patterned sequences against host cell infection. Although, our results showed that KMP-11 forms a bridge between LD and host plasma membrane, we cannot negate the possibility that active shedding of KMP-11 in close proximity of the host membrane may be an alternative strategy which facilitates LD invasion into the MΦs. The increase in the number of invading LD in the presence of exogenous KMP-11 suggests that KMP-11 might be acting across a broader area of the MΦ membrane, potentially altering cholesterol levels throughout the membrane rather than just at specific points of contact between the parasite and the host membrane. However, additional experiments are required to specifically conclude on the global versus local effects of KMP-11 during the process of promastigote infection.

# Methods

**Reagents and tools table**

| Reagent/resource | Reference or source | Identifier or catalog number |
| --- | --- | --- |
| **Experimental models** | | |
| BALB/c (*M. musculus*) | ICMR-NIN | N/A |
| RAW 264.7 (*M. musculus*) | ATCC | TIB-71 |
| **Recombinant DNA** | | |
| pCMV-LIC | PMID: 20816743 | N/A |
| pET28a (+) | Addgene | 69864-3 |
| pLdCN2 | PMID: 32221923 | N/A |
| pSPBle | Zhang and Matlashewski, 2015 | N/A |
| pXG-EGFP plasmid | B Kleczka, 2006 | N/A |
| **Oligonucleotides or other sequence-based reagents** | | |
| Site-directed mutagenesis primer | Quik-change Site-Directed Mutagenesis kit (Agilent) | 200518 |
| **Peptides** | | |
| MATTYEEFSAKLDRLDQEF, MATTYEEFSAKLDLRDQEF and MATTYEEFSALKLDRFDQE | Abgenex (KRIC distributor, USA) | N/A |
| **Chemicals, enzymes, and other reagents** | | |
| Dipalmitoylphosphatidylcholine (DPPC) | Avanti Polar Lipids Inc. (Alabaster, AL, USA) | 850355 |
| Dehydroergosterol (DHE) | Avanti Polar Lipids Inc. (Alabaster, AL, USA) | 810253 |
| NBD-C12 PC | Invitrogen | 810131 |
| Laurdan | Invitrogen | D250 |
| NBD cholesterol (NBD Cholesterol (22-(*N*-(7-Nitrobenz-2-Oxa-1,3-Diazol-4-yl)Amino)-23,24-Bisnor-5-Cholen-3β-Ol) | Invitrogen | N1148 |
| FITC | Invitrogen | F1906 |
| DiIC18(5) solid (1,1'-Dioctadecyl-3,3,3',3'-Tetramethylindocarbocyanine, 4-Chlorobenzene Sulfonate Salt) (DiD) | | D7757 |
| DPH (Diphenylhexatriene) | Invitrogen | 60021 |
| Alexa fluor-488 maleimide | Invitrogen | A10254 |
| Cholera Toxin B- FITC conjugate | Sigma-Aldrich | C1655 |
| Paraformaldehyde | Himedia | TC703 |
| DAPI | SRL | 28488 |
| Giemsa stain | Himedia | S011 |
| Methyl-β-cyclodextrin | Sigma-Aldrich | 332615-25MG |
| HBSS | Gibco | 14025092 |
| Penicillin-Streptomycin | Invitrogen | 15140122 |
| RPMI 1640 | Invitrogen | 31800022 |
| | Sigma-Aldrich | R8758 |

| Reagent/resource | Reference or source | Identifier or catalog number |
|---|---|---|
| M199 | Sigma | M0393 |
| FBS | Gibco | 16000044 |
| Amplex-Red assay kit | Invitrogen | A22188 |
| Pierce BCA Protein Assay Kit | Thermo Scientific | 23225 |
| IL-2 ELISA kit | BD biosciences | 55148 |
| IL-10 ELISA kit | R&D Systems | DY417 |
| IL-12 ELISA kit | R&D Systems | DY499 |
| G418 | Gibco | 10131027 |
| Bleomycin | Sigma-Aldrich | B5507 |
| Sodium Lauryl Sulphate | SRL | 32096 |
| Acrylamide (40% solution) | Invitrogen | HC2040 |
| APS | SRL | 65553 |
| TEMED | Invitrogen | HC2006 |
| **Antibodies** | | |
| KMP-11 Mouse mAb | Invitrogen | MA1-83818 |
| CD11b/ITGAM Rabbit mAb | Cell signaling Technology | 17800 |
| β-Actin Mouse mAb | Cell signaling Technology | 12262 |
| BiP Rabbit mAb | Cell signaling Technology | 3177 |
| Alexa Fluor 488 goat anti-mouse IgG | Invitrogen | A-11001 |
| Alexa Fluor 594 goat anti-rabbit IgG | Invitrogen | A-11037 |
| Alexa Fluor 488 goat anti-rabbit IgG | Invitrogen | A-11008 |
| Alexa Fluor 594 goat anti-rabbit IgG | Invitrogen | A-11005 |
| Goat anti-Rabbit IgG-HRP | SRL | 88495 |
| Goat anti-Mouse IgG-HRP | SRL | 57404 |
| Na, K-ATPase Rabbit pAb | Cell signaling Technology | 3010 |
| GAPDH Mouse mAb | Cell signaling Technology | 97166 |
| **Softwares** | | |
| GraphPad Prism 9.0 | | N/A |
| Adobe Photoshop | | N/A |
| ImageJ | https://imagej.nih.gov/ij/ | N/A |
| Endnote | | N/A |
| FlowJo v10 | http://www.flowjo.com | N/A |
| iGemdock | http://gemdock.life.nctu.edu.tw | N/A |
| BioRender | Scientific Image and Illustration Software \| BioRender | N/A |
| **Others** | | |
| Olympus VELITA (2000 × 2000) charge-coupled device camera | Olympus | N/A |
| Nitrocellulose membrane | BioRad | 1620115 |

## Structure modeling of KMP-11

We have discussed in detail the structure modeling of KMP-11 in an earlier paper (Sharma et al, 2013). Briefly, the sequence analysis of KMP-11 using NCBI protein−protein BLAST does not show any close homolog with the available solved structures. However, a profile-based search performed using PSI BLAST indicated the presence of a few remote homologs, whose crystal structures are available. As the sequence identities of the remote homologs are low, we have used a composite approach using iterative threading assembly refinement (ITASSER) of the Zhanglab server (Yang and Zhang, 2015), which combines various techniques, such as threading, ab-initio modeling, and atomic level structure refinement methods. From the I-TASSER analyses, we have chosen a model structure that provides the best confidence score.

## Computational docking and alignment

We employed computational docking using iGemdock (http://gemdock.life.nctu.edu.tw) software. Using this we determined the binding energies and the most stable docked structures of cholesterol bound proteins of different origins. The core of iGEMDOCK software is GEMDOCK, which is a robust and well-developed tool. The predicted poses generated from the GEMDOCK are able to be directly visualized by a molecular visualization tool and analyzed by post-analysis tools. iGEMDOCK also provides the post-analysis tools using k-means and hierarchical clustering methods using docked poses (i.e. protein-ligand interactions) and compound properties (i.e. atomic compositions). Using iGEMDOCK, we evaluated the best posed structures of KMP-11 (using the model structure as input) individually with CHOL, ERG and DPPC lipid.

## OPM (orientations of proteins in membranes)

To gain an insight as to how the KMP-11 and peptides interact with the membrane, we used computational approaches. Protein orientations in membranes were theoretically calculated by minimizing a protein's transfer energy from water to a planar slab that serves as a crude approximation of the membrane hydrocarbon core. For WT KMP-11, we referred to the model structure obtained from I TASSER using Zhang Lab server. The membrane binding propensity was calculated by submitting the coordinate information of the protein forms to the OPM server. A protein was considered as a rigid body that freely floats in the planar hydrocarbon core of a lipid bilayer. Accessible surface area is calculated using the subroutine SOLVA from NACCESS with radii of Chothia and without hydrogen. In OPM, solvation parameters are derived specifically for lipid bilayers and normalized by the effective concentration of water, which changes gradually along the bilayer normal in a relatively narrow region between the lipid head group regions and the hydrocarbon core.

## Protein expression, purifications and mutagenesis methods

### Site directed mutagenesis
Wild type KMP-11 gene cloned previously in pCMV-LIC vector was further sub-cloned in pET28a vector between NcoI and BamHI

using the following set of primers: Fwd: 5'- CCCCATGGCCAC-CACGTACGAGGAG-3' and Rev: 5'- AAGGATCCCTCCTGAT-GATGATGATGATGCTTGGAACGGGTACTGCGCAGC-3'. Site-directed mutagenesis was performed with the help of Quik-change Site-Directed Mutagenesis kit (Stratagene, Agilent Technologies,USA) to generate several single tryptophan (W) mutants, using the wild type KMP-11 construct as a template. All the mutations were confirmed by DNA sequencing. All the primers required for the generation of eight single tryptophan mutant of KMP-11 have been mentioned in extended view files (Appendix Table S7).

### Purification of the WT and mutants of KMP-11 and protein labeling

Amino terminal His (His x 6) tagged recombinant KMP-11 constructs (both WT and mutants) were expressed and purified using Ni-NTA affinity chromatography. A cell pellet from two liter bacterial culture was re-suspended in 75 mL buffer (20 mM sodium phosphate buffer at pH 7.4 and 1 mM PMSF) and sonicated by a probe sonicator for 20 min (35% amplitude). Subsequently, the cell lysate was subjected to $100,000 \times g$ centrifugation for 45 min and the resulting supernatant was loaded on a Ni-NTA column. The column was washed with 50 mL sodium phosphate buffer containing 25 mM imidazole and bound proteins were eluted with sodium phosphate buffer containing 300 mM imidazole. The collected fractions containing the protein were dialyzed using a 20 mM to remove excess imidazole. Next, the dialysed protein sample was passed through endotoxin removal column (Pierce, Thermoscientific, USA) and further purified by gel filtration chromatography on a Tricorn 10/300 superose 6 column (GE Healthcare, USA). The purified protein fractions were confirmed using 15% sodium dodecyl sulfate-polyacrylamide gel electrophoresis and MALDI (Matrix Assisted Laser Desorption/Ionization) experiment. The purified protein concentration was determined using the BCA Protein Assay Kit (Pierce, Thermo Scientific, USA).

After purification, some aliquots of the purified r-KMP-11 (S38C mutant of KMP-11) were mixed with Alexa-488 maleimide dye and incubated at room temperature in a dark place for 4 h. Subsequently, the mixed sample was passed through Ni NTA column to separate free dye. Next, the eluted sample was passed through superdex G200 column to remove rest of the free dye. The labeling percentage of our labeled r-KMP-11 was found to be 86%.

### Peptide synthesis

Our three designed peptides (MATTYEEFSAKLDRLDQEF, MAT-TYEEFSAKLDLRDQEF and MATTYEEFSALKLDRFDQE) containing different hydrophobic moments were synthesized and supplied by Abgenex (KRIC distributor, USA). Peptides are ~95% pure.

## Biophysical methods

### Measurement of fluorescence anisotropy (FA)

Membrane fluidity was measured using the method of Shinitzky and Inbar (Shinitzky and Henkart, 1979). Briefly, the fluorescence probe, DPH, was dissolved in tetrahydrofuran at 2 mM concentration. To 10 ml of rapidly stirring PBS (pH 7.2), 2 mM DPH solution was added. For labeling, $10^6$ cells (different sets of MΦ cells were treated with 50 μM rKMP-11or LD parasites or 1.5 mM liposomal CHOL or 1 mM MβCD or 1 mM staurosporine and subsequently incubated for 12 h and then washed) were mixed with an equal volume of DPH in PBS (Cf 1 μM) and incubated for 2 h at 37 °C.

Thereafter the cells were washed thrice and re-suspended in PBS. The DPH probe bound to the membrane of the cell was excited at 365 nm and the intensity of emission was recorded at 430 nm using a spectro-fluorometer. The FA value was calculated using the equation:

$$FA = \frac{I_{II} - I}{I_{II} + 2I} \qquad (2)$$

Where, $I_{II}$ and $I$ are the fluorescence intensities oriented, respectively, parallel, and perpendicular to the direction of polarization of the excited light.

### Laurdan generalized polarization (GP) measurement in cell and model membrane

Laurdan fluorescence was measured using a PTI fluorescence spectrophotometer. For labeling, $10^6$ cells (different sets of raw 264.7 MΦ cells were treated with 50 μM r-KMP-11 or LD parasites or 1.5 mM liposomal CHOL or 1 mM MβCD or 1 mM staurosporine and subsequently incubated for 12 h and then washed) were mixed with an equal volume of Laurdan (dissolved in DMSO) to attain 5 μM working concentration and incubated at RT for 20 min. Thereafter, the cells were washed thrice and re-suspended in PBS.

Cellular aliquots and model membranes of DPPC (labeled with Laurdan) in presence of different protein concentrations were excited at 340 nm. Emission spectra were recorded in the range of 400–550 nm at a speed of 60 nm/min. Generalized polarization parameters defined as

$$GP = \frac{(I_{440} - I_{490})}{(I_{440} + I_{490})} \qquad (3)$$

were obtained as a function of temperature. Here, $I_{440}$ and $I_{490}$ stand for the intensities at the wavelength at 440 nm and 490 nm, respectively. Phase transition midpoints (Tm) were determined by nonlinear least-squares fitting using a sigmoidal fitting function (Veatch and Keller, 2003).

### Preparation of small unilamellar vesicles (SUVs) from DPPC

An appropriate amount of DPPC lipid in chloroform (concentration of the stock solution is 25 mg mL$^{-1}$) was transferred to a 10 ml glass bottle. The organic solvent was removed by gently passing dry nitrogen gas. The sample was then placed in a desiccator connected to a vacuum pump for a couple of hours to remove traces of the leftover solvent. A required volume of 20 mM sodium phosphate buffer at pH 7.4 was added to the dried lipid film so that the final desired concentration (10 mM) was obtained. The lipid-film with the buffer was kept overnight at 4 °C to ensure efficient hydration of the phospholipid heads. Vortexing of the hydrated lipid film for about 30 min produced multilamellar vesicles (MLVs). Long vortexing was occasionally required to make uniform lipid mixtures. This MLV was used for optical clearance assay. For preparing the small unilamellar vesicles, MLVs was sonicated using a probe sonicator at amplitude 45% for 30 min and after that sample was centrifuged at 5000 rpm to sediment the tungsten artifacts and finally it was filtered by 0.22 μm filter unit. Size of the small unilamellar vesicles was measured by DLS and the average diameter was found to be ~70 nm.

### Binding assay using fluorescence spectroscopy

We have used a fluorometric assay for studying the binding of WT KMP-11 and its tryptophan mutants with SUVs composed of DPPC. All samples were prepared in a 20 mM sodium phosphate buffer at pH 7.4. A set of samples were prepared using a 1 mM concentration of uniformly synthesized lipid vesicles. In each sample vial, 0.5 wt% of membrane-specific DiD dye was added, and the samples were kept at 37 °C for overnight incubation. Subsequently, the required amount of protein was added into the vials by maintaining the lipid/protein molar ratio between 1:0 and 50:1. The samples were then incubated at room temperature (25 °C) for 2 h. The steady-state fluorescence emission spectra of the dye were recorded at an excitation wavelength of 600 nm. The peak intensity values at 683 nm for DPPC were plotted with protein concentration. The data have been fitted using the sigmoidal Hill equation, as follows:

$$F = F_0 + \frac{(F_e - F_0)x^n}{(x^n + k^n)} \qquad (4)$$

Where, $F$ and $F_0$ refer to the fluorescence intensities of DiD in the presence and absence of protein, respectively. $F_e$ denotes the minimum intensity in the presence of a higher concentration of protein, and $K$ is the equilibrium dissociation coefficient of the lipid–protein complex. $n$ is the Hill coefficient, which measures the co-operativity of binding, and $x$ is the concentration of the protein. PTI fluorimeter (Photon Technology International, Inc) and a cuvette with a 1 cm path length were used for the fluorescence measurements. To trace the mechanism of DiD fluorescence decrease due to KMP-11 binding, we performed time resolved study of DiD-labeled DPPC SUVs in absence and presence of KMP-11 which showed decrease in fluorescence lifetime after protein membrane binding (Appendix Table S1). This quenching may occur because of the expulsion of the dye molecules as a result of protein binding. Alternatively, the hydration dynamic at the membrane−solvent interface may get altered due to the lipid–protein interaction modulating the relaxation behavior of DiD. It is also known that DiD diffuses laterally within the membrane and therefore the change in the translational diffusion due to protein adsorption may lead to dynamic quenching.

### Circular dichroism measurement

Near and Far-UV CD spectra of WT KMP-11 in absence and presence of DPPC SUVs were recorded using a JASCO J720 spectro-polarimeter (Japan Spectroscopic Ltd, Ishikawa-cho-hachioji-shi, Japan). Far-UV CD measurements (between 200 and 250 nm) were performed using a cuvette of 1 mm path length. A protein concentration of 10 μM was used for the CD measurements. The scan speed was 50 nm min⁻¹, with a response time of 2 s. The bandwidth was set at 1 nm. Three CD spectra were recorded in continuous mode and averaged. Typical protein–lipid ratio was maintained at 1:200.

### Phase state analysis by steady state and time resolved fluorescence

Phase state of DPPC bilayer was monitored by using a PTI fluorimeter for the measurement of steady state fluorescence at different temperatures in absence and presence of KMP-11 protein ratio-metric study was also done. Red Edge Excitation shift of NBD-labeled DPPC in absence and presence of protein was also determined using steady state fluorescence by applying different excitation wavelengths (465, 475, 485, 495, 505, and 515 nm).

Trans-bilayer movement (Kol et al, 2004) of symmetrically labeled NBD-DPPC bilayer was measured from the fluorescence quenching by using dithionite as reducing agent of NBD fluorophore. This fluorescence decline corresponds to the dithionite-mediated reduction of analogs localized in the outer membrane leaflet. Subsequently, fluorescence intensity adopted a new plateau representing the analogs on the inner leaflet. The very slow decrease of fluorescence indicates that the dithionite permeation across the membrane and/or the analog flip-flop were negligible under these conditions within the time scale of the experiment. From the fluorescence intensities before ($F_0$) and after ($F_R$) addition of dithionite, the relative fraction of the analog on the outer leaflet ($P_o$) was estimated using the following equation:

$$P_0(\%) = \left(1 - \frac{F_R}{F_0}\right) \times 100\% \qquad (5)$$

Upon addition of Triton X-100 all analogs became accessible to dithionite, resulting in a complete loss of fluorescence.

### SAXS experiment

Samples (DPPC multilamellar vesicles in absence and presence of r-KMP-11 at different L/P ratio) for small-angle X-ray scattering (SAXS) studies were taken in 1.0 mm glass capillaries (Hampton Research) and were flame sealed. SAXS data were collected using a HECUS S3-Micro system, equipped with a 1D position sensitive detector. Exposure times varied from 60 to 90 min. Data were collected over a range of the magnitude of the scattering vector (q) from 0.17 to 3.7 nm⁻¹. Error bar in the measurement of the lamellar periodicity (d) varied from 1.70 nm at very low q to 0.02 nm at very high q.

### FTIR spectroscopy

FTIR experiment was performed for determination of the morphological changes of DPPC bilayer in presence and absence of KMP-11 using Bruker Tensor 27 FTIR spectrometer (Markham,ON,Canada) in liquid mode. For FTIR measurements, we considered the frequency range from 1280 cm⁻¹ to 1440 cm⁻¹ which corresponds to the methylene wagging band ($CH_2$) of DPPC. In addition, we considered the spectral position at 1367 cm⁻¹, which indicates the kink and gauche rotamer conformation signature of lipids. The percentages of different conformers in absence and presence of KMP-11 and its other tryptophan mutants were determined by using fitting the peaks using Gaussian distribution functions.

### Tryptophan quenching experiment

Steady state fluorescence spectroscopy and acrylamide quenching measurements in free and in membrane bound conditions were carried out using a PTI fluorimeter (Photon Technology International, USA). A cuvette with 1 cm path length was used for the fluorescence measurements. For the tryptophan fluorescence quenching experiments, an excitation wavelength of 295 nm was used to eliminate the contributions from the tyrosine fluorescence. Fluorescence data were recorded using a step size of 1 nm and an integration time of 1 s. Excitation and emission slits were kept at 5 nm in each case. Emission spectra between 305 nm and 450 nm were recorded in triplicate for each experiment. Typical protein concentration of 10 μM was used for each quenching experiment and 1:200 protein–lipid molar ratio was maintained. The protein solutions were incubated at room temperature for 1 h and then titrated using a stock of 10 M acrylamide. Necessary background corrections were made for each experiment.

### Acrylamide quenching data analysis

Assuming I & Io represent tryptophan fluorescence intensity of the proteins in the presence and -absence of acrylamide concentration [Q], Stern–Volmer Equation can be represented as follows:

$$\frac{I_0}{I} = 1 + K_{sv}[Q] \tag{6}$$

$K_{sv}$ is the Stern–Volmer constant, which can be determined from the slope of the linear plot of $I_0/I$ vs. acrylamide concentrations [Q].

## Energy transfer measurement

All experiments were done using small unilamellar vesicles (SUVs) containing 1280 nmol of DPPC with required amount of DHE (2.5 mol %). Tryptophan fluorescence of the eight mutants (Y5W, F19W, F30W, Y48W, F51W, F62W, and Y89W) was monitored using steady state fluorescence technique. The tryptophan residue of the mutant served as the donor whereas DHE was used as the acceptor. The energy transfer efficiencies were calculated using the equation (Lakowicz, 1999):

$$E = \left(1 - \frac{F}{F_0}\right) \tag{7}$$

Where $E$ is the efficiency of energy transfer and $F$ and $F_o$ are fluorescence intensities of the donor (tryptophan of KMP-11 mutants) in the presence and absence of the acceptor (DHE), respectively. Using FRET, the distance "r" between tryptophan residues of different tryptophan residues and membrane bound DHE could be calculated by the equation.

$$E = (1 - F/F_0) = \frac{R_0^6}{(R_0^6 + r_0^6)} \tag{8}$$

$R_0$ (Forster radius), measured in Å unit, is the critical distance when the efficiency of transfer is 50%; $r_0$ is the distance between donor and acceptor.

$$R_0^6 = 8.79 \times 10^{-25} \chi^2 \eta^{-4} \varphi J \tag{9}$$

$\chi^2$ is the orientation factor related to the geometry of the donor and acceptor of dipoles and 2 /3 for random orientation as in fluid solution; $n$ is the average refractive index of medium in the wavelength range where spectral overlap is significant; $\varphi$ is the fluorescence quantum yield of the donor; $J$ is the effect of the spectral overlap between the emission spectrum of the donor and the absorption spectrum of the acceptor, which can be calculated by the equation:

$$J = \int_0^\infty F(\lambda)\epsilon(\lambda)\lambda^4 d\lambda / \int_0^\infty F(\lambda)d\lambda \tag{10}$$

where, $F(\lambda)$ is the corrected fluorescence intensity of the donor in the wavelength range from $\lambda$ to $\lambda + \Delta\lambda$; $\epsilon(\lambda)$ is the extinction coefficient of the acceptor at $\lambda$.

## Fluorescence correlation spectroscopy (FCS) experiments and data analysis

FCS experiments were carried out using a dual channel ISS Alba V system equipped with a ×60 water-immersion objective (NA 1.2).

Samples were excited with an argon laser at 488 nm. All protein data were normalized using the $\tau_D$ value obtained with the free dye (Alexa488) which was measured under identical conditions. For a single-component system, diffusion time ($\tau_D$) of a fluorophore and the average number of particles ($N$) in the observation volume can be calculated by fitting the correlation function [$G(\tau)$] to:

$$G(\tau) = 1 + \left(\frac{1}{N\left(1 + \frac{\tau}{\tau_D}\right)}\right) \frac{1}{\sqrt{1 + S^2 \frac{\tau}{\tau_D}}} \tag{11}$$

where, $S$ is the structure parameter, which is the depth-to-diameter ratio. The characteristic diffusion coefficient ($D$) of the molecule can be calculated from $\tau_D$ using:

$$\tau_D = \frac{\omega^2}{4D} \tag{12}$$

where, $\omega$ is the radius of the observation volume, which can be obtained by measuring the $\tau_D$ of a fluorophore with known $D$ value. The value of hydrodynamic radius ($r_H$) of a labelled molecule can be calculated from $D$ using the Stokes–Einstein equation:

$$D = \frac{kT}{6\pi\eta r_H} \tag{13}$$

where, $k$ is the Boltzmann constant, $T$ is the temperature and $\eta$ corresponds to the viscosity of the solution.

## FRET-based CHOL transport assay

The final lipid concentration in the reaction was 1 mM, with donor and acceptor liposomes added at a 1:1 ratio (only acceptor liposomes contain 2.5% DAUDA). Reactions were initiated by the addition of protein to a final concentration of 5 and 20 mM. The fluorescence intensity of DAUDA (i.e. FRET signals), resulting from FRET between DAUDA and DHE (excited at 310 nm), was monitored at 525 nm over 180 s in a PTI fluorimeter. The values of blank solution (buffer only) were subtracted from all the values from each time point, and data were presented by setting the first value to zero at $t = 0$. In some experiments, data were expressed as the number of DHE molecules transferred using the calibration curve. For the generation of the calibration curve, FRET signals were measured for the liposomes containing 0%, 5% (1.25 nmol), 10% (2.5 nmol) or 15% (3.75 nmol) DHE and 2.5% DAUDA (0.5 mM lipids in total: compositions of the liposomes). The mean of FRET signals at $t = 0$ from three replicates were plotted against the DHE mole number in liposomes. Then, the mole number of the transferred DHE from the donor to acceptor liposomes in in vitro DHE transfer assay was obtained using the following formula: $y = 7422.40 + 44091.24x$ (derived from the linear fit of the calibration curve). To obtain $x$ (the amount of transferred DHE in nmol shown in the $y$ axis of), the FRET values from each time point of the in vitro lipid transfer assay were substituted for the $y$ of the equation. Transfer rates were obtained from the fit of the growth curves using origin pro 8.5.

## Dynamic light scattering

The hydrodynamic radius of the vesicles was measured from the DLS experiments. In the DLS experiments, the intensity fluctuations of scattered light were measured and the intensity

autocorrelation function was fit to the exponential decay function to obtain the values of the diffusion constant. The Einstein–Stokes relation was used to calculate the hydrodynamic radius of the vesicle. For size measurements, a 100 µM concentration of DOPC-CHOL SUVs and DOPC-ERG SUVs were used and typical r-KMP-11 concentration was taken 2 µM.

## Substrate cleaning and microchannel assembly for TIRFM

Two holes (~0.5 mm) were drilled on opposite sides of a glass slide (Blue Star) along the short axis. Cover slips (0.13 mm thickness, VWR) and slides were cleaned in piranha solution (3:1 v/v $H_2SO_4$) for 45 min in a fume hood. Then, they were rinsed with copious amounts of MilliQ water and dried under a stream of $N_2$ gas. The coverslip was further cleaned by exposure to plasma (Harrick) for 4 min. On either side of the pair of holes, strips of double-sided tape (3 M, Scotch, thickness ~70 µm, width ~0.6 mm) were placed parallel to the short axis. The cleaned coverslip was placed on the taped slide to form a microchannel with the drilled holes serving as inlets and outlets for buffer exchange.

## PEG-supported lipid bilayer

PEG-SLBs were synthesized as described earlier (Sathyanarayana et al, 2018) with some modifications. PEG-SUVs were diluted 1:1 in PBS (containing 3 mM $CaCl_2$). In total, 20 µL of this vesicle solution was introduced into the chamber using a micropipette. The channel assembly was incubated at 37 °C for 1 h in a humidifying chamber. The channels were washed with PBS to remove unfused vesicles.

## Fluorescence microscope setup

The single-particle experiments were performed on an inverted microscope (Olympus IX81). A laser (Sapphire; Coherent) was used to excite the fluorescence labeled molecules on a custom-built objective-type total internal reflection microscope. A combination of 25.4- and 300-mm biconvex lenses (Thor Laboratories) were used to expand the laser beam before a lens of focal length 150 mm was used to focus the (16-mW) laser beam on its back focal plane (BFP) of the objective (UAPON 100× OTIRF; Olympus). The laser spot at the BFP was translated away from the optical axis to achieve total internal reflection. Fluorescence emission from 80 µm × 40 µm area was collected by the objective and passed through a dichroic mirror (FF545/650-Di01-25 × 36; Semrock) and a long-pass filter (BLP02-561R-23.3-D; Semrock) before detection on an electron multiplying charge-coupled device (Andorixon Ultra 897). Shutter (LS6; Vincent Associates) was used to control the laser illumination time.

## TEM imaging

Purified native KMP-11 in the presence of DPPC ERG (30%) SUVs were visualized by negative staining electron microscopy (EM) to analyze homogeneity and particle distribution. The samples were prepared by conventional negative staining methods. A carbon-coated copper grid (EM grid, 300 mesh; TedPella) was glow-discharged for 30 s at 20 mA. The purified protein was dialyzed without glycerol buffer for negative staining analysis [(50 mM tris

(pH 8.0) (HiMedia Laboratories, Mumbai, India) and 150 Mm NaCl (SISCO Research Laboratories)]. A total of 3.5 µl of the sample (0.1 mg/ml) was added to the glow-discharged (GloQube glow dis-chargesystem,Quorum) carbon-coated copper grid for 30 s. The extra sample was blotted out. Negative staining was performed using 1% uranyl acetate (98% uranyl acetate; ACS Reagent, Polysciences Inc. Warrington, PA, USA) solution for 20 s. The grid was air-dried. The negatively stained sample for native KMP-11 was visualized at room temperature using Tecnai T12 electron microscope equipped with a LaB6 filament operated at 120 kV, and images were recorded using a side-mounted Olympus VELITA (2000 × 2000) charge-coupled device camera at a magnification of ×220,000 (2.54 Å per pixel).

## AFM studies

A 5 µL aliquot was taken from the diluted sample (KMP-11 + DPPC-ERG SUVs) and deposited on freshly cleaved mica for 10 min. The typical protein concentration was 500 nM. After removing the excess liquid, the protein samples were rinsed with MilliQ water and then dried with a stream of nitrogen. Images were acquired at ambient temperature using a Bioscope Catalyst AFM (Bruker Corporation, Billerica, MA) with silicon probes. The standard tapping mode was used to image the morphology of oligomers. The nominal spring constant of the cantilever was kept at 20–80 N/m. The spring constant was calibrated by a thermal tuning method. A standard scan rate of 0.5 Hz with 512 samples per line was used for imaging the samples. A single third-order flattening of height images with a low pass filter was done followed by section analysis to determine the dimensions of oligomers.

## Photobleaching step analysis

PEG cushioned SLB was prepared using DPPC-ERG (30%) lipid composition. Next 20 µl of 50 nM alexa-488 maleimide labeled KMP-11 (S38C mutant) was added on the SLB surface. After 30 min of incubation in a humidifying chamber, TIRFM imaging was done by irradiating the particles on SLB. The oligomeric status of KMP-11 on parasite mimicking DPPC-ERG membrane was estimated through manual step counts from 235 number of spots.

## Cell biology methods

### Cell culture, parasite culture and infection

Murine macrophage cell line RAW264.7 cells were maintained at 37 °C and 5% $CO_2$ in Roswell Park Memorial Institute medium (RPMI-1640, Sigma-Aldrich, USA) containing 10% heat-inactivated fetal bovine serum (HI-FBS) (Thermo Fisher Scientific, USA), penicillin–streptomycin 1% (Thermo Fisher Scientific, USA). After 70-80% confluency the cells were scraped off and harvested at a required concentration to allow them to re-equilibrate a day before the start of each experiment. LD parasite line MHOM/IN/1983/AG83 maintained in mice was used for this study. Amastigotes were obtained from the spleen of infected mice and they were subsequently transformed into promastigotes and maintained in MI199 medium with 10% FBS. Peritoneal exudates cells conveniently named PEC were harvested from BALB/c mice as described elsewhere (Mukhopadhyay et al, 2011) and maintained at 37 °C and 5% $CO_2$ in RPMI-1640 medium (Sigma-Aldrich, USA)

containing 10% heat-inactivated fetal bovine serum, penicillin–streptomycin 1%. After 48 h of resting stage, cells were taken for subsequent experiments.

For infection, RAW264.7 were plated and washed after 6 h to get adhered and incubated overnight. Infection was performed with 1:10 (macrophage: parasite) MOI (multiplicity of infection) LD strain AG83 (MHOM/IN/1983/AG83) for 4 h. After these, cells were washed with PBS to remove the extracellular parasites and kept for another 12 h (Mukherjee et al, 2013). For attachment and invasion assay, equal numbers of metacyclic LD promastigotes were sorted from WT_LD, LD_KMP-11_KO, and LD_KMP-11_KO /KMP-11 complemented cultures using CytoFLEX Flow Sorter (Beckman Coulter) in a 15 ml tube and MΦ infection was performed as described previously (Saraiva et al, 2005). Infection was performed for 4 h, and cells were washed thoroughly to remove extracellular LD. Attached/Invaded LD per 100 MΦs were determined by Giemsa staining.

## IL-2, IL-10 and IL-12 ELISA

RAW264.7 cells were cultured overnight in a 35 mm tissue culture plate ($1 \times 10^6$ cells/ml) and LD infection was performed as already described. For IL-2, the cells were co-cultured with LMR7.5T cell hybridoma specific for *Leishmania* antigen for 12 h in presence of 10 µg/ml LACK antigen along with r-KMP-11 (low dose 5 µM, medium dose 30 µM and high dose 50 µM), respectively (Roy et al, 2014). After incubation cell free supernatant was collected and subjected to sandwich ELISA using IL-2 ELISA kit (BD Biosciences, CA, USA) as per manufacturer instruction. IL-10 and IL-12 were measured in the macrophage supernatant infected with LD or treated with r-KMP-11 or LPS or left untreated, using IL-10 and IL-12 ELISA kit, respectively (BD Biosciences, CA, USA). LPS (1 mg/ml) was used as control for the experiment.

## Parasite count measurements

RAW264.7 cells were cultured (10,000 cells/well) in chamber slides with 200 µl/well RPMI-1640 media supplemented with 10% FBS overnight. After overnight incubation, cells were treated with r-KMP-11 (50 µM) and subsequently infected with *Leishmania donovani* (MHOM/IN/1983/AG83) (1:10 macrophage to parasite ratio) for 4 h, washed to remove extracellular parasites and stained with Giemsa staining solution and the number of intracellular amastigotes were calculated and represented per 100 macrophages.

## Confocal microscopy to detect membrane lipid raft

RAW264.7 cells were cultured ($1 \times 10^4$) on coverslip in RPI1640 medium supplemented with 10% FCS overnight. After incubation cells were treated with r-KMP11 and infected with L. donovani parasite for 12 h (4 h wash+12 h incubation) and washed with 1×PBS. The cells were then processed with lipid raft markers using cholera toxin B-FITC following manufacturer's instruction (C1655, Sigma). Briefly, the adhered cells were treated with 4 µg/ml of FITC conjugated cholera toxin B subunit and incubated for 30 min in dark. After incubation, the cells were washed with 1×PBS thrice followed by fixation with 4% paraformaldehyde for 15 min and then mounted in a vectashield mounting media with DAPI and subjected to observation under confocal microscope.

It should be noted that we followed the similar protocol of a published paper (Chakraborty et al, 2005) for liposomal CHOL treatment. In brief, 12 h treated cells (LD/rKMP-11) were washed and subsequently liposomal CHOL (1.5 mM) was added and incubated at 37 °C for 12 h. The cells were then processed with lipid raft marker using cholera toxin B-FITC following manufacturer's instruction and then mounted in a vectashield mounting media with DAPI and subjected to observation under confocal microscope.

## Liposomal-CHOL preparation and delivery

Liposomes were prepared with CHOL and phosphatidyl ethanolamine at a molar ratio of 1.5:1. A thin dry film of lipids (5.8 mg CHOL and 8.0 mg phosphatidylethanolamine) was dispersed in 1 ml of RPMI-1640 and sonicated at 4 °C three times, 1 min each, at maximum output. To understand the effect of liposomal CHOL on distorted lipid rafts we follow previously reported protocol of liposomal CHOL delivery (Chakraborty et al, 2005). In brief, $10^4$ cells (pretreated with WT_LD or r-KMP-11)were incubated with liposomes for 12 h at 37 °C. The cells with altered fluidity were then washed three times in serum-free RPMI-1640 medium and finally re-suspended in 10% FBS containing RPMI-1640.

## Apoptosis analysis by flow cytometry

Cell apoptosis induced by r-KMP-11 and WT_LD was assessed using an Annexin V-FITC/PI assay. MΦs cells were plated in 6-well plates ($0.5 \times 10^5$ cells/well, 2 mL) and exposed to r-KMP-11 protein (50 µM) or WT_LD for 12 h. Post-treatment, cells were detached with 0.25% trypsin, rinsed in cold phosphate-buffered saline (PBS), and centrifuged at $300 \times g$ and 4 °C for 3 min. The supernatant was discarded, and the cells were reconstituted in 400 µL binding buffer from BD Biosciences, comprising 0.1 M HEPES/NaOH (pH 7.4), 1.4 M NaCl, and 25 µM $CaCl_2$. Subsequently, 5 µL of Annexin V-FITC and 5 µL of PI were gently added and mixed with the cells in the dark at room temperature. Following a 15-min incubation, cell apoptosis was assessed within 1 h of staining using flow cytometry (BD Biosciences). FlowJo software was utilized for apoptosis data analysis.

## KMP-11 knock-out generation in *Leishmania donovani*

The pLdCN2 plasmid is used to co-express Cas9 and two gRNAs in *Leishmania*. The pSPBle plasmid is used as a template to make donor DNA containing the bleomycin-resistant gene for selection of edited clones as described previously. Eukaryotic Pathogen CRISPR guide RNA Design Tool (EuPaGDT) (http:// grna.ctegd.uga.edu/) were used for Ld_KMP-11 specific gRNA (gRNA 1a_F_KMP-11-5'-TTGTGCAGAACGCCAAGTTCTTTG-3',gRNA 1b_R_KMP-11- 5'-AAACCAAAGAACTTGGCGTTCTG C-3', gRNA 2a_F_KMP-11- 5'-GGTGGAAGTTCGAGCGCAT-GATCAGT-3', gRNA 2b_R_KMP-11- 5'-TAAAACTGATCAT GCGCTCGAACTTC-3') generation. gRNAs were cloned into the pLdCN2 by BbsI-mediated digestion. Bleomycin cassette (KMP-11-Ble-KMP-11) inserted into the with KMP-11 genomic locus using 25 nt of homology sequence flanking the Cas9 cleavage site within KMP-11 (KMP-11_Ble Forward primer: 5'-GAAGTTCGAG CGCATGATCAAGGATCTTCATCGGATCGGGTAC-3', KMP-11_Ble Reverse primer: 5'-TTCTTGTTGAACTTCTCTGTGTGT

TTCAGTCCTGCTCCTCGGCCA-3'). Transfection was done as described previously with slight modification (Yagoubat et al, 2020). Briefly, we use a cytomix buffer with 10 mM ATP, GSH. KMP-11-Ble-KMP-11 cassette was co-transfected with 10 µg of cloned pLDCN2 plasmid to the *Leishmania* cells. After 24 h of transfection, G418 and Bleomycin (Ble) were used to select the KO parasites. LD_KMP-11_KO clones were isolated by performing serial dilution in 96-well plate with G418 (50 µg/ml) and Ble (50 µg/ml) selection as described previously. LD_KMP-11_KO clones were confirmed by checking for bleomycin integration and by western blot analysis by blotting against anti-KMP-11 antibody.

## Complementation of KMP-11 KO LD lines

To confirm the phenotypic changes are due to absence of KMP-11, complementation with wild-type KMP-11 was achieved by episomal expression using pXG-EGFP plasmid. Entire *KMP-11*(279 nt) were fused with EGFP by Gibson assembly using primers (KMP-11_fwd ACGAGCTGTACAAGATCTGCATGGCCACCACGTACGAG, KMP-11_revTGGATCACAATTGGATCCGCCTACTTGCACATGTTCTG CG) and transfected inLD_KMP-11_KO lines with higher selection of G418 (100 µg/ml). Positive clones were screened in 96-well plates with wells expressing GFP-positive signals using Fluorescence Microscopy. KMP-11 complemented lines (LD_KMP-11_KO /KMP-11 complemented) were further confirmed by checking for the presence of GFP (717 nt) by PCR and expression of GFP with wild-type LD morphology.

## In vitro proliferation assay, viability assay, and percentage of metacyclic LD

For measuring in vitro proliferation, equal inoculums of promastigotes ($1 \times 10^2$) for each line (WT_LD, LD_KMP-11_KO, and LD_KMP-11_KO /KMP-11 complemented were washed and re-suspended in an equal volume of growth medium (M199–10% FBS) (5 mL). Total numbers of parasites were enumerated each day (day 1 to day 6) microscopically by a light microscope (Leica).To determine cell viability, stationary phase LD promastigotes (WT_LD, LD_KMP-11_KO, and LD_KMP-11_KO /KMP-11 complemented) were seeded ($1 \times 10^5$) in 96-well plates in M199–10% FBS media. MTT (Sigma-Aldrich, St. Louis, MO) was added and incubated for 4 h at room temperature. After incubation solubilizing agents [0.04 N HCL (Merck) in isopropanol (Merck)] were added. Optical density (O.D) was measured after 30 min in a plate reader at 570 nm. The relative number of live cells was determined based on the optical absorbance of the treated and untreated samples and of blank wells, as described previously (Mukherjee et al, 2012). For measuring the percentage of metacyclics, $10^6$ parasites/mL for each parasite line were diluted accordingly to have approximately 100 parasites/mL which was then used to set up the initial culture for all the parasite line. Culture was set in 5 mL medium, and they attend stationary phase by 6th day. Stationary phase promastigotes from the 6th day cultures of parental, LD KMP-11_KO, and LD KMP-11_KO /KMP-11 complemented were washed, re-suspended in PBS and percent metacyclics were determined by Flow Cytometry using light scatter parameter as reported previously (Saraiva et al, 2005). For measurement of interaction of LD with supported lipid bilayer LD promastigotes from WT_LD and LD_KMP-11_KOwere labeled with Carboxyfluor-esceinindiacetatesuccinimidylester (CFSE, Invitrogen,) labeling was performed at a 2 µM concentration for $10^7$ parasites/mL for 30 min in

dark after which LD promastigotes were washed twice in PBS and added to supported lipid bilayer.

## In-vitro CHOL trafficking assay

MΦs were pre-incubated with NBD-CHOL for 16 h. After 16 h of incubation, cells were washed with RPMI-1640 and cultured for an additional 2 h in RPMI-1640 to remove any background (Maxfield and Wüstner, 2012). Labeled cells were infected with LD for 4 h, washed to remove the loosely attached parasites, and fixed immediately or left uninfected. Cells were fixed with 2% PFA and stained with anti-KMP-11 (secondary antibody: Alexa Fluor 594 goat anti-rabbit IgG Catalog # A-11037) and CD11b antibodies and observed under a confocal microscope.

## Quantification of macrophage plasma membrane cholesterol

The MΦ plasma membrane was prepared following a previously established method (Ghosh et al, 2012) (with slight modifications). Initially, macrophages underwent multiple cycles of freezing and thawing, followed by partial cell disruption through syringe-passing techniques. The resulting homogenate was then centrifuged at 900 $\times g$ for 10 min at 4 °C which essentially pellet down Leishmania parasites. The supernatant thus obtained was again filtered before subsequent processing. The supernatant obtained was filtered through a nylon mesh (100 µm), while the pellet was discarded. Subsequently, the supernatant underwent further centrifugation at 50,000 $\times g$ for 20 min at 4 °C. The resultant pellet was re-suspended in a buffer solution containing 50 mM Tris, 1 mM EDTA, 0.24 mM PMSF, and 10 mM iodoacetamide (pH 7.4), centrifuged again at 100,000 $\times g$ for 20 min at 4 °C. The final pellet, representing native plasma membranes, was suspended in a minimal volume of buffer solution provided with the Amplex-Red assay kit. Immediately after the membrane isolation, cholesterol concentration was measured by the Amplex-Red assay kit. The protein content of the membrane was quantified using a Bio-Rad reagent following the manufacturer's instructions. The total cholesterol content represented here by the µg of cholesterol/µg of total protein (Ghosh et al, 2012).

## *FRET study for the attachment of LD_KMP-11_KO/GFP KMP-11 complemented with MΦ*

LD_KMP-11_KO/GFP KMP-11 complemented lines were added to Rhodamine DHPE labelled MΦ and FRET between GFP (attached with KMP-11 in LD) and Rhodamine on MΦ membrane was measured after 30 min of post-infection. In another condition, KMP-11_KO/GFP KMP-11 complemented lines were pretreated with anti KMP-11 antibody and then added to MΦ and subsequently FRET measurement was performed. FRET was measured in a PTI fluorometer. Here GFP was used as donor (excitation 475 nm and emission 510–540 nm) and Rhodamine (excitation at 546 nm, emission 570–640 nm) was used as acceptor.

## Statistics and reproducibility

All results were confirmed in at least three independent experiments performed on different days with different batches of samples. The results were expressed as the mean ± SD derived from

multiple data points. Statistical significance in the observed differences was determined through analysis of variance (ANOVA) utilizing GraphPad Prism (9.0) software. A significance level of $P < 0.05$ was considered indicative of meaningful differences.

## Data availability

No data has been deposited in any public database. All Source Data is available via download from the respective figure.

The source data of this paper are collected in the following database record: biostudies:S-SCDT-10_1038-S44319-024-00302-7.

## Peer review information

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

## Acknowledgements

Author AS acknowledges SERB (PDF/2020/000678), Govt. of India for providing national postdoctoral fellowship. Author KC acknowledges the support from the ICMR grant (Number: 6/9-7(295)/2022-ECD-II). SR acknowledges fellowship from the INSA Senior Scientist position (Number: INSA/SP/SS/2024/253). SK acknowledges the financial support from DBT-funded research project (BT/PR8475/BRB/10/1248/2013). BM acknowledges SERB funding from the project SRG SRG/2020/000624/LS. SG is a recipient of GATE fellowship. SP is a recipient of CSIR-NET and PMRF fellowship. KC acknowledges funding from the CSIR network project grant HOPE. We thank Professors Carl Frieden and Elliot Elson of Washington University School of Medicine for their critical comments and suggestions on manuscript. We thank Professor Roop Mallik of Indian Institute of Technology, Mumbai for multiple exciting discussions on this study, which were essential for the shaping up of this manuscript. We thank the director, CSIR-IICB for help and encouragement.

## Author contributions

**Achinta Sannigrahi**: Conceptualization; Formal analysis; Validation; Investigation; Methodology; Writing—original draft; Writing—review and editing. **Souradeepa Ghosh**: Formal analysis; Investigation. **Supratim Pradhan**: Formal analysis; Investigation. **Pulak Jana**: Formal analysis; Investigation. **Junaid Jibran Jawed**: Formal analysis; Investigation. **Subrata Majumdar**: Resources. **Syamal Roy**: Conceptualization; Supervision; Writing—review and editing. **Sanat Karmakar**: Conceptualization; Methodology; Writing—review and editing. **Budhaditya Mukherjee**: Conceptualization; Resources; Project administration; Writing—review and editing. **Krishnananda Chattopadhyay**: Conceptualization; Resources; Supervision; Project administration; Writing—review and editing.

Source data underlying figure panels in this paper may have individual authorship assigned. Where available, figure panel/source data authorship is listed in the following database record: biostudies:S-SCDT-10_1038-S44319-024-00302-7.

## Disclosure and competing interests statement

The authors declare no competing interests.

# Expanded View Figures

**Figure EV1. Complementation and characterization of LD_KMP-11_KO lines.**

(A) In vitro growth kinetics comparing WT_LD, LD_KMP11_KO and LD_KMP11_KO /KMP-11 GFP complemented lines with equal inoculums ($10^2$/ml) over a period of 6 days. Growth rate of WT_LD is significantly higher post 4th day of in vitro culture. The data were expressed as the mean ± SD derived from three independent experiments ($n = 3$) for each group. (B) MTT assay comparing viability of WT_LD, LD_KMP-11_KO and LD_KMP11_KO /KMP-11 GFP complemented lines in equal number ($10^5$) of WT_LD, LD_KMP-11 KO and LD_KMP11_KO /KMP-11 GFP complemented lines from a 6th day culture used for primary inoculum. Data has been presented as mean ± SD derived from three independent experiments ($n = 3$). The level of significance has been estimated using unpaired t-test in GraphPad Prism (version 9) application. Here $^{ns}P$ value = 0.5391. (C) Detection of stationary LD promastigotes by Flow cytometry on the basis of scatter light. First panel showing the WT_LD, middle panel showing the KMP-11KO LD and the right panel showing the KMP-11 complemented KMP-11 KO LD. This data is a representative of three independent experiments. (D) Flow cytometry representing percent metacyclic in stationary phase (6th day culture) of WT_LD, LD_KMP-11_KO and LD_KMP11_KO /KMP-11 GFP complemented lines. WT_LD lines showing slightly higher percentage of metacyclic than LD_KMP-11_KO and LD_KMP11_KO /KMP-11 GFP complemented lines. Data has been represented as mean ± SD ($n = 5$, the number of independent experiments for each group).The level of significance has been estimated using ANOVA in GraphPad Prism (version 9). Here, $^{ns}P$ value = 0.3382 (WT_LD vs LD-KMP-11_KO) and $^{ns}P$ value = 0.8312 (WT_LD vs LD-KMP-11_KO/KMP-11 complemented). (E) A significant higher expression of KMP-11 was observed by western blot in case of WT_LD (passage 3, P3) as compared to LD lines having more than 200 passages (LD > P200) in M199 in vitro culture. Macrophage membrane fluidity change and CHOL depletion by KMP-11. (F) The values of FA of MΦ membrane upon the treatment of WT_LD, LD_KMP-11_KO, LD > P200, LD_KMP-11_KO/KMP-11 GFP Complemented and LD_KMP-11_KO, LD > P200 in presence of exogenous r-KMP-11 (50 µM). Data has been represented as mean ± SD ($n = 2$, number of independent experiments). (G) Membrane fraction was isolated from the MΦs by ultracentrifugation. Total membrane cholesterol (free and esterified) was estimated using the Amplex-Red kit and expressed as µg of cholesterol/µg of total protein. MΦ membranes were treated with WT_LD, LD_KMP-11_KO, LD > P200, LD_KMP-11_KO/KMP-11 GFP Complemented and LD_KMP-11_KO, LD > P200 in presence of exogenous r-KMP-11 (50 µM). Data has been represented as mean ± SD ($n = 2$, number of independent experiments).

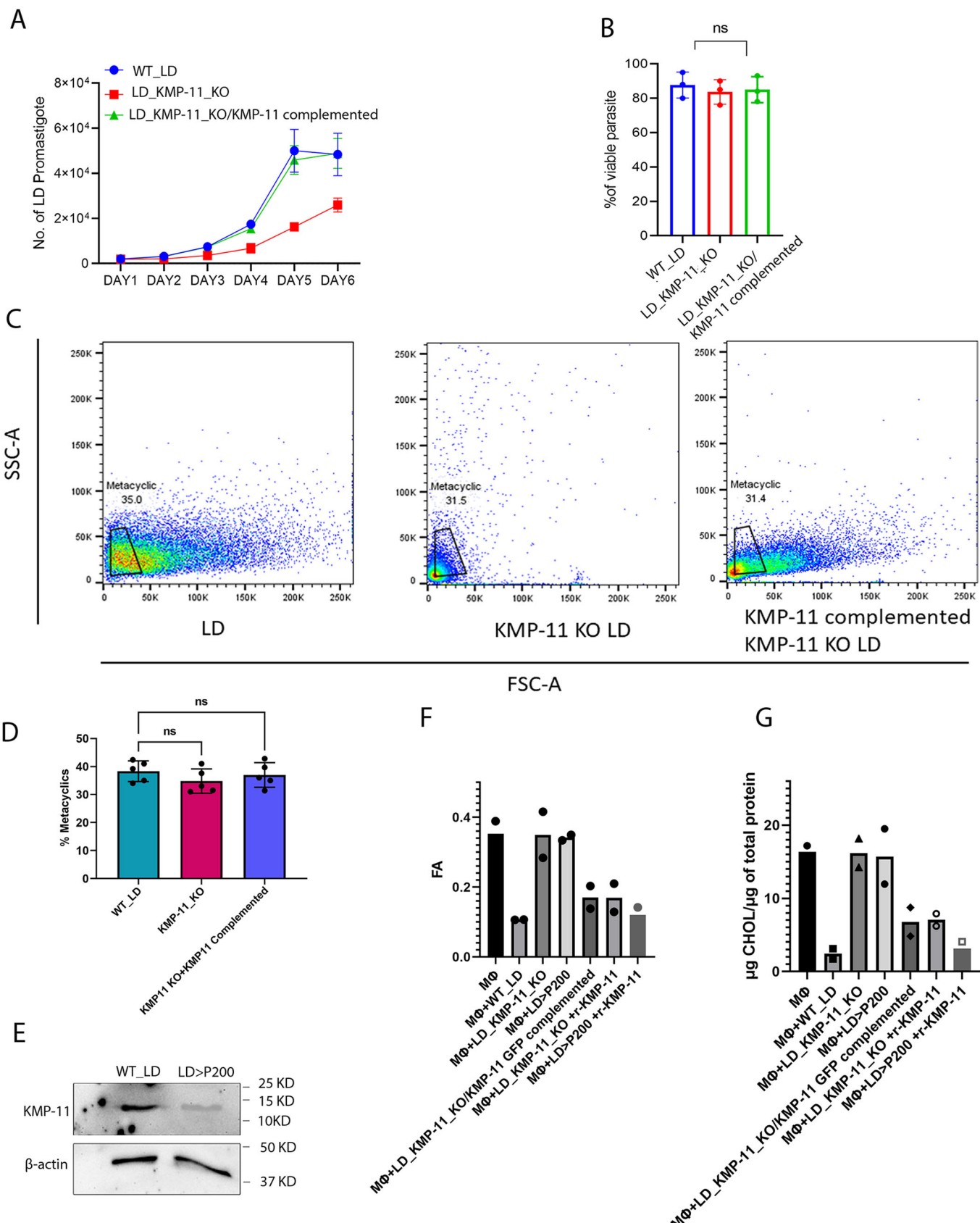

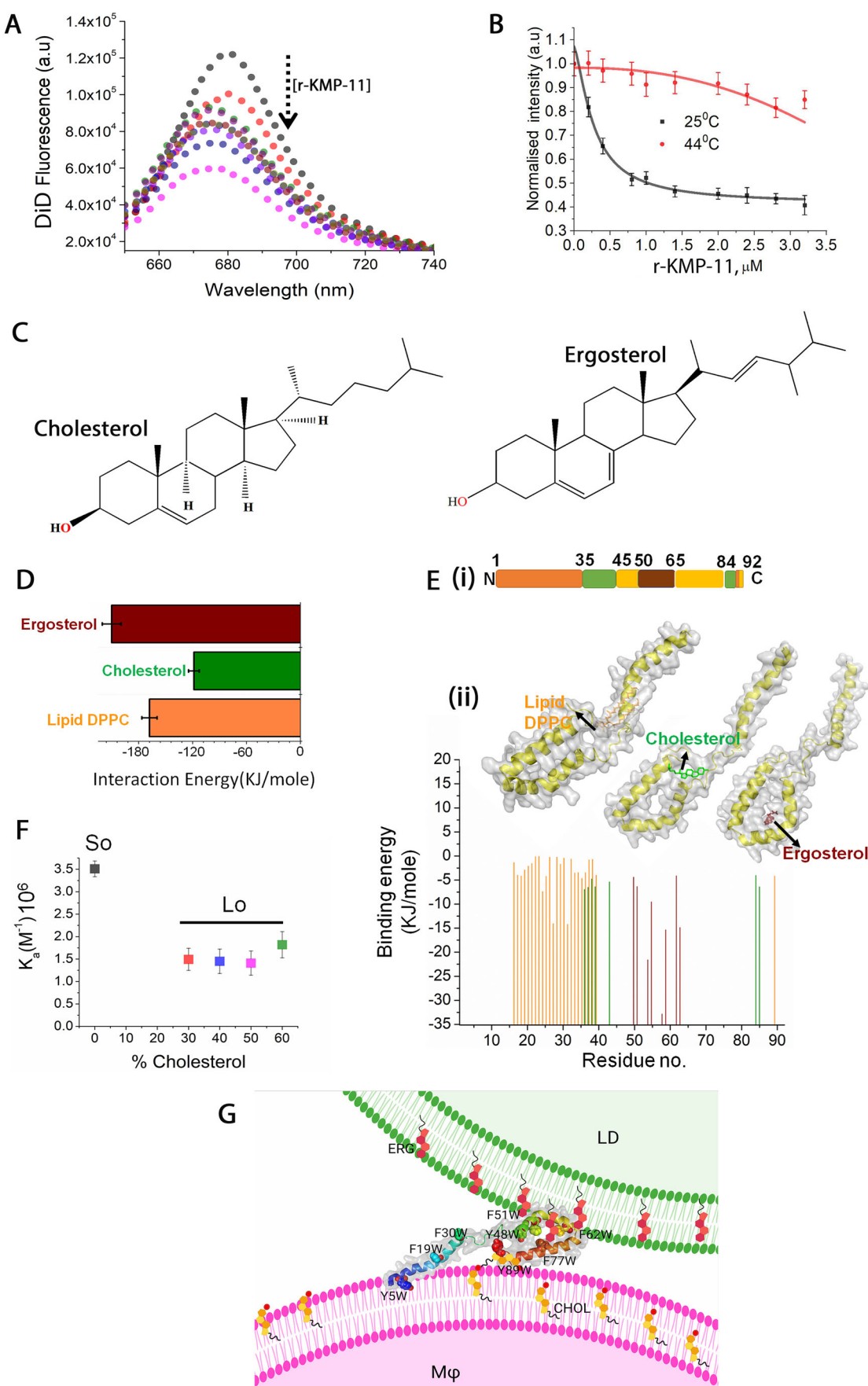

**Figure EV2.  KMP-11 binding with model DPPC SUVs.**

(A) Representative DiD fluorescence titration spectra shows the binding of r-KMP-11 with DPPC model membranes as was monitored by the decrease in DiD fluorescence intensity (left *y* axis) as well as the blue shifts of the fluorophore (right *y* axis) with increasing concentration of added proteins. (B) Plot of normalized membrane embedded DiD fluorophore intensities against increasing WT r-KMP-11 concentration to estimate the binding of r-KMP-11 with DPPC at two different temperatures (25 °C and 44 °C). Two different temperatures were used to see the binding of r-KMP-11 with gel and fluid DPPC membrane. The solid lines indicated the fit of the data to evaluate the binding affinities. Our data showed the binding affinity with Lo/gel phase DPPC (25 °C) is considerably higher than the Ld/fluid phase DPPC membrane (44 °C).The data were expressed as the mean ± SD derived from three independent experiments ($n = 3$) for each temperature group. (C) Chemical structures of cholesterol and ergosterol. Region-specific binding of lipid and sterol molecules with KMP-11. (D) The binding energy variation of KMP-11 with DPPC lipid, CHOL and ERG as obtained by docking analysis. The data were expressed as the mean ± SD derived from three independent experimental repeats ($n = 3$) for each group. (E) A schematic drawing of the overall protein sequence which is marked based on the binding locations of CHOL (green), ERG (brown) and DPPC lipid (orange) as obtained from the molecular docking study. The best posed docked structure of KMP-11 as obtained from docking study was shown. The amino-terminal domain shows the DPPC lipid binding domain, 35–45 AA region and 85–90 AA regions are found to be CHOL binding region, 50–70 AA region is the ERG binding motif. Plot of binding energy of individual lipid and sterol components against the residue number of KMP-11 as obtained from gemdock tool has been shown. Snapshots of the best posed docked structures for CHOL-KMP-11; ERG-KMP-11 and DPPC lipid-KMP-11 as obtained from the molecular docking study. (F) Binding of r-KMP-11 with DPPC SUVs containing different mole percentages of CHOL as obtained from the membrane embedded DiD quenching assay. Data indicated a notable decrease in the r-KMP-11 binding with DPPC SUVs containing different percentage CHOL. Here, So and Lo stand for solid order and liquid order states respectively. This data signifies a higher binding affinity of r-KMP-11 towards So domain in comparison to Lo domain. The data were expressed as the mean ± SD derived from three independent experimental repeats ($n = 3$) for each CHOL percentage. (G) The plausible orientation of KMP-11 during attachment of the parasite (DPPC-ERG) with MΦ (DPPC-CHOL) membrane.

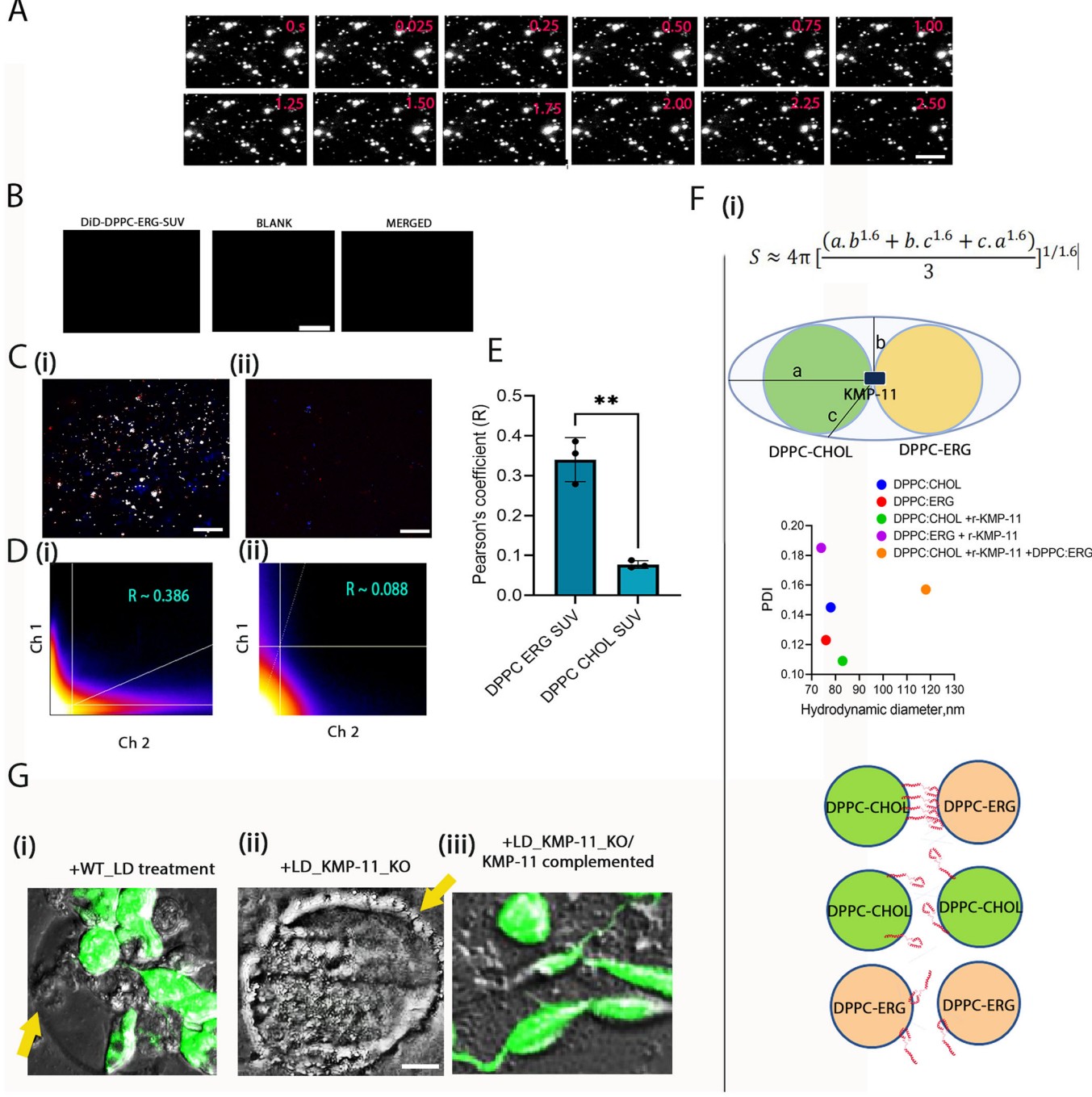

$$S \approx 4\pi \left[\frac{(a.b^{1.6} + b.c^{1.6} + c.a^{1.6})}{3}\right]^{1/1.6}$$

◀ **Figure EV3. The bridging nature of KMP-11.**

Co-localization study on PEG cushioned SLB. (**A**) Time lapse images of the DiD-labeled DPPC: ERG SUVs on DPPC: CHOL SLB platform in presence of r-KMP-11. The dimension of each image was kept 32.19×25.94 μm². Time lapse images presented here were processed from the continuous recording of Fig. 4Bi. Scale bar is 5 μm. (**B**) TIRF microscopic images of DiD-labeled DPPC ERG SUVs (red channel), blank (blue channel) and the merged image. Scale bar is 10 μm. (**C**) (i) Colocalised micrograph of DiD-labeled DPPC ERG SUVs (red channel) and Alexa-488 maleimide labeled r-KMP-11 (blue channel). Scale bar is 10 μm. (ii) colocalised micrograph of DiD-labeled DPPC CHOL SUVs (red channel) and Alexa-488 maleimide labeled r-KMP-11 (blue channel). Scale bar is 10 μm. (**D**) Scatter plots of colocalisation of (i) DiD-labeled DPPC ERG SUVs (red channel) and Alexa-488 maleimide labeled r-KMP-11 (blue channel) system and (ii) DiD-labeled DPPC CHOL SUVs (red channel) and Alexa-488 maleimide labeled r-KMP-11 (blue channel). The R (pearson's coefficient) values stand for the degree of colocalisation. (**E**) Plot shows the values of pearson's coefficients obtained from TIRF imaging using DPPC:CHOL and DPPC: ERG SUVs. Data were presented as mean ± SD derived from three independent experiments for both groups. The level of significance has been estimated using unpaired *t* test in GraphPad Prism (version 9) application. **$P$ value = 0.0012. (**F**) (i) Dynamic light scattering (DLS) study to estimate the hydrodynamic radii of different sterol-containing SUVs. We observed comparable hydrodynamic radii for DPPC-ERG and DPPC-CHOL. Hydrodynamic radius increased significantly when we added r-KMP-11 to the equimolar mixture of DPPC-ERG and DPPC-CHOL. Implying ellipsoid approximation while two vesicles are attached by a KMP-11 bridge, we calculated the average surface area of the combined system, which was found to be ~43,870.42 nm². Interestingly, we found that our calculated surface area matched well with the surface area obtained from DLS data for DPPC-ERG + DPPC-CHOL+r-KMP-11 system (observed surface area ~41,094 nm²), which further suggests that KMP-11 can act as a bridging molecule between CHOL and ERG rich membranes. Here, PDI stands for poly-dispersity index. (ii) Different orientations of KMP-11 in SUV environment in presence of CHOL and ERG in SUVs. (**G**) Images representing interaction of CFSE labeled WT_LD and LD_KMP-11_KO and complemented LD lines on supported Lipid Bilayer (SLB) composed of DPPC CHOL 30%. Significant attachment of (i) WT_LD parasites was observed on lipid surface which is absent for (ii) LD_KMP-11_KO parasites. Attachment is restored for (iii) LD_KMP-11_KO/complemented parasites on SLB surface. The yellow arrows indicate the edge of the supported lipid bilayer. Scale bar is 1 μm. Source data are available online for this figure.

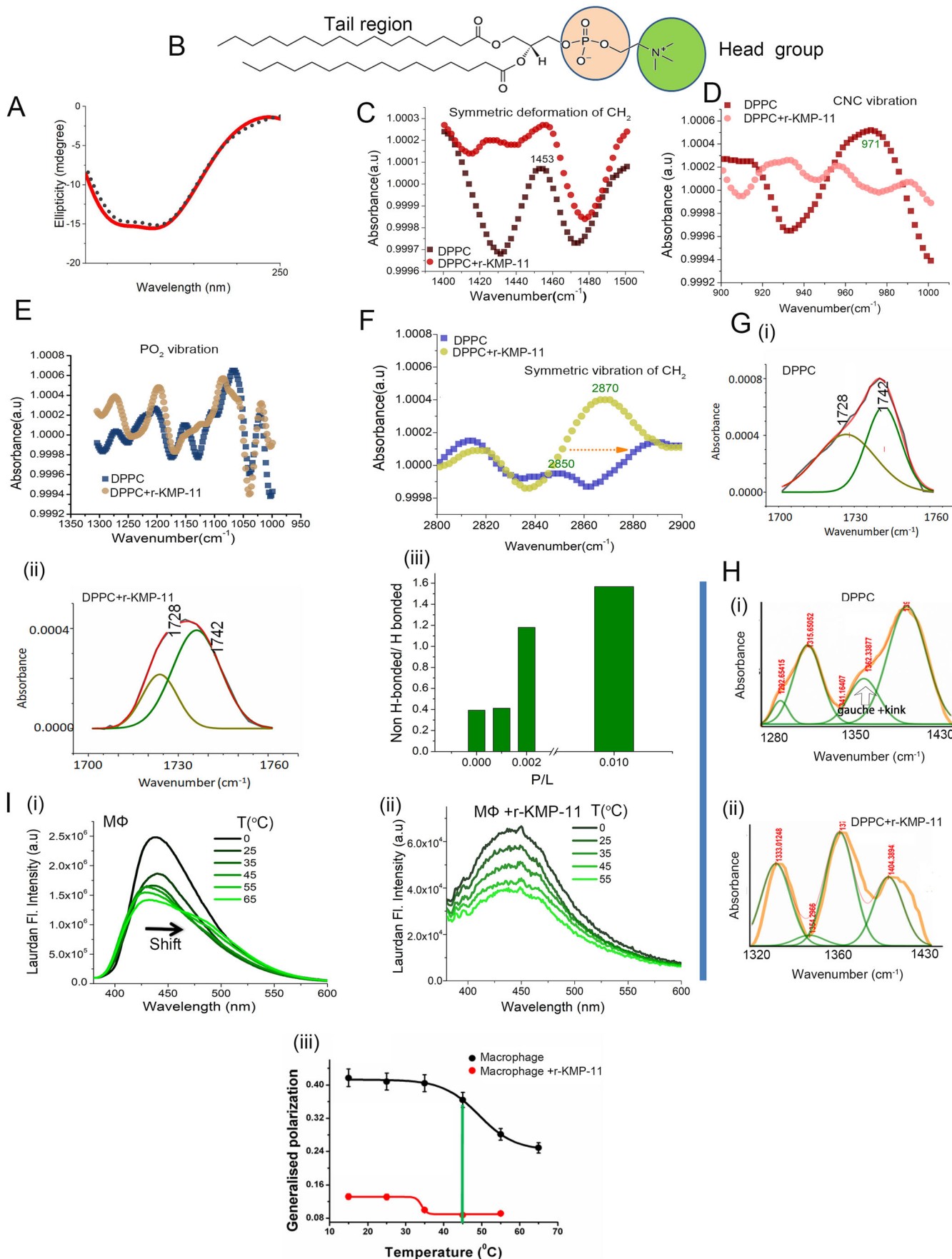

◀    **Figure EV4.  Interactions induced protein–lipid conformational perturbations.**

(**A**) Far UV-CD spectra of r-KMP-11 in the absence and presence of DPPC membrane. No significant conformational change in the secondary structure of r-KMP-11 due to membrane binding was evident from the CD data. Membrane perturbations study due to KMP-11-lipid binding. (**B**) The molecular structure of DPPC lipid in which the head group and tail group regions are marked. FTIR signatures of (**C**) C-N-C, (**D**) $PO_2$. (**E**) Symmetric deformation of $CH_2$. (**F**) Symmetric vibrations of $CH_2$, both in absence and presence of r-KMP-11. (**G**) Deconvolution of carbonyl (C = O) stretching vibrations in (i) absence and (ii) presence of r-KMP-11 to measure the population of hydrogen bonded carbonyl frequency (appears at 1728 $cm^{-1}$) and nonhydrogen bonded frequency (at 1742 $cm^{-1}$). Interestingly, r-KMP-11 binding increased the population of nonhydrogen bonded carbonyl frequency in a concentration-dependent manner as shown in (iii). Moreover, enhanced nonhydrogen bonded vibrational states due to r-KMP-11 binding also indicated bilayer thinning. (**H**) Deconvoluted FTIR spectral signatures of the $CH_2$ wagging band frequency of DPPC in absence (i) and the presence (ii) of r-KMP-11. This FTIR data suggested the increase of gauche rotamers of DPPC due to protein binding. (**I**) Measurement of MΦ membrane fluidity in the absence and presence of KMP-11. Plot of laurdan emission intensity of labeled MΦ with increasing temperature in the (i) absence and (ii) presence of r-KMP-11. (iii). Generalized polarization (GP) of laurdan labeled macrophage membrane with respect to temperature in absence (black) and presence (red) of r-KMP-11. Typically, concentration of r-KMP-11 was taken 50 µM. The data were represented as the mean ± SD derived from three independent GP experiments ($n = 3$) for each temperature.

    