## [Peer Review File · EMBO Reports]

Leishmania protein KMP-11 modulates cholesterol transport and membrane fluidity to facilitate host cell invasion

Achinta Sannigrahi, Souradeepa Ghosh, Supratim Pradhan, Pulak Jana, Subrata Majumdar, Syamal Roy, Sanat Karmakar, Budhaditya Mukherjee, Krishnananda Chattopadhyay, and Junaid Jawed

Corresponding author(s): Krishnananda Chattopadhyay (krish@iicb.res.in) , Budhaditya Mukherjee (bmukherjee@smst.iitkgp.ac.in)

Review Timeline:

Transfer Date:	16th Jul 24
Editorial Decision:	19th Aug 24
Revision Received:	30th Aug 24
Editorial Decision:	16th Sep 24
Revision Received:	23rd Sep 24
Accepted:	27th Sep 24

Transaction Report: This manuscript was transferred to EMBO reports following peer review at The EMBO Journal.

Point-by- point Discussion on the Reviewers' report

We thank the reviewers for their valuable comments and suggestions. The constructive comments by reviewers have been extremely useful in improving the manuscript. Please note that we have included several additional results in the revised manuscript to specifically address the concerns raised by the reviewers. The brief descriptions of these are appended below:

General Summary of the additional experiments and new results:

1. Quantification of cholesterol in macrophage plasma membranes to directly monitor the role of KMP-11 in cholesterol quenching process.
2. Detailed characterization of the KMP-11 knock out and complemented (*Leishmania donovani*) LD lines in terms of viability, growth kinetics, percentage metacyclic etc.
3. Specificity of r-KMP-11 treatment has been assessed by measuring apoptosis and the relative levels of IL12 and IL10 under r-KMP-11 and LPS control conditions.
4. Several infection experiments with LD lines (WT or KMP-11_KO) in macrophages in presence of KMP-11 antibody or exogenous r-KMP-11 has been included to convincingly show that KMP-11 plays role in LD attachment and infection of macrophages along with its biochemical complementation. Moreover, attachment assays with WT_LD, LD_KMP-11_KO and KMP-11 complemented LD on supported lipid bilayer have been used to show that macrophage attachment process depends on KMP-11.
5. And finally, FRET experiment has been included to show conclusively that KMP-11 on LD surface can bind with macrophage membrane during the attachment process.

Specific Points to address each of the two reviewers' concerns:

Referee #1:

Summary

In this work Sannigrahi and cols. investigate the involvement of a membrane protein (KMP-11), which is produced by the pathogen *Leishmania donovani*, in the process of cell invasion of macrophages. The authors started with in silico studies to demonstrate the homology between this protein and other proteins that can interact with cholesterol. Using the purified protein, knockout parasites, an impressive set of biochemical analysis together with infection assays, the authors conclude that some residues of this protein interacts with cholesterol present on host cell plasma membrane, that the protein work as oligomers and that it is a key factor promoting the interaction of the parasite with the host cells and parasite internalization.

Major Points

I recognize the incredible amount of data produced by the authors, I do believe this a very important subject, I recognize that KMP-11 is a very good candidate to interact with cholesterol, based on the first set of experiments, but I am convinced that

additional experiments and quantification are needed to support the conclusions made. I see an enormous potential for this work and I would be glad if I see it published latter with more substantial analysis and quantification.

Reply: We are grateful to the reviewer for the valuable comments. We have carefully considered the comments, carried out additional experiments and analyses and revised the manuscript thoroughly and extensively and provided better quality images to address the comments. We believe that the revised manuscript is significantly better, technically more sound with improved readability.

In this regard we would like to mention that we have included direct quantification of cholesterol in macrophage plasma membrane in the presence of KMP-11 to substantiate the role of KMP-11 in cholesterol removal. We have performed this experiment by treating primary macrophages isolated from BALB/C mice with r-KMP-11 (50 μ M). Membrane fractions isolated from r-KMP-11 treated macrophages were used to quantify the total cholesterol content using Amplex-Red Kit (Invitrogen) as compared to untreated control. Our result convincingly showed a significant drop in membrane cholesterol content in the presence of r-KMP-11 which is comparable to LD-infected macrophage membrane. Parallel experiments performed in RAW cells in identical experimental conditions also showed a prominent decrease in membrane cholesterol content in the presence of r-KMP-11 treatment or as a result of LD infection. Thus, these results along with previous results on CtxB clearly show a significant drop in membrane cholesterol content by r-KMP-11 treatment, which is similar to LD infection (Figure R1, Figure 1Ei,ii in the revised manuscript).

Figure for referee with unpublished data and its description has been removed upon request by the authors.

Comment 1: I think that the way the manuscript is written right now is not facilitating the reading. Information flow and the links between the different experiments are very often left to the reader. The manuscript seems too long and ideas should be presented in more concise and objective way.

Reply: We agree. We have worked on it extensively. Different experiments are now bridged using suitable texts, which will hopefully guide the readers to understand next experiments/ideas. We have worked hard to shorten the manuscript and to add more focus. With that in mind, we have deleted the portion related to effect of KMP-11 on cytoskeleton disassembly and phagocytosis. We have realized that although we see the effect of KMP-11 in these processes we cannot convincingly comment if lipid rafts, and cytoskeleton are interdependent and changes in one cause changes in another and causality can go in both directions. It is possible that KMP-11 can break down the rafts and that affect cytoskeleton, or it may target cytoskeleton and that affect the rafts. We believe more experimental work is required to completely substantiate this claim of the possible effect of KMP-11 on cytoskeleton disassembly, which itself could be a completely new study.

Comment 2: I think that more details about the knockout parasite, their cellular viability and their ability to normally perform metacyclogenesis is extremely needed. These are very important controls. The morphological changes observed in knockouts are major. I think that in the case of such striking morphological changes two major questions should be checked. 1- Are parasites in both cultures equally viable? Any viability assay such as erythrosin B, LDH, Propidium iodide were performed to ensure parasite viability? 2- Promastigotes must undergo a complex differentiation process known as metacyclogenesis in order to become infectious (the metacyclic promastigotes). This process takes place in the vector but it can be mimicked in vitro: at the stationary phase of growth parasites tend to become metacyclic whilst recently inoculated parasites are procyclic. Procyclic parasites are not infectious which leads to my major concern: are these parasites comparable in terms of metacyclogenesis? Both growth curves were accompanied and compared?

Reply: We thank the reviewer for these suggestions, which we completely agree. We have carried out these experiments and incorporated the results in the revised manuscript. To summarize:

- 1. We observed that LD_KMP-11_KO lines have compromised growth when compared to WT and complemented lines (Figure R2, Figure EV 3).**
- 2. WT_LD, LD_KMP-11_KO and LD_KMP-11_KO/KMP-11 complemented lines have comparable viability in promastigote culture (Figure R2, Figure EV 3).**
- 3. In terms of metacyclics, we did not observe any significant difference between percent metacyclics for WT_LD, LD_KMP-11_KO and LD_KMP-11_KO/KMP-11 complemented lines (Figure R2, Figure EV 3).**

Furthermore, for host cell attachment and invasion assays with the WT_LD, LD_KMP-11_KO and LD_KMP-11_KO/KMP-11 complemented lines we have performed these assays by sorting an equal number of metacyclics for WT_LD, LD_KMP-11_KO and LD_KMP-11_KO /KMP-11 complemented lines using CytoFLEX SRT (Beckman Coulter). These results have been included in the results section of revised manuscript and described in detail in materials and methods section.

Figure for referee with unpublished data and its description has been removed upon request by the authors.

Comment 3: Since the molecular interaction proposed happens during the first step of infection, when parasites are at the extracellular environment, experiments using antibodies against this protein during invasion by WT parasites may be more conclusive, much more than add backs (which are also important).

Reply: This is an important suggestion, and we are thankful to the reviewer for showing us the way. We have performed the infection by preincubating WT_LD with anti-KMP-11 antibody. We observed that blocking KMP-11 by the antibody drastically decreased the invasion of parasite to the macrophages (Figure 1F and G in the revised manuscript; Figure R3 below).

Figure for referee with unpublished data and its description has been removed upon request by the authors.

Comment 4: The whole cell biology part such as TIRF, TEM, AFM and confocal microscopies needs special attention. I think that the quality of the images provided should be improved, extra images should be provided and more quantification of cell biology data should be given to support conclusions.

Reply: We thank the reviewer for the comment and suggestion. We have revised all the images in the revised manuscript. Quality of the images are now better (few specific examples include, Figures 4B, F, H; Figure 5F, EV1A, Figure EV7C). We have added additional images (for example, Figure 6, Figure 2F, Figure EV5G). Quantifications have been performed using most of the images and the data have been provided with the details of the measurement included in the Materials and Method section.

Minor Points

Comment 1: "The first critical step, which determines the fate of any intracellular pathogen, relies on its ability to enter its host successfully. In this paper, we have studied the mechanism of parasite-host cell entry using Leishmania donovani (LD) as a model."The entry of the host during Leishmania infection occurs by the bite of the vector which inoculates the infective promastigotes in the dermis of the host. Host cell invasion and the entry in the host (the animal) are different things. "facilitating the process of host cell infection by intracellular protozoan". Sounds like it is being generalized for other parasites.

Reply: We appreciate the reviewer's comment and suggestion. We have modified the sentence in the revised manuscript and highlighted. Please refer to the text

"The successful infection of any intracellular pathogen relies on its efficient invasion through the host cell membrane, which strongly depends on host membrane properties (morphology, fluidity, and rigidity)."

"Through tryptophan-scanning mutagenesis and synthesized peptides, we developed a generalized mathematical model, which suggests that the hydrophobic moment and the symmetry sequence code at the membrane interacting protein domain are key factors in facilitating the membrane phase transition and, consequently, the host cell infection process by Leishmania parasites."

Comment 2 "a key component of eukaryotic membrane, plays crucial roles in the entry process into the host. Same. The entry of the host during Leishmania infection occurs by the bite of the vector which inoculates the infective promastigotes in the dermis of the host. Host cell invasion and the entry in the host (the animal) are different things.

Reply: We have removed this particular sentence and modified the introduction portion in the revised manuscript.

Comment 3: "Since the accessibility of CHOL in host membrane is essential for the CHOL scavenging pathogens to transport from host to pathogen, pathogens (Coppens, 2013; Kumar et al, 2016) often use special machinery to break the sequestered (non-accessible) CHOL complex in host cells to increase CHOL accessibility (Palladino et al, 2022; Zhang et al, 2009)." This Whole sentence is quite confusing. Refs are also misplaced. "the mechanism of CHOL transport and the host membrane modifications is still an unsolved question." Not clear whether this sentence is about host cell invasion or cholesterol transport during the intracellular life of these pathogens. Refs are also needed here.

Reply: Although it is true that similar to acquisition of host cholesterol during invasion intracellular pathogens can also acquire cholesterol by diverting components of endocytic pathway during their replicative phase inside their host. However, here we are focusing only towards acquisition of cholesterol from host membrane during invasion.

We have modified this portion using appropriate references in the revised manuscript. Please refer to the following text,

"In some cases, there are CHOL quenching events during the onset of infection, which are facilitated by pathogen-derived proteins that transfer CHOL 9, 10. Since the accessibility of CHOL in the host membrane is crucial for CHOL-scavenging pathogens to transport from the host to the pathogen, there is a requirement of disrupting the sequestered (non-accessible) CHOL complex in host cell membrane, thereby increasing CHOL accessibility^{11-13}. Although these events are known to exist for multiple intracellular pathogens^{9, 14, 15} such as *Mycobacterium*, *Toxoplasma*, *Plasmodium*, *Leishmania*, and even for viruses like SARS-CoV-2, the mechanism of CHOL transport and the modifications in the host membrane during the initial process of host cell invasion remain an unresolved problem."

Comment 4: "Since CHOL has been reported to be one of the critical determinants during *Leishmania* infection, we for the present study, used infection through *Leishmania donovani* (LD) as our model."

References are very important here. Punctuation needs attention. "These results made us hypothesize that there may be a specific LD-protein(s) which might regulate this transfer of CHOL from the host environment onto the parasite." Not clear whether the authors are talking about cholesterol acquisition during host cell entry or during the intracellular life cycle of the parasite in the context of the endocytic pathway of the host, where parasites live.

Reply: As already mentioned, here we are focusing only towards acquisition of cholesterol from host membrane during initial invasion. We have included appropriate references in the revised manuscript and modified the sentence as follows:

'These findings clearly indicate that the process of CHOL transfer might be important in *Leishmania* infection, and there might be specific *Leishmania* proteins, which would be responsible for executing the transfer of CHOL from the host membrane to parasite during invasion.'

Comment 5: "First, using a sequence analysis we found that KMP-11 sequence contains CHOL interaction motifs [CHOL recognition and consensus motif, CRAC](Azzaz et al, 2022; Wang et al, 2022) including two CRAC like domains and one

CARC domain, and these domains are widely conserved in different leishmania species (Figure 1A)."

The acronyms are confusing and one of them seem not introduced. "in the regulation of the overall lipid bilayer pressure of the parasite membrane" What exactly does it mean? Would authors mean "tension"? If it is lipid pressure this is a much less intuitive concept and should be better explained in my opinion. "including two CRAC like domains and one CARC domain, and these domains are widely conserved in different leishmania species (Figure 1A). Second, we found a significant structural similarity between KMP-11 and lipid transfer proteins (LTPs). Employing structural alignment with the Raptor X tool, we found a significant overlap between KMP-11 and both CHOL transport START domain and CHOL sensing GRAM domain of LTPs (Figure Bi,ii,iii). From these sequence analysis studies we speculated that KMP-11 can have an LTP-like function, which the parasite can presumably use to transport CHOL from the host membrane to facilitate productive infection (Figure 1B)."

Please, note that current results are presented within the introduction. From the paragraph that starts with "One of the major" to "process of infection" I believe that too much detail about the new findings are presented for an introduction.

Reply: We have now provided the brief description of CRAC in the revised manuscript and the appropriate references are added to eliminate confusion.

"in the regulation of the overall lipid bilayer pressure of the parasite membrane" By this statement we tried to say that KMP-11 maintains the morphology of the parasite bilayer as shown in previous literatures.

We have modified this statement by "KMP-11 has been implicated in regulating the overall lipid bilayer morphology of the parasite membrane²³ and highlighted.

Since, Cholesterol sequestration by parasite surface protein is the principal objective in our study; we validate this cholesterol quenching ability of KMP-11 by some two simple bioinformatics analysis. Therefore, we discussed the sequence similarity data and structural resemblance results in the introduction section only as an initial validation against the speculation that KMP-11 can be a candidate for macrophage membrane cholesterol sequestration.

Comment 6: "It may be noted that MΦ membrane contains approximately 30% CHOL, and hence we used DPPC containing 30% CHOL (DPPC-CHOL) as a simplified mimic of the host MΦ membrane"References are important here.

Reply: We have provided the reference in the revised manuscript (Subczynski, W.K., Pasenkiewicz-Gierula, M., Widomska, J., Mainali, L. & Raguz, M. High cholesterol/low cholesterol: effects in biological membranes: a review. *Cell biochemistry and biophysics* 75, 369-385 (2017).

Comment 7: "Collectively, our results demonstrate that KMP-11 plays a pivotal role in the host cell invasion process of leishmania by involving itself into a process of CHOL transport. We believe that this model can be applied by other infectious pathogen-derived membrane interacting proteins further paving the way for potential therapeutic interventions against infectious diseases."This sentence is important, but the way it is written sounds confusing to me.

Reply: We have modified this portion in the revised manuscript as follows:

To our knowledge, this is the first study which comprehensively characterized the pivotal role of KMP-11 to initiate host cell invasion process of *Leishmania* by mediating CHOL transport. Experiments are being carried out in our group to develop potential small molecules, which can target KMP-11 mediated phase transition as a potential therapeutic intervention against leishmaniasis.

Comment 8: "Furthermore, we compared the T-cell stimulating ability between KMP-M Φ and normal macrophages (M Φ) using different concentration of KMP-11 (Figure 1D). Using I-Ad restricted anti-LACK T-cells hybridoma and subsequent IL-2 production in the presence or absence of LACK antigen, we observed that co-culture of anti-LACK-T-cells with M Φ without LACK antigen failed to produce IL-2 (Figure 1D)."

I do believe that this is an interesting topic, but at this point and together with the structural studies and the biochemical characterization presented in Figs 1A - C it is a little bit confusing. The flow at this point was driving me to cell invasion and host cell attachment. I could not understand what is the point of this in vitro assay about macrophages as an antigen presenting cell (APC) here and taking into consideration the proposed flow presented in the introduction.

Reply: We agree with the reviewer. We have included this experiment to provide an indirect quantification of cholesterol removal efficiency of KMP-11 as removal of membrane cholesterol will make the membrane fluid leading to defective antigen presentation and low IL2 generation. We have now modified this portion in the revised manuscript to maintain the flow of the text.

Comment 9: "Our results show that as opposed to untreated and Cytochalasin D (CytoD) control, KMP-11 treatment causes a significant decrease in fluorescence intensity indicating actin depolymerization (Figure 2F, Gi,ii). KMP-11 mediated actin depolymerization was comparable with Methyl- β -cyclodextrin (m β CD), an actin-depolymerizing agent (Mundhara et al, 2019) used as a positive control in this experiment (Figure EV3). Interestingly, we found that the effect of KMP-11 on actin depolymerization was reverted when the M Φ s were treated with liposomal CHOL (Figure 2F, Giii) but not with a CHOL analog (4 cholesten-3-one) that was found to be silent towards the activity of KMP-11 (Figure EV4) further inferring the specificity of CHOL."The order of the figures is not clear, cytochalasin D is not cited in Fig 2F. Fig 2F is cited before Fig. 2E. Are fluorescence microscopy images showing single focal planes?

Reply: We have already mentioned before we feel that this section needs additional experiments and is diluting the focus from the main message of this work which is validating the role of KMP-11 in cholesterol transfer from the host membrane to set up the first step of LD infection. Hence, we have removed this section entirely from the revised manuscript.

Comment 10: "Since LD has been reported to infect host M Φ s by phagocytosis, we investigated the role of KMP-11 in activating phagocytosis in the host."We note that in silico findings were experimentally supported by the FCS data described above. Employing the sequence analyses, docking and FRET"Fig 3I or other already published data?

Reply: No, Figure 3I and related figures are new data presented for the first time in our manuscript.

Comment 11: "In contrast, when we performed similar experiments using DiD labeled DPPC-CHOL SUVs (and not with DPPC-ERG SUVs), we did not observe significant docked vesicle population on the surface of PEG-SLB (Figure 4Bii). Furthermore, we did not find any population when we did not treat SLB and DPPC-ERG SUVs systems with KMP-11 (Figure EV8B). Using co-localization, we found that KMP-11 molecules were present between SLB and DPPC-ERG SUVs (Figure 4Bi, Figure EV8C and D). Next, we estimated the mean diffusion coefficients of the DiD labeled DPPC-ERG vesicles bound onto the PEG SLB through KMP-11 bridge by pulling all particle trajectories (Figure 4C) using track mate module in image J. Interestingly, our data show that most of the particles possess mobilities $< 2 \mu\text{m}^2/\text{sec}$ which is comparable to that reported for peripheral membrane proteins ($0.8\text{-}2 \mu\text{m}^2/\text{sec}$) and transmembrane proteins ($0.02\text{-}0.2 \mu\text{m}^2/\text{sec}$) (Gambin et al, 2006; Vasquez et al, 2014; Ziemba & Falke, 2013) (Figure 4C,D)." The description of this results sounds very confusing to me. I believe that this figure is very important and deserve a better description.

Reply: We agree with the reviewer. We have modified this portion and described in a better way in the revised manuscript and highlighted. Additionally we modified the scheme (Figure 4A) for better understanding.

Please refer to the text

"To provide further support for the bridging between LD and $M\Phi$ and the involvement of KMP-11, using the membrane mimics we developed a single vesicle imaging method using TIRF microscopy within a microfluidic chamber⁵³. For this measurement, we created a PEG cushioned supported bilayer platform inside the chamber. The supported bilayer is comprised of DPPC-CHOL: PEG5000 PE (0.5%). We then introduced Alexa488-labeled-r-KMP-11 and subsequently added DiD-labeled DPPC-ERG SUVs onto the supported lipid bilayer (SLB) channel (Figure 4A). After 5-minute incubation, when we gently washed the channel to remove unbound DPPC-ERG SUVs, we observed a considerable number of bound SUVs on the PEG cushioned SLB surface (Figure 4B i, Figure EV5A, and movie 1). In contrast, when we performed the same experiment using DiD-labeled DPPC-CHOL SUVs (and not DPPC-ERG SUVs), we did not observe any significant population of docked vesicles on the surface of the PEG-SLB (Figure 4B ii). Furthermore, no bound population was observed when we did not add r-KMP-11 (Figure EV5B). By employing co-localization, we found that r-KMP-11 molecules were present between the SLB and DPPC-ERG SUVs (Figure 4Bi, Figure EV5C D and E). We then estimated the mean diffusion coefficients of the DiD-labeled DPPC-ERG vesicles bound to the PEG SLB through the r-KMP-11 bridge by analyzing all particle trajectories (Figure 4C). We found that most of the particles exhibited mobilities $< 2 \mu\text{m}^2/\text{sec}$, which was comparable to the reported range for peripheral membrane proteins ($0.8\text{-}2 \mu\text{m}^2/\text{sec}$) and transmembrane proteins ($0.02\text{-}0.2 \mu\text{m}^2/\text{sec}$)⁵⁴ (Figure 4C, D). We inferred from these measurements that KMP-11 can strongly bridge two membranes containing different sterol molecules. This inter-membrane bridging property of r-KMP-11 was also supported by dynamic light scattering data (Figure EV5Fi,ii). We then investigated the attachment of WT_LD and LD_KMP-11_KO and LD_KMP-11_KO/KMP-11 complemented on DPPC-CHOL SLB,

which showed remarkable adhesion of WT_LD and LD_KMP-11_KO/KMP-11 complemented on SLB but not for LD_KMP-11_KO (Figure EV5 Gi,ii,iii).”

Comment 12: "CHOL influences the structural properties of biological membranes and is central to the organization, dynamics, function, and sorting of lipid bilayers in vivo(Simons &Ikonen, 2000). As a result, CHOL has become an attractive target for many pathogens via which they can influence host cell dynamics. Several pathogens have developed ingenious ways to use CHOL towards recognizing and interacting with host cell membranes. Host cell CHOL accumulates at parasitophorous vacuoles of several intracellular pathogens. At later stages of infection, the Salmonella-containing vacuole (SCV) contains up to 30% of the cellular cholesterol pool in both epithelial cells and macrophages (Catron et al, 2002). CHOL may also be used as a nutrient, as was recently shown for mycobacteria (Pandey &Sasseti, 2008). CHOL may influence the interaction of pathogens containing vacuoles with other cellular organelles, which depends on membrane-trafficking pathways(Haas, 2007). Various intracellular trafficking pathways of eukaryotic cells are sensitive to CHOL, including ER to Golgi transport (Ridsdale et al, 2006), intra-Golgi transport(Stüven et al, 2003), endosomal transport (Simons & Gruenberg, 2000), and phagosomal maturation (Huynh et al, 2008). CHOL levels have been shown to affect 21 intracellular trafficking in various ways, including the recruitment of proteins and lipids to the pathogen-containing vacuole."

Cell invasion and intracellular life are different instances. Promastigotes exist only for a few hours after inoculation in the host dermis. Infection is amplified by the amastigote forms of the parasite, which reside and multiply inside parasitophorous vacuoles. However, this work seemed to put the focus on the onset of infection - in this context and for this reason this whole part sounds strange to me.

Reply: We agree with the reviewer’s comment. We have modified and rewritten the discussion section by putting focus only on the onset of infection, cholesterol quenching and membrane phase transition. Please refer to the discussion section in the revised manuscript.

Comment 13: The discussion just above leads me to some conceptual issues. Leishmania infection is still a black box. For example, macrophages seem not to be the first cells to be infected. Peter and cols in 2008 showed that after inoculation by the bite of the vector, promastigotes are mostly phagocytosed by neutrophils, and not macrophages. Macrophages seem to be infected by the ingestion of apoptotic bodies of previously infected cells, a mechanism named the "Trojan Horse Hypothesis". On the other hand, the infectious process within the mammalian host and the disease itself are promoted by the replication and infection of new cells by amastigote forms.

Reply: We agree with the conceptual dilemma the reviewer is trying to point out. We would also like to emphasize that while neutrophils are the primary phagocytes, some extracellular LD promastigotes can directly infect macrophages. This direct infection is also significant as macrophages are the primary host cells where promastigotes transform into amastigotes, the replicative form of the parasite (*J. Immunol.* 172, 4454–4462.). It is also to be noted, within neutrophils, *L. donovani* promastigotes likely remain in their promastigote form without converting into amastigotes probably owing to the impaired phagosome acidification and reduced ROS production in neutrophils, which are essential for the promastigote-to-amastigote conversion (Infect

Immun. 2021 Jun 16;89(7):e0000921.doi: 10.1128/IAI.00009-21. Epub 2021 Jun 16.). Thus, there is a possibility that some promastigotes will lyse neutrophils, becoming extracellular again and infect new macrophages (J Immunol. 2010 Oct 1;185(7):4319-27.doi: 10.4049/jimmunol.1000893. Epub 2010 Sep 8.). We also completely agree that the infectious process within the mammalian host mediated by amastigotes is out of the scope of this work and we have mentioned this in the discussion of the revised manuscript. The included sentence "It should be mentioned here although KMP-11 appears to play a critical role in initial macrophage infection, its role in amastigote infection process within mammalian host still needs to be validated."

Comment 14: "CHOL influences the structural properties of biological membranes and is central to the organization, dynamics, function, and sorting of lipid bilayers *in vivo* (Simons & Ikonen, 2000). As a result, CHOL has become an attractive target for many pathogens via which they can influence host cell dynamics. Several pathogens have developed ingenious ways to use CHOL towards recognizing and interacting with host cell membranes. Host cell CHOL accumulates at parasitophorous vacuoles of several intracellular pathogens. At later stages of infection, the Salmonella-containing vacuole (SCV) contains up to 30% of the cellular cholesterol pool in both epithelial cells and macrophages (Catron et al, 2002). CHOL may also be used as a nutrient, as was recently shown for mycobacteria (Pandey & Sasseti, 2008). CHOL may influence the interaction of pathogens containing vacuoles with other cellular organelles, which depends on membrane-trafficking pathways (Haas, 2007). Various intracellular trafficking pathways of eukaryotic cells are sensitive to CHOL, including ER to Golgi transport (Ridsdale et al, 2006), intra-Golgi transport (Stüven et al, 2003), endosomal transport (Simons & Gruenberg, 2000), and phagosomal maturation (Huynh et al, 2008). CHOL levels have been shown to affect intracellular trafficking in various ways, including the recruitment of proteins and lipids to the pathogen-containing vacuole.

Different pathogens have developed smart mechanisms of transferring CHOL from the host. For example, CHOL in the membranes of *Helicobacter* spp. (Hirai et al, 1995), *S. aureus* (Haque et al, 1995), *Anaplasma phagocytophilum* (Lin & Rikihisa, 2003) and *Chlamydia* EB (elementary body, infectious form) and RB (reticulate body, a vegetative form) membranes (Wylie et al, 1997) is of host origin. A sterol-binding protein in *Toxoplasma* has been identified that may optimize pathogen handling of host cell derived CHOL (Lige et al, 2009). It has been found that *Toxoplasma* contains an ancestral D-bifunctional protein containing two sterol-carrier protein-2 domains responsible for lipid uptake and trafficking. Recently it has been demonstrated that Mce1 functions as a fatty acid transporter in MTB (*Mycobacterium tuberculosis*) and it is determined that facilitating CHOL and fatty acid import via Rv3723/LucA is required for full bacterial virulence *in vivo* (Nazarova et al, 2017). Apart from the bacteria and protozoan pathogens, viruses also exploit the host lipidome by extracting CHOL from the host membrane thus modulating the property of membranes. It has been shown recently that SARS-COV-2 S protein can quench CHOL from the host and it is one of the most important parts of infection (Correa et al, 2021). CHOL scavenging from the host membrane results in increasing fluidity of the host membrane promoting defective antigen presentation promoting survival of intracellular pathogens. Although *Leishmania* infection has been reported to quench CHOL from the host membrane further increasing the membrane fluidity, the mechanism of CHOL transport was not known, neither the LD-specific proteins which could drive this transport were identified. In this study we show that *Leishmania donovani*, just like

many other intracellular pathogens like Mycobacterium and SARS-COV-2 is equipped with a surface protein for CHOL transport from the interacting host, suggesting that this might be a favored mechanism shared among several groups of intracellular pathogens."

These first paragraphs presented in the discussion seem more appropriate for an introduction.

Reply: We strongly appreciate the reviewer's suggestion. We have now moved some part of the discussion section to the introduction, and we did necessary changes to make the manuscript more focused. Kindly refer to the introduction part in the revised manuscript.

Comment 15: "Further it has been also reported recently that Leishmania parasites can infect non phagocytic cells by inducing changes in host cytoskeleton, mechanism that is distinct from phagocytosis and this invasion process involves in subversion of host cell functions"Cell invasion of non-phagocytic cells by Leishmania does not involve host cell cytoskeleton, actually infection seems to be improved when the cytoskeleton is dismantled.

Reply: We included this sentence just to point out that there is distinct mechanism other than phagocytosis which do account for invasion of leishmania parasite into host cell. KMP-11 also promotes depolymerization of host cytoskeleton as we have shown in the earlier version. However, we have removed this section as already mentioned above. Hence this sentence has also been removed.

Referee #2:

This study provides evidence that the Leishmania donovani KMP-11 protein is involved in cholesterol uptake from infected macrophages. Previous studies have shown that KMP-11 is a small, amphipathic protein that is expressed in the plasma membrane and/or associated with subpellicular microtubule of all stages of Leishmania (and several other kinetoplastid parasites). While numerous studies have shown that KMP-11 is highly immunogenic and potential vaccine candidate, its function has remained enigmatic. In this study, the authors show that targeted deletion of KMP-11 in L. donovani promastigotes leads to a dramatic decrease in cell size/motility which is associated with decreased promastigote attachment and uptake by macrophages. Subsequent studies suggest that addition of KMP-11 directly to macrophages leads to a decrease in host cell actin polymerization, and increased phagocytosis. The authors then utilize a variety of biophysical approaches to show that KMP-11 interacts with saturated phospholipids and sterols in small unilamella vesicles, allowing it to bridge opposing synthetic membrane bilayers and mediate lipid transfer. The authors bring these different lines of evidence together to propose a model in which surface exposed KMP-11 in the parasite membrane bridges intercalates into proximal host cell membranes, forming a junction zone that leads to changes in membrane fluidity and local depletion of cholesterol in the host membrane and transfer to the parasite membrane. Overall, the studies on the role of KMP-11 on model membranes/liposomes are robust and support a role for KMP-11 in modulating membrane fluidity. However, the link between these biophysical studies and loss of parasite virulence/macrophage function and lipid transport is more tenuous and indirect. The physiological relevance of these in vitro observations is therefore unclear. These points are articulated in more detail below.

Reply: We are grateful to the reviewer for the valuable comments. We revised the manuscript keeping these concerns in consideration.

We agree that this manuscript has two broad aspects, namely (a) the observation and (b) the mechanism, and we needed to strengthen the link between these two. We have modified our manuscript extensively and added new experiments and analyses to achieve this. We believe that our replies to the specific concerns would strengthen this point.

1. For (a) the observation aspect, the parameters we measured (for example, the membrane fluidity, cholesterol depletion and transport etc) are all carried out using RAW or primary macrophages. To address the physiological relevance issue more strongly, we have now included a direct quantification of cholesterol quenching potential of KMP-11 in LD infected macrophages (Figure 1Ei,ii in revised manuscript and Figure R1) convincingly establishing this link. Moreover, LD infections in presence of KMP-11 antibody and exogenous r-KMP-11 conclusively suggested KMP-11 as a principal factor initiating the process of infection (Figure 1F and 1G). In parallel, by performing infection with different LD lines e.g. WT_LD, LD_KMP-11_KO and LD_KMP-11_KO/KMP-11 complemented lines and LD>P200 parasite lines along with biochemical complementation with different levels of exogenous r-KMP-11, we have now conclusively shown that KMP-11 is essential for attachment, macrophage plasma membrane fluidity alteration and cholesterol depletion (Figure 1E, F, G, Figure 2F-H, Figure 5. Figure EV3F,G). We have shown comprehensively that KMP-11 increases the fluidity in macrophage membrane which can eventually relate to KMP-11 driven attachment and infection of macrophages by LD parasites (Figure 1C-G, Figure 2F-H and Figure EV3F,G)

2. However, for (b) the mechanism aspect, we needed to use model systems as some of the biophysical experiments were not possible using primary macrophages e.g. determination of the lipid specificity of KMP-11, the underlying mechanism behind membrane Lo/Ld transition, quantitative analysis of the conformational states of lipid molecule (trans/gauche conformational variation), flexibility and mobility of lipid molecules in KMP-11 bound condition etc. Furthermore, to determine the binding regions and the motif responsible for lipid fluidity change, we needed to use an ensemble of biophysical techniques using model membrane systems (Peptide experiments and mutational approaches). However, as much it is possible, we tried to carry bridging validation experiments to minimize this gap.

Major points

Comment 1: As noted by the authors, genetic deletion of KMP-11 leads to a contraction in flagellum length and loss of motility that, by itself, may explain the reduced binding and uptake of the KMP-11 KO line by macrophages. It therefore remains uncertain whether the membrane-membrane bridging and cholesterol transport properties of KMP-11 are physiologically relevant in vivo. This is particularly pertinent, as *L. donovani* promastigotes are coated by a thick glycocalyx composed of LPG and GP63 that prevent host proteins (e.g. C1/C3 complement protein) from accessing the parasite plasma membrane, and presumably prevent small parasite proteins, such as KMP-11 from bridging the gap to either macrophage plasma membrane or internal vacuolar membranes.

Reply: These are important questions. Since the reviewer's concerns have two different aspects (cholesterol transport and membrane-membrane bridging), we have addressed them separately.

First, we agree with the reviewer that contracted flagellum of KMP-11_KO LD may itself affect the macrophage binding. This is why, we added an experiment using KMP-11 antibody. KMP-11 antibody pre-treated WT_LD shows significantly reduced infectivity which strengthens the role of KMP-11 on the onset of infection (Figure 1F, G). Additionally, our LD >P200 (WT_LD which was repeatedly passed through in vitro culture for more than 200 passages; without transforming through animal) also showed significant drop in attachment and invasion process, which can be complemented by addition of exogenous KMP-11 (Figure 2G, H). Interestingly, although there is no appreciable changes in parasite morphology in this case, we found that KMP-11 expression was notably low (Figure EV3E in extended view file).

Regarding specific concerns of:

(a) Cholesterol transport:

Please note that, in the revised manuscript, the cholesterol transport between the macrophage membrane and the LD has been shown not only in the synthetic membrane, but also using direct CHOL quantification experiments in the primary infected macrophages (Figure R1, Figure 1E in the revised manuscript). We employed four different approaches to understand cholesterol transport process, which are- (i) Direct cholesterol quantification by isolating macrophage plasma membrane (Figure R1, Figure 1E in the revised manuscript); (ii). NBD CHOL transfer from macrophage to LD (Figure 5F); (iii) Lipid raft disruption (Figure EV2D,E) and (iv). *in vitro* FRET assay of lipid transfer (Figure 5A-E). Through our direct cholesterol quantification of the macrophage plasma membrane through membrane isolation after different treatments, we conclusively say that KMP-11 is responsible for cholesterol depletion in macrophage membrane (Figure R1 and Figure 1E, Figure EV3G)

It may also be noted that the cholesterol transport between the host and the parasite has also been established by other groups. LD infection leading to the removal of macrophage membrane cholesterol has been shown before (for example, *Microbiologyopen*. 2017 Aug;6(4):e00469. doi: 10.1002/mbo3.469. Epub 2017 Mar 27.; *J Lipid Res*. 2012 Dec;53(12):2560-72. doi: 10.1194/jlr.M026914. Epub 2012 Oct 10). There is also a clinical observation for LD-infected patients and animals (*Ann Trop Med Parasitol*, 2011 Apr;105(3):267-71. doi: 10.1179/136485911X12899838683566., *Biochimie* . 2020 Aug;175:13-22. doi: 10.1016/j.biochi.2020.04.024. Epub 2020 May 18.)

(b) Membrane-membrane bridging:

Previously published literature has reported that inter membrane distance between parasite and macrophages during invasion is approximately 10 nm (*Journal of cell science* 133: jcs232488; *Microbes and infection* 13: 1033-1044). In this manuscript we have observed that KMP-11 can form oligomers of nearly 10 nm size, and hence, these oligomers induced bridging is a strong possibility.

Importantly, other groups have reported a significant organizational change in glycocalyx component during the invasion of *Leishmania* parasites (*Frontiers in cellular and infection microbiology* 4: 193) enhancing the possibility that a small

membrane protein like KMP-11 can get exposed during the process of infection. Our data on solubility (western blot) and confocal microscopy did support this possibility showing that a significant amount of KMP-11 is retained on LD surface (Figure EV 1A-1C). We have addressed this further in a physiologically relevant system using Fluorescence resonance energy transfer (FRET) experiment. We used GFP labeled KMP-11 on LD surface as donor and Rhodamine DHPE (a strong membrane lipophilic dye) on macrophages membrane as acceptor to directly validate if KMP-11 is interacting with host membrane during attachment (Figure R4, Figure EV6). We found significant FRET between GFP and Rhodamine with a donor-acceptor distance ranges between 20-60Å which further suggests that KMP11 and macrophage membrane can interact strongly with one another. In contrast we did not see significant FRET when we pretreated KMP-11 GFP LD with anti KMP-11 antibody. Thus, our FRET data clearly pointed towards a strong interaction between KMP-11 of LD and macrophage membrane.

Figure for referee with unpublished data and its description has been removed upon request by the authors.

Figure for referee with unpublished data and its description has been removed upon request by the authors.

Comment 2:

Of related concern, the main evidence linking KMP-11 to the regulation host actin polymerization, phagocytosis and membrane Lo/So/Ld phase changes in vivo is derived from experiments using exogenous KMP-11 delivered at quite high concentrations (50 μ M). However, there is no evidence that KMP-11 is shed from invading promastigotes (Fig 5G) or would reach the concentration in used in these experiments. It therefore remains possible that exogenous addition of KMP-11 has an artefactual effect on macrophage actin and sterol distribution that is not replicated by

parasite pools of KMP-11. Have the authors tried to 'biochemically' complement the KMP-11 KO line with different levels of exogenous KMP-11 and see if they can reconstitute some of the effects on Mø with a more physiologically relevant 'delivery' system?

Reply:

We thank the reviewer for the valued comment and suggestion. We think there are two parts of this comment which should be addressed separately.

First, regarding whether concentration of KMP-11 used can justify physiological relevant concentration?

We brainstormed extensively to arrive at the working concentration of r-KMP-11 for our experiments and decided to use a KMP-11 concentration for the exogenous addition which would replicate the effect observed in LD infection. We request you to note that we have in fact decided the optimum dosage of exogenous r-KMP-11 from the fluidity measurements induced by LD infection (and not the other way). Please note that, we did see a large increase in the fluidity in macrophage membrane (Figure 1C) at a much lower r-KMP-11 concentration (~500nM) of r-KMP-11 (Figure 1C, about 1.5 fold), and there was a dose dependence, although the effect saturated at a higher 50 μ M concentration. In addition, our calculation based on the P/L ratio and μ M protein concentration shows a significant availability of KMP-11 in LD (see expanded view file) to enable a productive infection.

To address if the use of 50 μ M KMP-11 generates any issues with the cellular health or other possible artefactual effects on macrophages, we performed different experiments. First, we studied the apoptosis of macrophage after treatment with 50 μ M r-KMP-11 by performing Annexin V staining (Invitrogen) in r-KMP-11 (50 μ M) treated MΦs for 12 hrs. MΦs treated with 50 μ M r-KMP-11 for 12hrs resulted in similar percent of apoptotic population (6.23%) like untreated (8.44%) or LD infected control (4.74%) (Figure EV2B) thus KMP-11 has no role in inducing apoptosis in the host cell membrane (Figure R6) at 50 μ M concentration and its role in membrane Lo/So/Ld phase changes seems very specific.

Figure for referee with unpublished data and its description has been removed upon request by the authors.

Figure for referee with unpublished data and its description has been removed upon request by the authors.

For the second part of the comment on biochemical complementation we would like to mention

1. Apart from successful second copy complementation of KMP-11 in KMP-11_KO_LD line, we have now also performed biochemical complementation of LD_KMP-11_KO line with r-KMP-11 as suggested by the reviewer in context of macrophage infection (Figure R6, Figure 2H). Our result showed that r-KMP-11 can partially restore macrophage infection for both KMP-11_KO and LD>P200 (with significantly reduced KMP-11 expression) LD line.

Figure for referee with unpublished data and its description has been removed upon request by the authors.

Comment 3: While there is no evidence (presented here or in the literature) that Leishmania promastigotes form stable/transient tight junctions with host membranes, it is very clear that intracellular amastigote stage do form tight junctions. Have the authors looked to see if KMP-11 is associated with the conspicuous tight junctions that form between the posterior end of amastigotes and the phagolysosomal membrane of infected macrophages, and are predicted to be involved in lipid transfer? Amastigotes also have a reduced glycocalyx which may increase the ability of amastigote KMP-11 to integrate into the bilayer of the macrophage PVM.

Reply: This is an important question and we have addressed this specifically in the context of Leishmania promastigote infection.

KMP-11 can bind with macrophage membrane has been shown by FRET measurement and some additional experiments (as discussed in the response against comment 1).

However, we would like to mention that although KMP-11 is expressed in all the life stages of LD, and we do agree with the reviewer that it seems very plausible that KMP-11 might play a similar role in the context of amastigote invasion, we have not investigated that. Amastigote infection in mammalian host itself requires special attention and is beyond the scope of this work which primarily focuses on the role of lipid transfer during promastigote infection.

Comment 4: Fig 2 A. The authors need to show that the KMP-11 KO line lacks the KMP-11 gene. They currently show that one allele has been replaced with a bleomycin cassette, but as these parasites are diploid, the KO line could still retain a wildtype copy of KMP-11. While a Western blot showing protein levels is shown in Fig2B, a faint band corresponding to Ld_KMP-11 is evident in the CI_2 clone, raising the possibility that the second allele has not been removed in a subset of the population. Ideally, both alleles would be sequentially deleted using two drug selection cassettes and/or the loss of all copies of the KMP-11 gene demonstrated.

Reply: We would like to point out that gRNA and Cas9 co-expression vectors has been reported to successfully delete multigene family in Leishmania spp by simultaneously targeting sites even from two different chromosomes Ref mBio. 2015 Jul 21;6(4):e00861.doi: 10.1128/mBio.00861-15., mSphere. 2017 Jan 18;2(1):e00340-16.doi:10.1128/mSphere.00340-16.). This was not previously possible with the traditional homologues recombination method, which requires multiple round of sequential transfection for replacement of individual allele (mSphere. 2017 Jan 18;2(1):e00340-16.doi:10.1128/mSphere.00340-16.). In fact, double-gRNA expression vector that was used in this study is a well-established method in successfully replacing both the alleles of target gene with single transfection (mBio. 2015 Jul 21;6(4):e00861.doi: 10.1128/mBio.00861-15.). It may be noted, that we used the same approach to Knock out both the allele of KMP-11.

We agree of the existence of a very negligible faint band in clonal population 2 after initial round of selection (Figure 2A ii in the revised manuscript). However, there was absolutely no existence of it under continuous drug selection (Figure 1D ii in the revised manuscript). We would clearly like to mention that we have continued further experiments of KMP-11 KO lines using clonal population where there was absolutely no detection of KMP-11 at protein level.

Comment 5: The subcellular localization of KMP-11 varies significantly in different images, from localized puncta (Fig 5G), to both intracellular/cell body staining (EV1) to strong flagella staining (Fig EV11). While some of these differences could reflect the use of antibodies to native protein versus localization of the GFP-fusion protein, these disparities need to be better defined. The extent to which KMP-11 is associated with the parasite cytoskeleton, as previously proposed needs to be better defined (the current Western blots after carbonate/detergent extraction in Fig EV1 are of poor quality).

Reply: We would like to clarify that in case of Figure EV1B (also Figure EV1B in revised manuscript), staining was performed on extracellular LD with and without permeabilization, while Fig 5G (Figure 5F in the revised Manuscript) specifically represent a particular field of LD promastigotes interacting with the surface of macrophages. We believe that punctuated staining of KMP-11 is observed under specific experimental condition when LDs are attached on macrophage membrane. However, it can be appreciated in all these cases Figure EV1B, Figure 5F and Figure 6 (earlier Figure EV11) where anti-KMP-11 antibody was used, KMP-11 positive staining appears on LD membrane, flagellum along with intracellular staining with varying

intensity depending on experimental conditions. Further, whenever KMP-11 has been stained in extracellular LD promastigotes either with anti-KMP-11 antibody (Figure EV1B) or in the case of KMP-11 GFP without KMP-11 antibody (Figure 1E, earlier Figure EV2B), similar subcellular localization was observed as opposed to infecting promastigotes. This clearly indicates that we do have localized KMP-11 correctly under different experimental conditions, and slight difference in staining might be reflecting upon these differential experimental conditions. We thank the reviewer for bringing this to our notice and we have now included this as a comment on the Figure Legend of Figure 5F of the revised manuscript as “Compared to extracellular LD, KMP-11 shows a slightly punctated distribution in LD attached with macrophage membrane”

We would also like to mention that we have replaced the western blot of Figure EV1 with a better representation as suggested by the reviewer.

Comment 6: The functionality of the KMP-11-GFP fusion protein needs to be further validated. Does expression of this protein in the KMP-11 KO line restore virulence/normal size/motility?

Reply: In the Figures 2B, 2C, 2E we have provided images, which clearly show that complementation with KMP-11-GFP restores the size and morphology of LD_KMP-11_KO line. Further Figures 2F, 2G and 2H show that this complementation also resulted in restoration of attachment and infection potential of the LD_KMP-11_KO in macrophages. We would further like to mention that we have included the biochemical complementation data of KMP-11_KO line (Figure 2H) by performing infection in the presence of externally added KMP-11 as suggested by the Reviewer 1.

Comment 7: The effect of KMP-11 on actin polymerization (Phalloidin staining) also varies in different images. In Fig 2F, addition of exogenous KMP-11 leads to very strong loss of Phalloidin staining, while in Fig EV3, the reduction in Phalloidin staining is marginal.

Reply: We agree with this comment of the reviewer. As we have already mentioned that further experiments are required to completely validate the role of KMP-11 on actin polymerization. Also, we wanted to develop a manuscript which is more focussed. As a result, this aspect is now removed.

Minor comments.

P20, para 3. The statement that 'it has been convincingly proven that neither LPG nor gp63 is essential for virulence' is not correct. Disruption of genes involved in LPG synthesis in *L. donovani* and *L. major* result in a strong loss of virulence in both macrophage and mice infections (although the small number of promastigotes that successfully differentiate to amastigotes can subsequently go onto cause an infection).

Reply: We agree with the reviewer. We have rewritten that portion in the revised manuscript. Please refer to the discussion portion in the current manuscript.

Please refer to the text:

Till date only two widely studied *Leishmania* specific molecules have been implicated in establishing *Leishmania* infection. These are zinc-metalloprotease gp63 and lipophosphoglycan (LPG), a complex glycolipid. While gp63 has primarily been implicated in the cleavage and degradation of various host derived kinases and transcription factors⁷⁴, the role of LPG has been suggested in the prevention of complement mediated lysis⁷⁵ during infection process. Interestingly, it has been reported that *L.mexicana* mutants lacking LPG are capable of causing infections⁷⁶. Similarly, it has been shown for LD and *L.major* that, although ablation of LPG and gp63 results in compromised infection, it does not abrogate the invasion process⁷⁶⁻⁷⁹. This observation, along with the fact that both LPG and gp63 have significantly low expression in amastigotes^{80, 81}, clearly indicate that apart from gp63 and LPG other parasite surface molecules can also contribute through invasion of *Leishmania* parasites ^{75, 81}

Dear Krish,

Thank you for submitting your revised manuscript to EMBO Reports and please apologize my delayed response, which is due to recent travel.

As you know, I have sent your revised manuscript back to the referees who had evaluated your manuscript at The EMBO Journal when it was first submitted. As you will see from the reports copied below the referees consider the revised manuscript strengthened and support publication at EMBO Reports. Please address the remaining concerns from Referee #2 by discussing and integrating the alternative scenario of KMP-11 action outlined in point (1), by discussing the limitations mentioned in (2) and by performing the suggested experiment (point 3). Please also provide a point-by-point response.

I list below the general formatting guidelines for EMBO Reports (B) and a few specific comments for your manuscript (A).

A) Specific comments.

- The red and green arrows in figure 2F are difficult to see.
- We need the exact p-values instead of the cut-off values in the figure legends.
- For 'n=' you need to define whether this refers to technical replicates or independent experiments.
- In 'Supporting Information' you write: 'Mathematical exposition of 'Hydrophobic Moment and sequence symmetry-oriented phase transition model', and the calculation of the number of KMP-11 molecules needed for MΦ phase transition have been described.' Could you please specify where this model was described?
- References need to be alphabetical. Please list all authors up to 10, followed by et al.
- Please include a dedicated "Data Availability" section at the end of the Methods (suggested wording: "The [structural coordinates | microarray | mass spectrometry] data from this publication have been deposited to the [name of the database] database [URL] and assigned the identifier [accession | permalink | hashtag]."). Should this not apply, this should still be stated as "This study includes no data deposited in external repositories."
- Please note the requirement of a Reagents and Tools table (point 12).
- We will need all source data. Please see point (8) below.

B) General formatting guidelines. When submitting your revised manuscript, we will require:

2) individual production quality figure files as .eps, .tif, .jpg (one file per figure).

Please download our Figure Preparation Guidelines (figure preparation pdf) from our Author Guidelines pages <https://www.embopress.org/page/journal/14693178/authorguide> for more info on how to prepare your figures.

4) a complete author checklist, which you can download from our author guidelines

(<<https://www.embopress.org/page/journal/14693178/authorguide>>). Please insert information in the checklist that is also reflected in the manuscript. The completed author checklist will also be part of the RPF.

5) Please note that all corresponding authors are required to supply an ORCID ID for their name upon submission of a revised manuscript (<<https://orcid.org/>>). Please find instructions on how to link your ORCID ID to your account in our manuscript tracking system in our Author guidelines

(<<https://www.embopress.org/page/journal/14693178/authorguide#authorshipguidelines>>)

6) We replaced Supplementary Information with Expanded View (EV) Figures and Tables that are collapsible/expandable online. A maximum of 5 EV Figures can be typeset. EV Figures should be cited as 'Figure EV1, Figure EV2' etc... in the text and their respective legends should be included in the main text after the legends of regular figures.

- Additional Tables/Datasets should be labeled and referred to as Table EV1, Dataset EV1, etc. Legends have to be provided in

a separate tab in case of .xls files. Alternatively, the legend can be supplied as a separate text file (README) and zipped together with the Table/Dataset file.

7) Please note that a Data Availability section at the end of Materials and Methods is now mandatory. In case you have no data that requires deposition in a public database, please state so instead of refereeing to the database. See also < <https://www.embopress.org/page/journal/14693178/authorguide#dataavailability>>. Please note that the Data Availability Section is restricted to new primary data that are part of this study.

Additional information on source data and instruction on how to label the files are available <<https://www.embopress.org/page/journal/14693178/authorguide#sourcedata>>.

10) Figure legends and data quantification:
The following points must be specified in each figure legend:

- the name of the statistical test used to generate error bars and P values,
- the number (n) of independent experiments (please specify technical or biological replicates) underlying each data point,
- the nature of the bars and error bars (s.d., s.e.m.)

- If the data are obtained from n {less than or equal to} 5, show the individual data points in addition to the SD or SEM.
- If the data are obtained from n {less than or equal to} 2, use scatter blots showing the individual data points.

See also the guidelines for figure legend preparation:
<https://www.embopress.org/page/journal/14693178/authorguide#figureformat>

11) Our journal encourages inclusion of *data citations in the reference list* to directly cite datasets that were re-used and obtained from public databases. Data citations in the article text are distinct from normal bibliographical citations and should directly link to the database records from which the data can be accessed. In the main text, data citations are formatted as follows: "Data ref: Smith et al, 2001" or "Data ref: NCBI Sequence Read Archive PRJNA342805, 2017". In the Reference list, data citations must be labeled with "[DATASET]". A data reference must provide the database name, accession number/identifiers and a resolvable link to the landing page from which the data can be accessed at the end of the reference. Further instructions are available at <<https://www.embopress.org/page/journal/14693178/authorguide#referencesformat>>.

12) All Materials and Methods need to be described in the main text using our 'Structured Methods' format, which is required for all research articles. According to this format, the Methods section includes a Reagents and Tools Table (listing key reagents, experimental models, software and relevant equipment and including their sources and relevant identifiers) followed by a Methods and Protocols section describing the methods using a step-by-step protocol format. The aim is to facilitate adoption of the methodologies across labs. More information on how to adhere to this format as well as a downloadable template (.docx) for the Reagents and Tools Table can be found in our author guidelines:
<https://www.embopress.org/page/journal/14693178/authorguide#structuredmethods>.

An example of a Method paper with Structured Methods can be found here:
<https://www.embopress.org/doi/10.15252/msb.20178071>.

13) As part of the EMBO publication's Transparent Editorial Process, EMBO Reports publishes online a Review Process File to accompany accepted manuscripts. This File will be published in conjunction with your paper and will include the referee reports, your point-by-point response and all pertinent correspondence relating to the manuscript.

Kind regards,

Martina

Referee #1:

As mentioned in my first review I believe this is a good article, with an impressive amount of work. The protein and the process studied are of major interest in the field. I had major concerns about some experiments performed, the lack of important controls and the way the manuscript was organized. However, I really appreciated the revised version and the new data added. Now, I believe this paper deserves publication.

Referee #2:

This study provides evidence that the *Leishmania donovani* KMP-11 protein may facilitate parasite attachment and invasion of host macrophages by forming a membrane 'bridge' between the parasite and macrophage PMs. It is proposed that this protein bridge allows transfer of host cholesterol to invading promastigotes with concomitant perturbation of cholesterol levels in the host membrane. This revised manuscript was previously submitted to EMBO J, and the authors have addressed many of the concerns with the original manuscript. Notably, the manuscript text has been tightened up and further evidence supporting the biological significance of the biophysical studies provided. Overall, this is a very interesting study with broad implication for understanding other host-microbial pathogen interactions where structurally related proteins may regulate lipid transfer between the host and pathogen. A number of issues remain outstanding which the authors could address below.

Comments

1. The authors conclude that KMP-11 forms a protein bridge between the parasite and host plasma membrane, which facilitates the exchange of cholesterol and that this is a primary/sole site of action. An additional/alternative possibility is that most of the KMP-11 is actively shed in the presence of an proximal 'acceptor/host' membrane and that the transferred KMP-11 induces global changes in cholesterol organization in the macrophage PM which facilitate phagocytosis or other processes involved in parasite internalization. This conclusion is supported by the fact that exogenous addition of rKMP-11 to macrophages enhanced promastigote infection. Given that macrophage membrane would be the largest target membrane in these in vitro experiments, it is more likely that 'exogenous' KMP-11 is broadly affecting macrophage membrane, rather than a very focussed region at the point of contact between the parasite and host membrane. Other experiments reported in this paper also support the possibility that KMP-11 alters the fluidity of any membrane it inserts into without requiring an opposing membrane (e.g no need for the bridge). This broader hypothesis also gets around the problem of how a small peptide like KMP-11 can span the extended glycocalyxes of both promastigote and host cell (~60 nm).

2. The authors can still not discount the possibility that other factors contribute to loss of Δ KMP-11 promastigote uptake by macrophages, such as contraction of the flagellum in the mutant and/or inability of the mutant to transform into infective metacyclics. With regard to the latter, the authors utilize light scattering to identify the number of metacyclic promastigotes in their culture, but this approach is invalid given the dramatic changes in size and shape of the Δ KMP-1 mutant. Other measures of metacyclogenesis, such as changes in LPG structure/epitopes or cell density would also not be valid - the only assay that is valid is increased virulence and that is not appropriate when looking at a mutant line. Acknowledgement that it is not possible to confirm whether the mutant is able to differentiate to metacyclics is warranted.

3. Finally, the authors suggest that analysis of the role of KMP-11 in amastigote infection is beyond the scope of this study. However, repeating the infection assays with WT and Δ KMP-11 axenic amastigotes (the major stages responsible for perpetuating infections) as well as test for the effect of exogenous rKMP-11 on amastigote binding and internalization would be straightforward and would significantly add to the impact and relevance of the study complement.

Response to reviewer's comments

Referee #1:

As mentioned in my first review I believe this is a good article, with an impressive amount of work. The protein and the process studied are of major interest in the field. I had major concerns about some experiments performed, the lack of important controls and the way the manuscript was organized. However, I really appreciated the revised version and the new data added. Now, I believe this paper deserves publication.

Response: We are grateful to Referee #1 for the appreciation of our work.

Referee #2:

This study provides evidence that the *Leishmania donovani* KMP-11 protein may facilitate parasite attachment and invasion of host macrophages by forming a membrane 'bridge' between the parasite and macrophage PMs. It is proposed that this protein bridge allows transfer of host cholesterol to invading promastigotes with concomitant perturbation of cholesterol levels in the host membrane. This revised manuscript was previously submitted to EMBO J, and the authors have addressed many of the concerns with the original manuscript. Notably, the manuscript text has been tightened up and further evidence supporting the biological significance of the biophysical studies provided. Overall, this is a very interesting study with broad implication for understanding other host-microbial pathogen interactions where structurally related proteins may regulate lipid transfer between the host and pathogen. A number of issues remain outstanding which the authors could address below.

Comments

1. The authors conclude that KMP-11 forms a protein bridge between the parasite and host plasma membrane, which facilitates the exchange of cholesterol and that this is a primary/sole site of action. An additional/alternative possibility is that most of the KMP-11 is actively shed in the presence of an proximal 'acceptor/host' membrane and that the transferred KMP-11 induces global changes in cholesterol organization in the macrophage PM which facilitate phagocytosis or other processes involved in parasite internalization. This conclusion is supported by the fact that exogenous addition of rKMP-11 to macrophages enhanced promastigote infection. Given that macrophage membrane would be the largest target membrane in these in vitro experiments, it is more likely that 'exogenous' KMP-11 is broadly affecting macrophage membrane, rather than a very focussed region at the point of contact between the parasite and host membrane. Other experiments reported in this paper also support the possibility that KMP-11 alters the fluidity of any membrane it inserts into without requiring an opposing membrane (e.g no need for the bridge). This broader hypothesis also gets around the problem of how a small peptide like KMP-11 can span the extended glycocalyxes of both promastigote and host cell (~60 nm).

Response: We agree with the reviewer. There could be an alternative mechanism of KMP-11 functioning during *Leishmania* promastigote infection. Although, we have

biophysical evidence suggesting the existence of a protein bridge between LD and macrophage membrane, we could not completely negate the possibility of active shedding of KMP-11 in proximity of host membrane which could also contribute to the process of promastigote infection. We have included this statement in the discussion section of the revised MS and highlighted.

Please refer to the text

“Although, our results showed that KMP-11 forms a bridge between LD and host plasma membrane, we cannot negate the possibility that active shedding of KMP-11 in close proximity of the host membrane may be an alternative strategy which facilitates LD invasion into the MΦs. The increase in the number of invading LD in the presence of exogenous KMP-11 suggests that KMP-11 might be acting across a broader area of the MΦ membrane, potentially altering cholesterol levels throughout the membrane rather than just at specific points of contact between the parasite and the host membrane. However, additional experiments are required to specifically conclude on the global versus local effects of KMP-11 during the process of promastigote infection.”

2. The authors can still not discount the possibility that other factors contribute to loss of ΔKMP-11 promastigote uptake by macrophages, such as contraction of the flagellum in the mutant and/or inability of the mutant to transform into infective metacyclics. With regard to the latter, the authors utilize light scattering to identify the number of metacyclic promastigotes in their culture, but this approach is invalid given the dramatic changes in size and shape of the ΔKMP-1 mutant. Other measures of metacyclogenesis, such as changes in LPG structure/epitopes or cell density would also not be valid - the only assay that is valid is increased virulence and that is not appropriate when looking at a mutant line. Acknowledgement that it is not possible to confirm whether the mutant is able to differentiate to metacyclics is warranted.

Response: We agree with the reviewer on the point that dramatic change in the size and morphology of LD_KMP-11_KO lines with severely reduced flagella could also contribute to loss of infection exhibited by LD_KMP-11_KO promastigotes. It is mentioned in the discussion section of the MS as follows:

“The absence of KMP-11 severely reduces parasite size and the length of the flagella. Since it is well known that the persistent flagellar activity leads to initial attachment (stage 1, Figure 8) with host cell membrane, the stunted flagella probably explain defective attachment and subsequent host cell invasion as observed with LD_KMP-11_KO parasites.”

We would also like to thank the reviewer for understanding that this dramatic change in size of LD_KMP-11_KO lines is an additional constrain for truly evaluating its metacyclogenesis potential. As suggested by the reviewer we have acknowledged this point in the result section of the revised MS.

Please refer to the text

“However, it should be mentioned that as LD_KMP-11_KO lines exhibit a dramatic reduction in their size and morphology, the light scattering based determination might

not provide an accurate measurement of percent metacyclics for LD_KMP-11_KO promastigotes.”

3. Finally, the authors suggest that analysis of the role of KMP-11 in amastigote infection is beyond the scope of this study. However, repeating the infection assays with WT and Δ KMP-11 axenic amastigotes (the major stages responsible for perpetuating infections) as well as test for the effect of exogenous rKMP-11 on amastigote binding and internalization would be straightforward and would significantly add to the impact and relevance of the study complement.

Response: We would like to clarify that we have already tried to generate axenic amastigotes for LD_KMP-11_KO lines to perform a comparative infection assay. However, this has been unsuccessful so far. This is probably because KMP-11 depletion severely alters the morphology of the LD promastigotes, and hence, they presumably fail to transform into amastigotes. Typically, *Leishmania* promastigotes are approximately two to three times larger in size than its amastigotes. We feel it is also possible that we are not able to detect the axenic amastigotes of LD_KMP-11_KO lines by staining due to its severely reduced size, and as they might not represent a proper amastigote morphology. Hence, we could not proceed with this experiment as in this case we have to infect the macrophages blindly with any valid proof that axenic amastigotes were generated for LD_KMP-11_KO lines. So, we feel that even if we see a severe defect with axenic amastigote infection for LD_KMP-11_KO lines we will not be able to definitively conclude whether the failure is due to a lack of amastigote generation or an inability to invade. Therefore, we believe it might be challenging to make a convincing claim in this scenario.

We have discussed this possibility in the discussion section of the revised MS. Please refer to the text

“It should be mentioned here although KMP-11 appears to play a critical role in initial M Φ infection; its role in amastigote infection process within mammalian host still needs to be validated. We would also like to mention that our attempts to generate axenic amastigotes for LD_KMP-11_KO lines were unsuccessful either due to inability of KMP-11 depleted promastigotes to get transformed into axenic amastigotes or due to our inability to properly detect KMP-11 depleted axenic amastigotes owing to their severe reduction in size and possibly altered morphology. Hence, it was further not possible for us to determine the role of KMP-11 in the process of amastigote infection.”

Manuscript number: EMBOR-2024-60001V2

Title: Mechanism of host cell invasion by Leishmania through KMP-11 mediated cholesterol-transport

Author(s): Achinta Sannigrahi, Souradeepa Ghosh, Supratim Pradhan, Pulak Jana, Junaid Jawed, Subrata Majumdar, Syamal Roy, Sanat Karmakar, Budhaditya Mukherjee, and Krishnananda Chattopadhyay

Dear Krish,

Thank you for your patience while we have reviewed your revised manuscript from the editorial side. This has revealed a few points that still need your attention. I am therefore writing with an 'accept in principle' decision, which means that I will be happy to accept your manuscript for publication once a few minor issues/corrections have been addressed, as follows.

- 1) Please reduce the number of keywords to 5.
- 2) Please place the Competing Interests statement after the Acknowledgments
- 3) We noticed an author name discrepancy - Junaid Jibrán Jawed in the manuscript vs. Junaid Jawed in the online submission system. Please correct this.
- 4) The funding information in the manuscript must match that in the online manuscript tracking system. In the latter the following funders are currently missing:
 - SERB (PDF/2020/000678), Govt. of India
 - Bose Fellowship (Number: SB/S2/JCB-65/2014)
 - DBT-funded research project (BT/PR8475/BRB/10/1248/2013)
 - GATE fellowship
 - CSIR-NET and PMRF fellowship
 - CSIR network project grant HOPE
- 5) Please add a callout for Fig. 8A.
- 6) Movies: the title, source file name and manuscript callout should be Movie EV1; a legend is also needed, it can be provided as a readme.txt file and it should be zipped up with the movie file and uploaded as one folder
- 7) Please remove the Reagents and Tools table from the manuscript and upload it as separate file (file type Reagent table).
- 8) Please check the format of your reference list. Some of the references lack the Journal name, such as Cavalcante-Costa 2019, or the year such as the 2nd Cavalcante-Costa reference, some contain information on Epub date and the DOI, such as Beneke T 2017. Please see our Author Guidelines for the required formatting of the reference list (<https://www.embopress.org/page/journal/14693178/authorguide#referencesformat>)
- 9) Please proof-read the Appendix text, also the first part on the evaluation of the P/L ratio, which seems a bit difficult to read and might benefit from a native speaker's input. I also noticed several typos in the figure or table legends. The legend of Appendix Table S1 lacks spaces before the brackets; Appendix Table S2 lacks a space after "." etc. For Appendix Table S6 I would suggest increasing the width of the first column to avoid having the word 'and' separated in two lines.
- 10) Appendix Fig. S3: please add scale bars.
- 11) The Appendix text is called "Appendix material" in the manuscript. Please be consistent in the naming, "Appendix methods" might be most appropriate.
- 12) Methods, page 30: the mathematical formulas are very small and difficult to read. Please also carefully proof-read the methods part.
- 13) Source data: Please sort the files in subfolders. The source data should be uploaded as one folder per figure and within each folder, the data should be organized as one folder/file per panel.
- 14) The source data for the KMP-11 staining in Figure 5F (uninfected) appears to be missing. Please provide the data for all three channels. (The composite image for 'infected' lacks the label '2' in the source data.)
- 15) Please also provide the source data for Figure EV3B to avoid any ambiguities. The panels appear to be black only.
- 16) Our production/data editors have asked you to clarify several points in the figure legends (see below). Please incorporate

these changes in the manuscript and return the revised file with tracked changes with your final manuscript submission.

A) Statistical test information. Only p-values that are actually shown in the figure panel(s) should (and must) be defined in the legends, all others should be removed from (or added to) the legend. Moreover, we ask for the specification of exact p-values:
- Please note that the exact p values are not provided in the legends of figures 1c-d, f-g; 2g-h; 7j.

B) Data presentation:

- Please note that the scale bar needs to be defined for figure EV 3b.
- Please note that scale bar and its definition are missing for figure EV 3c.
- Please note that the white arrowheads are not defined in the legend of figure 5f(ii). This needs to be rectified.

17) Figure presentation:

- Figure 1D: the column titles do not fit well into the column. I suggest to increase column width and to have a uniform width for all the columns.
- Figure 2 A and D: I suggest to use the same font size for both panels, ideally the one used for (A), which fits better. The scale bar in panel E is too thin and in panel F the scale bar thickness differs between the images.
- Figure 4: it would visually look better if you repositioned the panel in a more aligned manner. As it stands, the panels look a bit scattered over the page. Panel F lacks scale bars for the zoomed images.
- Figure 6B: the scale bars are very thin and not well visible. Panel D lacks scale bars.

18) May I suggest to use the more common abbreviation "L. donovani" instead of LD? LD is commonly used for "lipid droplets" and I find the referral to LD instead of L. donovani irritating.

19) As a standard procedure, we edit the title and abstract of manuscripts to make them more accessible to a general readership. Please find the edited versions below my signature and let me know if you do NOT agree with any of the changes.

20) Finally, EMBO Reports papers are accompanied online by

A) a short (1-2 sentences) summary of the findings and their significance,

B) 2-3 bullet points highlighting key results and

C) a schematic summary figure that provides a sketch of the major findings (not a data image).

Please provide the summary figure as a separate file in PNG or JPG format at a size of 550x300-600 pixels (width x height).

Please note that the size is rather small and that text needs to be readable at the final size. Please send us this information along with the revised manuscript.

Once you have made these minor revisions, please use the following link to submit your corrected manuscript:

Link Not Available

If all remaining corrections have been attended to, you will then receive an official decision letter from the journal accepting your manuscript for publication in the next available issue of EMBO reports. This letter will also include details of the further steps you need to take for the prompt inclusion of your manuscript in our next available issue.

On a different note, I would like to alert you that EMBO Press offers a new format for a video-synopsis of work published with us, which essentially is a short, author-generated film explaining the core findings in hand drawings, and, as we believe, can be very useful to increase visibility of the work. This has proven to offer a nice opportunity for exposure i.p. for the first author(s) of the study. Please see the following link for representative examples and their integration into the article web page:

<https://www.embopress.org/doi/full/10.15252/embj.2019103932>

Thank you for your contribution to EMBO reports.

Kind regards,

Martina

Leishmania protein KMP-11 modulates cholesterol transport and membrane fluidity to facilitate host cell invasion

Abstract:

The first step of successful infection by any intracellular pathogen relies on its ability to invade its host cell membrane. However, the detailed structural and molecular understanding underlying lipid membrane modification during pathogenic invasion remains unclear. In this study, we show that a specific *Leishmania donovani* (LD) protein, KMP-11, forms oligomers that bridge LD and host macrophage (M Φ) membranes. This KMP-11 induced interaction between LD and M Φ depends on the variations in cholesterol (CHOL) and ergosterol (ERG) contents in their respective membranes. These variations are crucial for the subsequent steps of invasion, including (a) the initial attachment, (b) CHOL transport from M Φ to LD, and (c) detachment of LD from the initial point of contact through a liquid ordered (Lo) to liquid disordered (Ld) membrane-phase transition. To validate the importance of KMP-11, we generate KMP-11 depleted LD, which fail to attach and invade host M Φ . Through tryptophan-scanning mutagenesis and synthesized peptides, we develop a generalized mathematical model, which demonstrates that the hydrophobic moment and the symmetry sequence code at the membrane interacting protein domain are key factors in facilitating the membrane phase transition and, consequently, the host cell infection process by *Leishmania* parasites.

The authors have addressed all minor editorial requests.

Dr. Krishnananda Chattopadhyay
CSIR-Indian Institute of Chemical Biology
4, Raja S.C. Mullick Road
Kolkata 700032
India

Dear Krish,

I am very pleased to accept your manuscript for publication in the next available issue of EMBO reports. Thank you for your contribution to our journal.

Best regards,

Martina
